# The BAF chromatin remodeler synergizes with RNA polymerase II and transcription factors to evict nucleosomes

**Sandipan Brahma** [1,3] ✉ **& Steven Henikoff** [1,2] ✉

Chromatin accessibility is a hallmark of active transcription and entails ATP-dependent nucleosome remodeling, which is carried out by complexes such as Brahma-associated factor (BAF). However, the mechanistic links between transcription, nucleosome remodeling and chromatin accessibility are unclear. Here, we used a chemical–genetic approach coupled with time-resolved chromatin profiling to dissect the interplay between RNA Polymerase II (RNAPII), BAF and DNA-sequence-specific transcription factors in mouse embryonic stem cells. We show that BAF dynamically unwraps and evicts nucleosomes at accessible chromatin regions, while RNAPII promoter-proximal pausing stabilizes BAF chromatin occupancy and enhances ATP-dependent nucleosome eviction by BAF. We find that although RNAPII and BAF dynamically probe both transcriptionally active and Polycomb-repressed genomic regions, pluripotency transcription factor chromatin binding confers locus specificity for productive chromatin remodeling and nucleosome eviction by BAF. Our study suggests a paradigm for how functional synergy between dynamically acting chromatin factors regulates locus-specific nucleosome organization and chromatin accessibility.

The positioning of nucleosomes relative to gene regulatory elements such as promoters and enhancers is pivotal in transcription regulation. Nucleosomes occlude DNA sequences from transcription factors (TFs) and prevent loading of RNAPII and basal transcription machineries[1,2]. Regulatory elements of transcriptionally active genes are typically associated with nucleosome-depleted regions (NDRs) that are accessible to protein factors[3]. Identification of such accessible regions is customary for inferring transcription activity[4]. However, recent studies indicate that transcriptionally active regulatory elements do not remain stably nucleosome-depleted at steady-state[5–8]. Instead, a dynamic cycle of nucleosome loading and eviction ensures that most regulatory elements are never completely occluded within a cell population[5,6,9].

In general, the causal relationship between chromatin accessibility and transcription remains poorly understood[6,10,11]. Current models posit that TFs recruit transcriptional activators such as ATP-dependent nucleosome remodelers that evict nucleosomes to facilitate RNAPII binding[12,13]. However, how factors functionally cooperate in vivo remains unclear[8,14–16]. Several TFs and ATP-dependent remodelers bind to chromatin transiently in living cells, with residence times as short as a few seconds[17–19]. Genome-wide analyses of RNAPII and nucleosome occupancy in *Drosophila* have shown that promoter-proximal pausing of RNAPII counteracts promoter nucleosome occupancy, and, therefore, may stabilize NDRs[20]. In mammalian cells, promoter and enhancer accessibility is consistently associated with paused RNAPII downstream of the NDRs[21,22]. These studies suggest a role

[1]Basic Sciences Division, Fred Hutchinson Cancer Center, Seattle, WA, USA. [2]Howard Hughes Medical Institute, Seattle, WA, USA. [3]Present address: Department of Genetics, Cell Biology & Anatomy, University of Nebraska Medical Center, Omaha, NE, USA. ✉e-mail: sbrahma@unmc.edu; steveh@fredhutch.org

for RNAPII pausing in promoter and enhancer nucleosome eviction. RNAPII pausing was first described at *Drosophila* heat shock genes, where RNAPII initially incorporates the first 25–50 ribonucleotides and then 'pauses', while its active site remains engaged with DNA[23]. Activating signals such as heat shock rapidly trigger transcription elongation with RNAPII traversing the gene body[24]. Paused and elongating RNAPII are distinguished by phosphorylation of a heptapeptide repeat within the RPB1 subunit C-terminal domain at serine 5 (RNAPII-S5P) or serine 2 (RNAPII-S2P), respectively. Despite widespread evidence of RNAPII pausing in animals, its functional roles remain unclear.

We hypothesized that RNAPII pausing may facilitate ATP-dependent nucleosome remodeling to form NDRs. In animals, SWI/SNF (SWItch independent/Sucrose Non-Fermenting) family remodelers, such as the mammalian BAF complex, evict nucleosomes from active gene promoters and enhancers[9]. BAF consists of the catalytic subunit BRG1 (Brahma-related gene 1; also known as SMARCA4) or BRM (Brahma; also known as SMARCA2) and 15–20 additional subunits, most of which are evolutionarily conserved. At least three biochemically distinct BAF complexes have been identified: canonical BAF, noncanonical or ncBAF, and PBAF, which contains a polybromo protein subunit[25,26]. Consistent with their fundamental roles in regulating nucleosome organization, BAF complexes are essential for almost all developmental gene regulation, and BAF subunits are recurrently mutated in more than 20% of human cancers[27]. We have previously found that the *Saccharomyces cerevisiae* SWI/SNF remodeler RSC (Remodeling the Structure of Chromatin), which is similar to mammalian PBAF, is associated with partially unwrapped nucleosomal intermediates at transcriptionally active gene promoters[5]. We further showed that the general-regulatory TFs Abf1 and Reb1 bind their cognate sequence motifs within these partially unwrapped nucleosomes and proposed that a dynamic cycle of nucleosome formation and depletion characterizes transcriptionally active promoters[5,28]. However, the potential role of RNAPII in these dynamic processes was unknown.

In this study, we used highly specific small molecule inhibitors to block RNAPII at either transcription initiation or elongation to determine kinetic changes in RNAPII, BAF and nucleosome occupancy in mouse embryonic stem cells (mESCs). We show that RNAPII promoter-proximal pausing promotes BAF occupancy and ATP-dependent nucleosome remodeling, leading to enhanced nucleosome eviction and DNA accessibility. We find that although RNAPII and BAF engage chromatin genome-wide, including at developmentally repressed genes, effective chromatin remodeling occurs only at active regulatory elements where coincident binding of DNA-sequence-specific TFs drives nucleosome eviction. Our study broadly explains how modulating the dynamics of chromatin factors can result in altered chromatin structure and gene expression, such as in development and in cancer.

## Results

### RNAPII promoter-proximal pausing promotes BAF occupancy

We used the Cleavage Under Targets and Tagmentation (CUT&Tag)[29] method to determine genome-wide RNAPII and BAF occupancy in mESCs. For RNAPII, we chose antibodies against RNAPII-S5P (paused), or RNAPII-S2P (elongating), or another core subunit, RPB3 (all RNAPII). CUT&Tag showed strong RNAPII occupancy near the transcription start sites (TSSs) of genes and promoter-distal regions corresponding to annotated transcriptional enhancers (Extended Data Fig. 1a–d). RNAPII-occupied sites showed strong enrichment for histones containing post-translational modifications (PTMs) characteristic of active transcription[11], for example, histone H3 with mono- or tri-methylated lysine 4 (H3K4me1 and H3K4me3), which mark enhancers and promoters, respectively, and histone H3 with acetylated lysine 27 (H3K27ac), characteristic of both active enhancers and promoters. In contrast, promoters enriched for histone H3 with the repressive

lysine 27 trimethylation (H3K27me3) showed little RNAPII occupancy (Extended Data Fig. 1a–d). To determine BAF occupancy, we applied CUT&Tag by targeting the BRG1 subunit containing the catalytic ATP hydrolysis and DNA translocation domains essential for nucleosome remodeling[30]. As the alternative BRM ATPase subunit is not expressed in mESCs, BRG1 represents all functional BAF complexes in these cells: the mESC-specific esBAF, ncBAF and PBAF. CUT&Tag revealed that BAF occupies the same genomic regions as RNAPII and showed the highest association with RNAPII-S5P (Fig. 1a and Extended Data Fig. 1a–e). We have recently demonstrated that low-salt tagmentation conditions for RNAPII-S5P CUT&Tag produce high-resolution maps of transcription-coupled accessible regulatory sites including active promoters and enhancers[21,22]. RNAPII-S5P-associated accessible chromatin regions in mESCs, hereby referred to as S5P CUTAC (Cleavage Under Targeted Accessible Chromatin) peaks, mapped within NDRs determined by micrococcal nuclease (MNase) digestion of chromatin, and immediately upstream of nascent RNA TSSs mapped by START-seq[31,32] (Extended Data Fig. 1a,f). Therefore, S5P CUTAC peaks represent NDRs upstream of genic promoter TSSs and start sites of enhancer RNA transcription, selectively in cells where these loci are occupied by RNAPII. When aligned over S5P CUTAC peaks, BRG1 CUT&Tag showed strong BAF occupancy, consistent with the function of BAF in generating and/or maintaining the NDRs (Fig. 1a and Extended Data Fig. 1a,b).

To determine whether and how RNAPII regulates BAF, we used fast-acting cell-permeable small molecule inhibitors to acutely inhibit transcription at distinct stages and modulate RNAPII dynamics (Fig. 1b). The inhibitors Triptolide, Flavopiridol and Actinomycin D affect the RNAPII transcription cycle in specific and distinct ways[33]. The natural diterpene triepoxide Triptolide inhibits ATPase activity of the XPB subunit of TFIIH, which is a part of the transcription pre-initiation complex of RNAPII with several additional cofactors. Triptolide prevents transcription initiation by blocking ATP-dependent XPB activity to translocate DNA into the RNAPII active site, and so induces a fast proteasomal degradation of RNAPII subunit RPB1 (ref. 34). We subjected mESC cultures to a time-course of treatment with 10 µM Triptolide. Although colony morphology and cell viability were unaffected for up to 2 h, cells started to dislodge and lose viability at 4 h. CUT&Tag showed a dramatic and rapid loss of RNAPII-S5P genome-wide (Fig. 1c,d and Extended Data Fig. 2a). Occupancy was reduced by 50% within 30 min, and almost all RNAPII-S5P was lost within 2 h. Here and elsewhere, we used spike-in calibration, which is vital for quantifying such genome-wide differences. BRG1 showed a similar rapid genome-wide loss upon Triptolide treatment (Fig. 1c,d and Extended Data Fig. 2b), implying that RNAPII and transcription initiation (either RNAPII loading or promoter-proximal pausing) promotes BAF chromatin occupancy. Interestingly, BRG1 was lost at a slower rate than RNAPII-S5P, suggesting that BAF may bind independently of RNAPII and have distinct chromatin binding dynamics (Fig. 1d).

To distinguish whether BAF occupancy is facilitated by RNAPII loading or by RNAPII pausing, we used inhibitors that accumulate paused RNAPII by inhibiting productive elongation. The semi-synthetic flavonoid Flavopiridol inhibits the transcription elongation factor pTEFb subunit CDK9, which phosphorylates RNAPII-S2P and increases RNAPII pausing genome-wide[33,35]. As expected, CUT&Tag showed a rapid increase in RNAPII-S5P over a 4-h time-course following 1 µM Flavopiridol treatment. A corresponding rapid increase in BRG1 CUT&Tag established that RNAPII pausing promotes BAF chromatin occupancy (Fig. 1e,f and Extended Data Fig. 2d–f). We observed similar rapid increases in both RNAPII-S5P and BRG1 over a time-course of 5 µg ml$^{-1}$ Actinomycin D treatment (Fig. 1g,h and Extended Data Fig. 2d–f). While Triptolide and Flavopiridol both directly affect RNAPII, Actinomycin D inhibits transcription by a distinct mechanism as it intercalates within unwound DNA strands at the active site of RNAPII and acts as a roadblock to RNAPII elongation[33]. Live-cell single-molecule imaging has shown that Actinomycin D increases the residence time of RNAPII-S5P

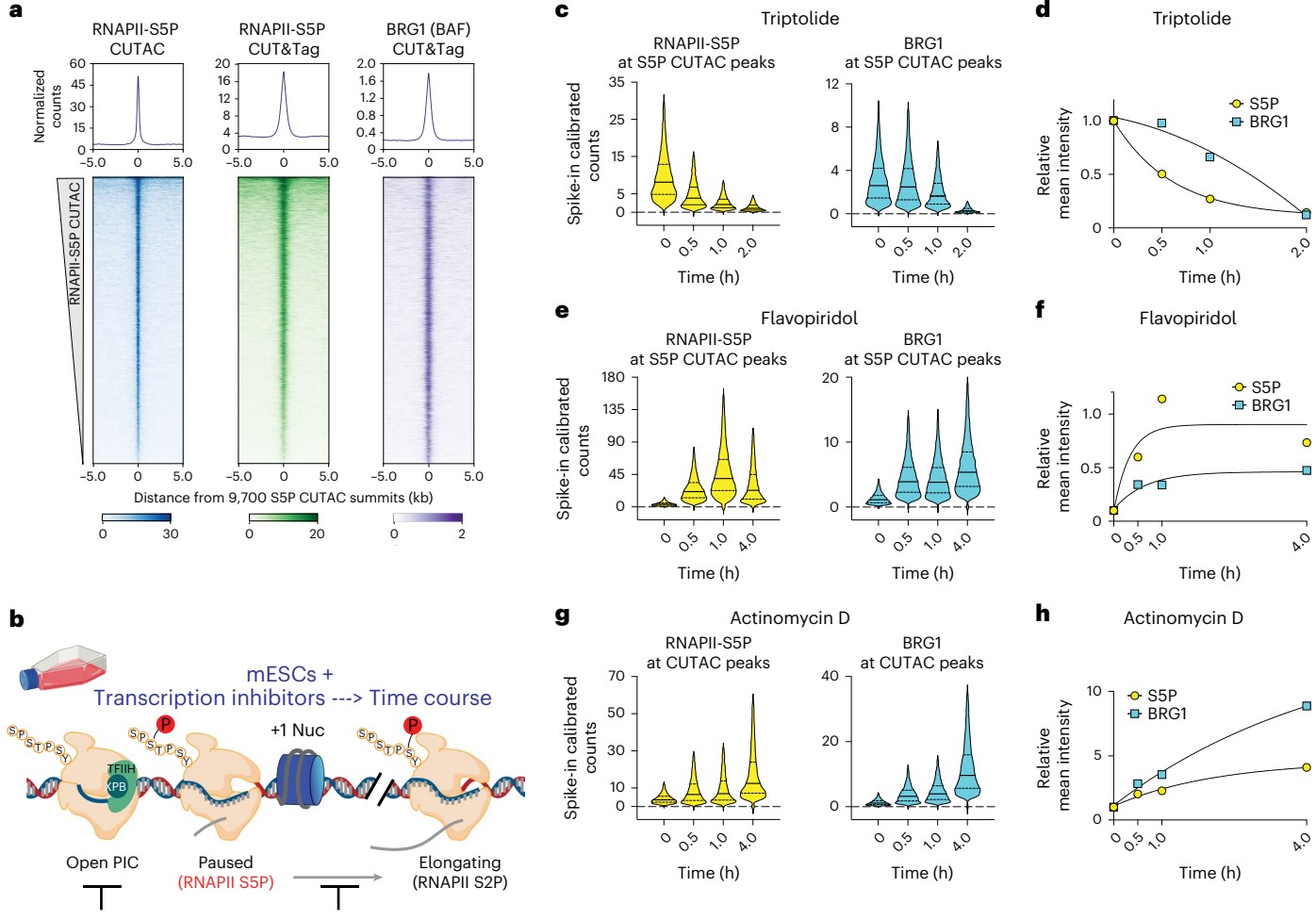

**Fig. 1 | RNAPII promoter-proximal pausing promotes BAF chromatin occupancy. a**, Heatmaps (bottom) and average plots (top) comparing chromatin accessibility assayed by S5P CUTAC, with RNAPII-S5P and BRG1 occupancy (CUT&Tag), relative to the primary peaks (summits) of S5P CUTAC and sorted by decreasing accessibility (CUTAC signal). **b**, Schematic showing distinct stages in RNAPII transcription that are inhibited by drugs used in this study. **c,e,g**, Violin plots of spike-in calibrated CUT&Tag signal distribution comparing RNAPII-S5P and BRG1 occupancy over S5P CUTAC peaks at time points after drug treatments for transcription inhibition: Triptolide (**c**), Flavopiridol (**e**) and Actinomycin D (**g**). Median values (solid lines), upper and lower quartiles (broken lines) and outliers were calculated using the Tukey method; $n = 9,700$. **d,f,h**, Fold changes in mean RNAPII-S5P and BRG1 occupancy (spike-in calibrated CUT&Tag) over S5P CUTAC peaks at time points after drug treatments: Triptolide (**d**), Flavopiridol (**f**) and Actinomycin D (**h**). All datasets are representative of at least two biological replicates. The RNAPII illustration was created with BioRender.com. Nuc, nucleosome; PIC, pre-initiation complex.

on chromatin, while Flavopiridol does not[36]. This implies that Actinomycin D 'traps' paused RNAPII, while RNAPII that is inhibited from elongating by Flavopiridol is displaced. In Flavopiridol, CUT&Tag signal for RNAPII-S5P increased early but plateaued, while in Actinomycin D RNAPII-S5P occupancy continued to increase (Fig. 1f,h). BRG1 CUT&Tag showed the same patterns as RNAPII-S5P, further demonstrating the strong correspondence between BAF and paused RNAPII for chromatin occupancy (Fig. 1f–h and Extended Data Fig. 2f), and trapped RNAPII in Actinomycin D resulted in a stronger buildup of BRG1 over time (Fig. 1h).

Next, we compared RNAPII-S5P and BAF occupancy at gene promoters and promoter-distal regulatory regions. To obtain a comprehensive catalog of distal regulatory sites, we combined S5P CUTAC peaks and peaks of the TFs NANOG and SOX2 that are promoter-distal, or more than 2 kilobases (kb) away from annotated TSSs. We determined TF binding by CUT&RUN (Cleavage Under Targets & Release Using Nuclease) mapping. These TFs along with OCT4 (POU5F1) form the embryonic stem cell (ESC) core pluripotency TF network and strongly occupy mESC enhancers[37]. We did not include OCT4 peaks in this analysis as it was relatively less enriched in CUT&RUN. Compared with promoters (TSS ± 1 kb), distal sites showed stronger enrichment of H3K4me1 and H3K27ac, and reduced H3K4me3, characteristic of transcriptionally active enhancers (Extended Data Fig. 1d). Although RNAPII-S5P and BRG1 showed higher occupancy at promoter-distal sites (Extended Data Fig. 1d), treatment with the transcription inhibitors resulted in very similar kinetics at both sets of regions, confirming that paused RNAPII promoting BAF occupancy occurs broadly across the mESC genome (Extended Data Fig. 2c,g,h). As a control, and to rule out the possibility that CUT&Tag may nonspecifically capture accessible DNA at RNAPII bound sites, we compared H3K27ac levels in the presence of Actinomycin D. Interestingly, CUT&Tag showed a sharp decline in H3K27ac at RNAPII-S5P CUTAC sites within 30 min, which is opposite to the effects observed for RNAPII-S5P and BRG1 (Extended Data Fig. 2i).

## BAF unwraps and evicts nucleosomes in an ATP-dependent manner

Having found that paused RNAPII promotes BAF chromatin occupancy, we next investigated its effect on nucleosome occupancy

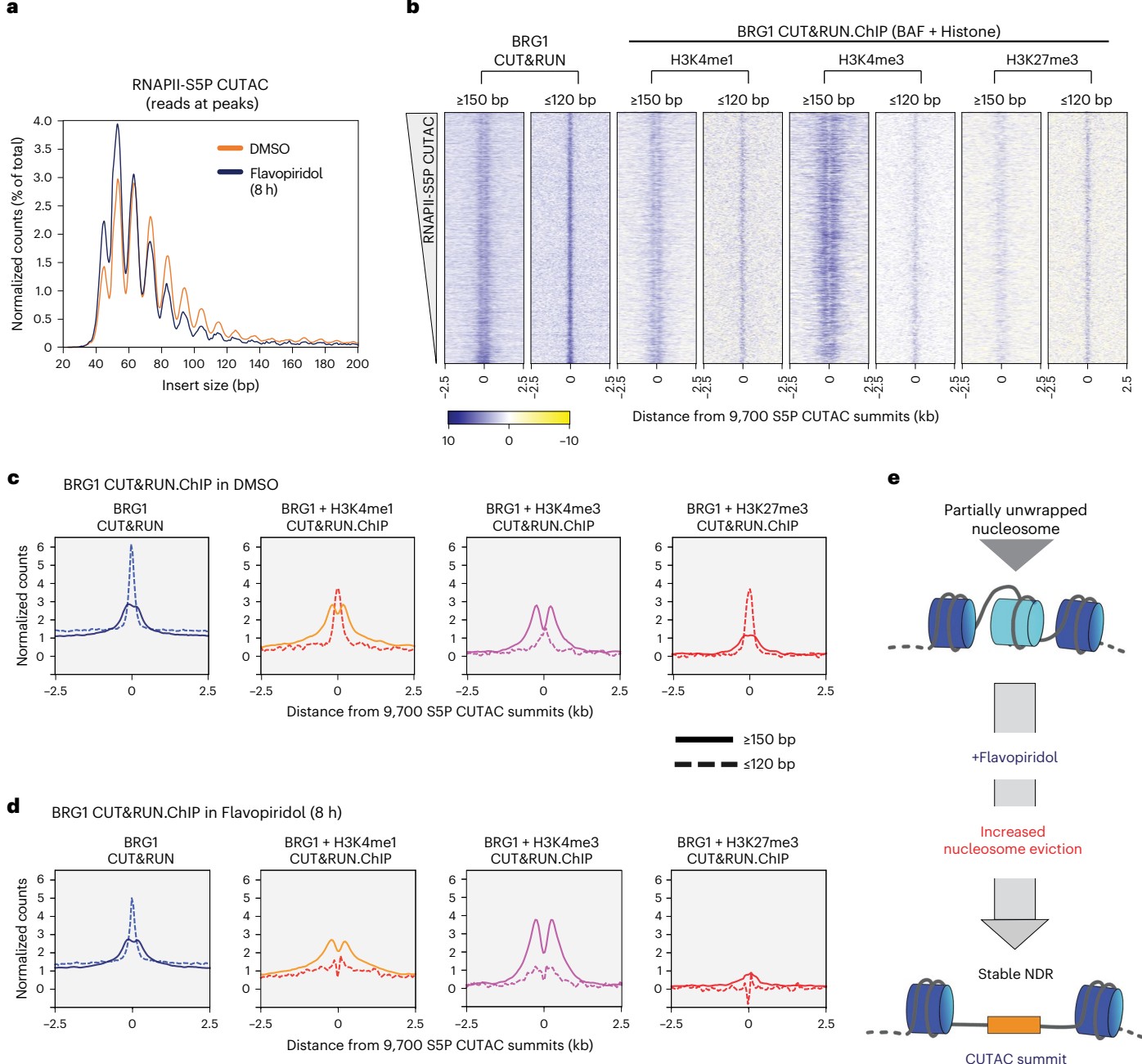

**Fig. 2 | Enrichment of BAF increases nucleosome eviction. a**, Comparison of chromatin structure and chromatin accessibility by means of S5P CUTAC fragment size distribution over peaks (promoter and enhancer NDR spaces) in cells treated with DMSO (control) and Flavopiridol. Peaks were called with DMSO control. **b**, Heatmaps comparing nucleosomal (≥150-bp reads) and subnucleosomal (≤120-bp reads) protection by BAF (BRG1 CUT&RUN) and BAF-associated histones (BRG1 CUT&RUN.ChIP) in untreated (DMSO control) cells. Heatmaps were plotted relative to S5P CUTAC summits and sorted by decreasing accessibility (CUTAC signal). CUT&RUN.ChIP heatmaps show enrichment over IgG isotype control (for ChIP). **c,d**, Enrichment of nucleosomal (≥150-bp, solid lines) and subnucleosomal (≤120-bp, broken lines) reads from BRG1 CUT&RUN and CUT&RUN.ChIP experiments, relative to S5P CUTAC summits, in DMSO- (**c**) or Flavopiridol-treated (**d**) cells. **e**, Flavopiridol treatment causes eviction of partially unwrapped nucleosomes through enrichment of BAF, leading to NDR persistence. All datasets are representative of at least two biological replicates. DMSO, dimethylsulfoxide.

using RNAPII-S5P CUTAC. Analysis of DNA-insert sizes in CUTAC libraries can show how closely two Tn5 molecules could integrate into the same DNA, providing a protein footprint. RNAPII-S5P CUTAC restricts this analysis to genomic loci occupied by RNAPII. RNAPII-S5P CUTAC produced fragments that were mostly shorter than 120 base pairs (bp) (>85% of total reads overlapping CUTAC peaks; Fig. 2a and Extended Data Fig. 3a,b; dimethylsulfoxide only), suggesting that transcriptionally active gene promoters and distal regulatory

regions are mostly occupied by proteins with footprints smaller than nucleosomes.

We previously introduced the CUT&RUN.ChIP technique to demonstrate that the PBAF-like RSC complex in budding yeast binds partially unwrapped nucleosomal intermediates at more than two-thirds of all promoter NDR spaces[5]. We used this method to ask whether BAF is similarly associated with unwrapped nucleosomes in mESCs. We performed CUT&RUN targeting BRG1, followed by

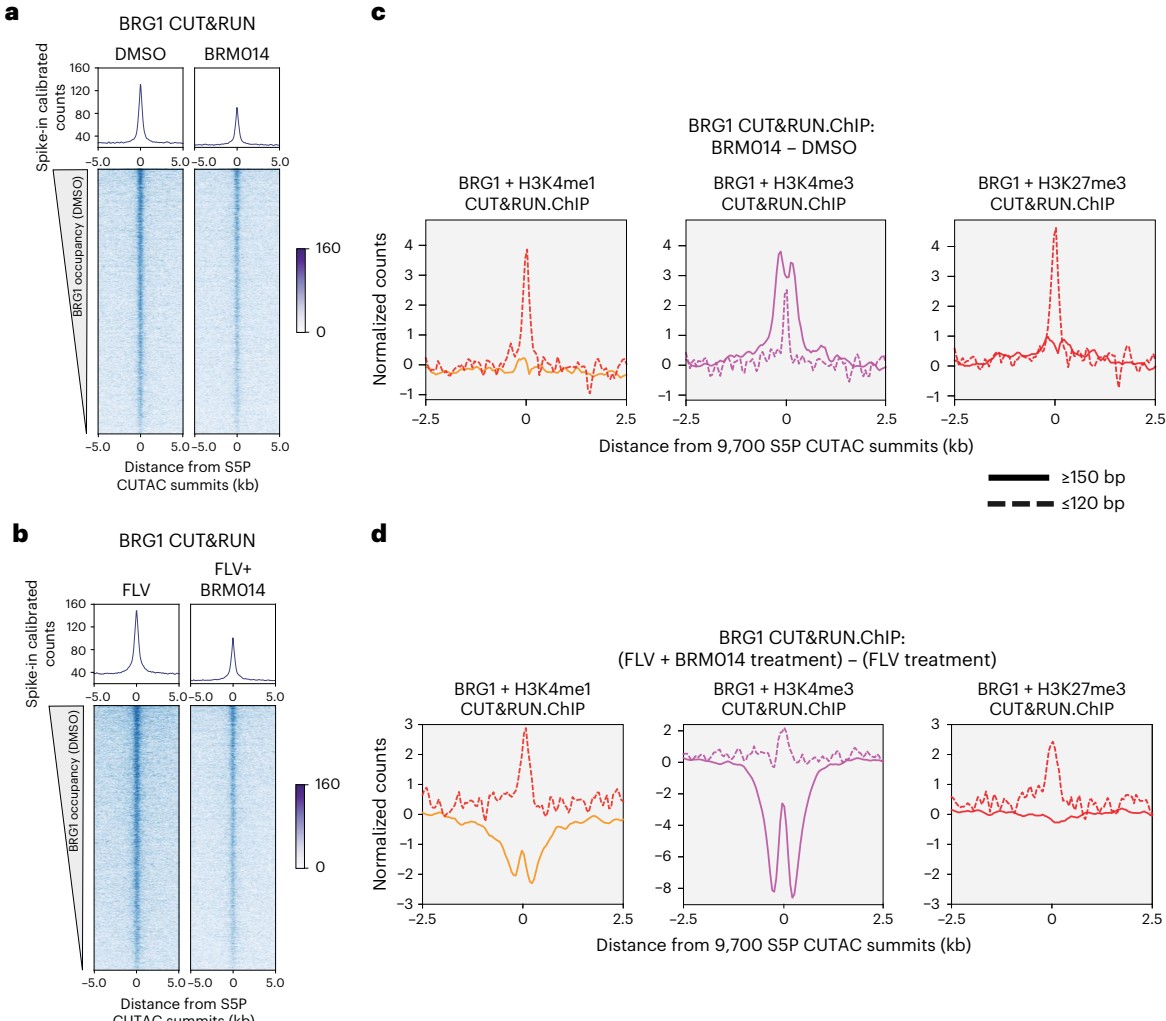

**Fig. 3 | Nucleosome eviction by BAF is ATP-dependent. a,b,** Heatmaps (bottom) and average plots (top) comparing BRG1 occupancy by CUT&RUN upon 8 h of BRM014 treatment versus DMSO control (**a**), and dual inhibition with BRM014 plus Flavopiridol versus Flavopiridol only (**b**). Data are plotted relative to the primary peaks (summits) of S5P CUTAC and sorted by decreasing BRG1 CUT&RUN in DMSO. **c,d,** Enrichment of nucleosomal- (≥150-bp, solid lines) and subnucleosomal- (≤120-bp, broken lines) reads from BRG1 CUT&RUN.ChIP experiments relative to S5P CUTAC summits, in BRM014 over DMSO control (**c**) and upon dual inhibition with Flavopiridol and BRM014 over Flavopiridol only (**d**). Normalized counts of reads in DMSO were subtracted from BRM014 (**c**), and normalized counts of reads in Flavopiridol only were subtracted from the dual inhibitor treatment (**d**). All datasets are representative of two biological replicates. FLV, Flavopiridol.

chromatin immunoprecipitation (ChIP) for histone PTMs, and analyzed nucleosomal- (≥150 bp) and subnucleosomal- (≤120 bp) sized DNA fragments protected from Protein A–MNase (pA–MN) digestion. This analysis showed that subnucleosomal particles protecting <120 bp of DNA over the S5P CUTAC peaks contain both BRG1 and histones, implying that these are partially unwrapped nucleosomal intermediates in BAF remodeling (Fig. 2b,c). In contrast, nucleosomes flanking the CUTAC peaks were fully wrapped and protected >150 bp of DNA (Fig. 2c, compare solid versus broken lines). BAF-associated partially unwrapped nucleosomes were enriched for H3K4me1 and H3K4me3, as well as H3K27me3 catalyzed by the Polycomb Repressive complex 2 (PRC2). Partially unwrapped nucleosomes were enriched immediately upstream of promoter TSSs and at promoter-distal sites (Extended Data Fig. 3c,e). BAF-associated nucleosomes were relatively more enriched for H3K4me3 at promoters and H3K4me1 at distal sites, consistent with the differential enrichment of the histone PTMs in these regions (Extended Data Fig. 1d).

Strikingly, treating mESCs with Flavopiridol resulted in a dramatic depletion of the partially unwrapped nucleosomal intermediates, while the flanking fully wrapped nucleosomes were retained (Fig. 2d). Analysis of RNAPII-S5P CUTAC fragment sizes upon Flavopiridol treatment for 8 h revealed a shift towards shorter fragments, implying enhanced DNA accessibility and indicating that partially unwrapped nucleosomes were evicted, although total S5P CUTAC signals remained the same (Fig. 2a and Extended Data Fig. 3a,b). These effects could be observed with 4-h Flavopiridol treatment, but 8 h showed more robust results. Colony morphologies were identical with dimethylsulfoxide controls and there was no reduction in cell numbers. Loss of the partially unwrapped nucleosomes was more pronounced at distal sites compared with promoter NDRs (Extended Data Fig. 3d,f), consistent with previous reports that BAF primarily maintains DNA accessibility at enhancer regions in mESCs[9,38–40], and likely opposing action of other remodelers competing with BAF to regulate promoter nucleosome organization[41,42]. BAF binding and evicting H3K27me3 nucleosomes is consistent with its role in opposing H3K27me3 and PRC2 to activate transcription[43], despite mild repression of BAF remodeling by H3K27me3 seen in vitro[44]. We speculate that this difference may be explained by BAF inhibition with both histone tails modified in vitro versus highly dynamic promoter

nucleosomes allowing one of the two histone tails to be methylated, which may not be inhibitory.

Taken together, BRG1 CUT&RUN.ChIP and RNAPII-S5P CUTAC before and after Flavopiridol treatment show that enrichment of paused RNAPII and subsequently elevated BAF occupancy leads to increased nucleosome eviction to form stable NDRs (Fig. 2e). To confirm that nucleosome eviction is indeed catalyzed by BAF, we used BRM014, a small molecule inhibitor of BRG1 ATPase activity[9,40,45]. We treated mESCs with either 10 μM BRM014-alone or BRM014 in combination with 1 μM Flavopiridol. CUT&RUN showed that BRG1 binding is moderately reduced upon ATPase inhibition (Fig. 3a,b). BRG1 CUT&RUN.ChIP showed selective enrichment of subnucleosomal fragments in cells treated with BRM014 relative to dimethylsulfoxide controls, implying that BAF-associated partially unwrapped nucleosomes were preferentially retained (Fig. 3c). Dual inhibition with BRM014 and Flavopiridol also resulted in enrichment of BAF-associated partially unwrapped nucleosomes over Flavopiridol alone, confirming that their eviction is dependent on BRG1 ATPase activity (Fig. 3d). Retention of the partially unwrapped nucleosomes in BRM014-alone suggests that unlike nucleosome eviction, nucleosome unwrapping by BAF may not be ATP-dependent, consistent with analysis of nucleosome disruption by human SWI/SNF in vitro[46], and may result from disruption of histone–DNA contacts due to compensating interactions of the remodeler with nucleosomal DNA and histones[47]. Intriguingly, dual inhibition resulted in a striking depletion of nucleosome-sized fragments flanking the subnucleosomal intermediates (Fig. 3d, solid lines), possibly attributable to compensatory action of other nucleosome remodelers and chromatin regulators whose occupancy might also be stabilized by elevated paused RNAPII-S5P (ref. 48).

## RNAPII and BAF dynamically probe facultative heterochromatin

In multicellular eukaryotes, repressed genes are packaged into nucleosome-dense constitutive or facultative heterochromatin containing H3K9me3 or H3K27me3, respectively. To compare transcriptionally active and PRC2-repressed facultative heterochromatin regions, we categorized mESC promoters as RNAPII-S5P enriched (active) or H3K27me3 enriched (PRC2-repressed) (Extended Data Fig. 4a). PRC2-repressed promoters showed low occupancy of RNAPII-S5P and BRG1 by CUT&Tag, while H3K27me3 was enriched over the promoter and gene-body regions (Fig. 4a,b). Consistent with previous work, PRC2-repressed promoters showed much reduced DNA accessibility by RNAPII-S5P CUTAC (Fig. 4b)[49].

Similar to S5P CUTAC sites and RNAPII-high promoters, treating cells with Flavopiridol or Actinomycin D gradually increased RNAPII-S5P and BRG1 at H3K27me3-high promoters including the *Hox* gene clusters (Fig. 4a,c,d and Extended Data Fig. 4b,c). Although BAF opposes PRC2 repression[43], we observed only a slight difference in H3K27me3 occupancy and RNAPII-S5P CUTAC fragment size distribution in Flavopiridol (Fig. 4e,f and Extended Data Fig. 4d), suggesting that increased RNAPII-S5P and BAF occupancy is not sufficient for persistent

chromatin remodeling and NDR maintenance at PRC2-repressed promoters.

These data also show that RNAPII and BAF are not excluded from PRC2-repressed genes in mESCs. Rather, RNAPII and BAF likely continuously probe PRC2-repressed chromatin and transiently initiate transcription, consistent with low-abundance transcripts detectable from PRC2-repressed chromatin[50] and reduced RNAPII burst frequency[51]. In contrast to PRC2-repressed facultative heterochromatin, RNAPII-S5P and BRG1 appear to be excluded from H3K9me3-marked constitutive heterochromatin, where we did not observe occupancy even upon the drug treatments (Extended Data Fig. 4e).

## TF–chromatin binding drives nucleosome eviction

Since BAF and RNAPII probe both transcriptionally active and PRC2-repressed chromatin, what determines their specificity for persistent chromatin remodeling to maintain NDRs? In mESCs, these regions have differential binding of DNA-sequence-specific TFs NANOG, SOX2, OCT4 and KLF4, which are master regulators of ESC self-renewal and pluripotency. CUT&RUN mapping confirmed that these pluripotency TFs strongly occupy S5P CUTAC sites and RNAPII-high (transcriptionally active) promoters but not PRC2-repressed H3K27me3-high promoters, showing strong correspondence with RNAPII-S5P and BRG1 occupancy (Fig. 5a,b and Extended Data Fig. 5a). Mechanistically, pluripotency TFs such as OCT4 and SOX2 can bind nucleosomes in vitro[12,52], and depletion experiments show that they have critical roles in maintaining chromatin accessibility in vivo[53,54], part of which could be mediated by facilitating BAF recruitment via direct protein–protein interaction or other mechanisms[55–59]. However, ATP-dependent remodeling by BAF is in turn required for chromatin accessibility and pluripotency TF binding for ESC self-renewal[9,42,54,60], during reprogramming[61] and in developing blastocysts[62]. These studies highlight a cooperativity between pluripotency TFs and BAF in maintaining accessible chromatin regions, but the mechanism has remained unclear. To characterize this cooperativity, we used BRG1 CUT&RUN.ChIP and examined binding sites of pluripotency TFs, for example, NANOG (Extended Data Fig. 5b), and observed enrichment of subnucleosomal fragments over the TF foci. Analysis of the correlation between pluripotency TF–chromatin binding affinity as CUT&RUN signal intensity and S5P CUTAC chromatin accessibility showed positive associations, with moderately high correlation coefficients for NANOG and KLF4, lower for SOX2 and weak correlation for OCT4 (Extended Data Fig. 5d). These data together with the enrichment of BAF-associated partially unwrapped nucleosomes over TF foci suggest that pluripotency TFs, particularly NANOG and KLF4, may capture transiently exposed sites within these partially unwrapped nucleosomes to drive nucleosome eviction in cooperation with BAF. This model predicts that increased TF DNA-binding affinity or TF concentration would result in enhanced nucleosome depletion and chromatin accessibility[63].

To test this model, we cultured mESCs in a medium that moderately increases NANOG expression, thereby avoiding nonphysiological overexpression. mESCs cultured in media containing serum

---

**Fig. 4 | Transcription inhibition shows RNAPII and BAF occupancy at Polycomb (PRC2)-repressed gene promoters. a**, Representative genomic tracks comparing enrichment of histone PTMs, RNAPII-S5P and BRG1 by CUT&Tag at transcriptionally active and PRC2-repressed genes, and changes in RNAPII-S5P and BRG1 occupancy upon Flavopiridol (Flv.) and Actinomycin D (Act.) treatment. RNAPII-S5P and BRG1 CUT&Tag read counts were spike-in calibrated. **b**, Heatmaps (bottom) and average plots (top) comparing histone PTMs (CUT&Tag), chromatin structure (RNAPII-S5P CUTAC and ATAC-seq) and transcriptional activity (START-seq) at RNAPII-enriched (active) and H3K27me3-enriched (PRC2-repressed) promoters. Promoters were grouped based on *k*-means clustering of RNAPII-S5P and H3K27me3 CUT&Tag reads mapping to a 5-kb window around the TSSs of RefSeq-annotated mESC genes (Extended

Data Fig. 4a). **c,d**, Violin plots of spike-in calibrated CUT&Tag signal distribution comparing RNAPII-S5P (**c**) and BRG1 (**d**) occupancy over PRC2-repressed promoter TSSs ± 1 kb at time points after drug treatments. **e**, Violin plot comparing spike-in calibrated H3K27me3 CUT&Tag at PRC2-repressed promoter TSSs ± 1 kb in cells treated with DMSO and Flavopiridol. Median values (solid lines), upper and lower quartiles (broken lines) and outliers were calculated using the Tukey method; *n* = 2,767. Numbers on top of the violin plots are mean values. **f**, S5P CUTAC fragment size distribution to compare chromatin accessibility at PRC2-repressed promoter TSSs ± 1 kb in cells treated with DMSO and Flavopiridol. All datasets are representative of at least two biological replicates. ATAC-seq, assay for transposase-accessible chromatin using sequencing; Mb, megabase.

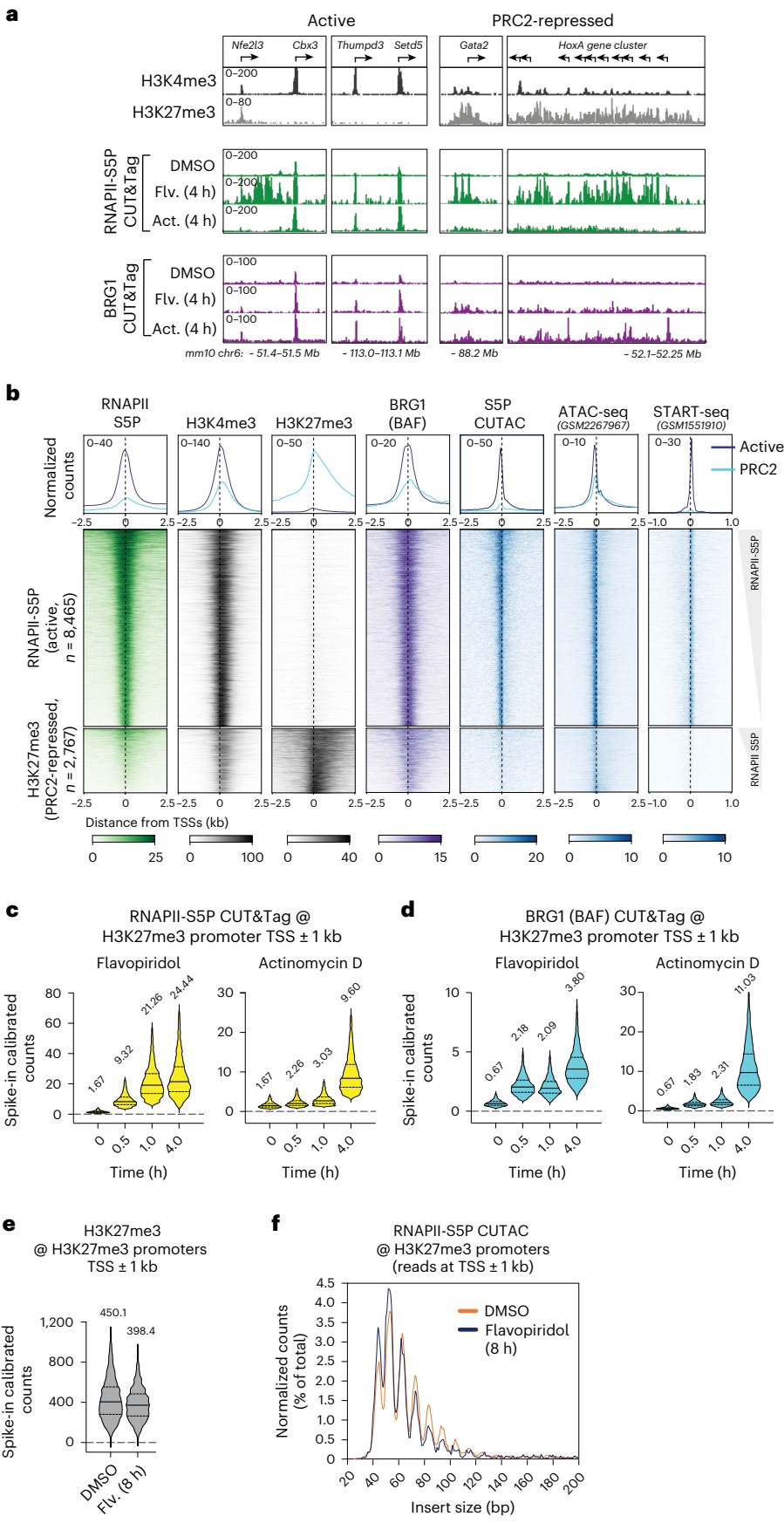

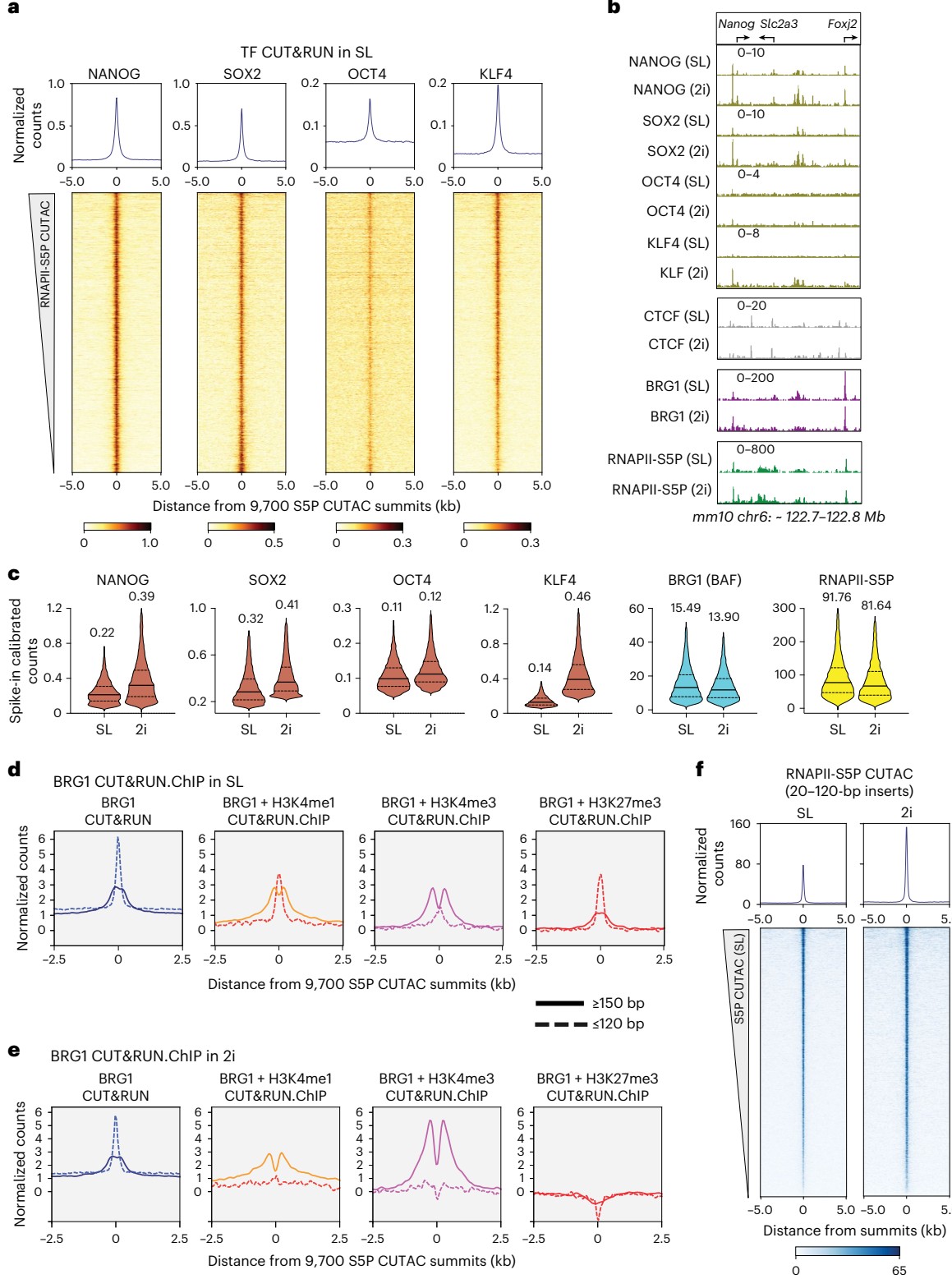

**Fig. 5 | Upregulation of pluripotency TFs results in enhanced nucleosome eviction. a,** Heatmaps (bottom) and average plots (top) of pluripotency TF CUT&RUN reads relative to RNAPII-S5P CUTAC summits, showing TF binding at sites of DNA accessibility. Heatmaps were sorted by decreasing accessibility (CUTAC signal). **b,** Representative genomic tracks comparing occupancy of TFs (CUT&RUN), BRG1 (CUT&Tag) and RNAPII-S5P (CUT&Tag) in SL and 2i conditions. All datasets were spike-in calibrated. **c,** Violin plots of spike-in calibrated CUT&RUN (TF) and CUT&Tag (BRG1 and RNAPII-S5P) signal distribution comparing factor occupancy over RNAPII-S5P CUTAC peaks in 2i versus SL. Median values (solid lines), upper and lower quartiles (broken lines) and outliers were calculated using the Tukey method; $n$ = 9,700. Numbers on top are mean values. **d,e,** Enrichment of nucleosomal (≥150-bp, solid lines) and subnucleosomal (≤120-bp, broken lines) reads from BRG1 CUT&RUN and CUT&RUN.ChIP experiments, relative to S5P CUTAC summits, in SL (**d**) and 2i (**e**). **f,** Heatmaps (bottom) and average plots (top) of RNAPII-S5P CUTAC (20–120-bp reads only) relative to S5P CUTAC summits, comparing chromatin accessibility in 2i versus SL. All datasets are representative of at least two biological replicates.

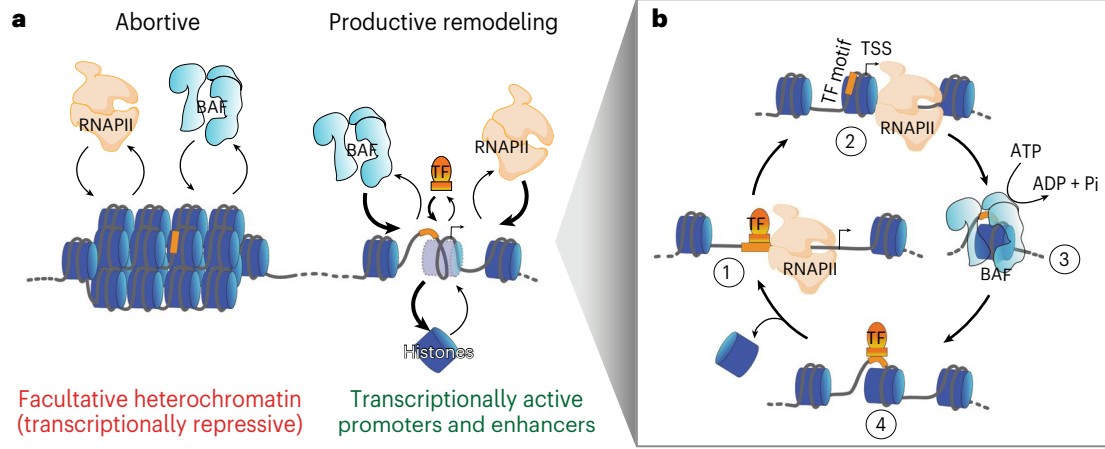

**Fig. 6 | RNAPII, BAF and DNA-sequence-specific TFs work synergistically in a dynamic cycle for productive chromatin remodeling and nucleosome eviction. a**, Model showing that RNAPII and BAF dynamically engage chromatin in an abortive manner (left-hand side) and require chromatin binding by DNA-sequence-specific TFs for productive chromatin remodeling and histone eviction to form/maintain an NDR (right-hand side). Relative thickness of arrows implies enrichment of factor-bound states in transcriptionally active chromatin as observed in steady-state bulk measurement. **b**, Steady-state promoter and enhancer chromatin structures can be explained by a dynamic cycle of nucleosome deposition and eviction and synergistic RNAPII, BAF and TF activity. We speculate that the cycle can start at any step: RNAPII loading at nucleosome-depleted regions (step 1) and transcription initiation (step 2), BAF binding to nucleosomes (step 3) or TF-binding nucleosomes that are partially unwrapped due to spontaneous thermal fluctuations in histone–DNA interactions or BAF binding and remodeling (step 4); and the cycle can continue as long as factor concentrations are high enough. The steps in the cycle facilitate each other, which we propose based on our observations that RNAPII promoter-proximal pausing promotes BAF occupancy and ATP-dependent nucleosome eviction, BAF is associated with partially unwrapped nucleosomes at pluripotency TF-binding sites and upregulated TF protein expression promotes nucleosome eviction by BAF leading to stable NDR formation, which facilitates new RNAPII loading. The RNAPII illustration was created with BioRender.com. Pi, inorganic phosphate.

and leukemia inhibitory factor (serum-LIF or SL condition) mimic a post-implantation embryonic stage (embryonic day 4.5) and express pluripotency TFs[64]. However, NANOG and KLF4 protein levels vary among individual cells in SL culture[65,66]. Dual inhibition of the signaling kinases GSK3β and MEK ('2i' condition) promotes a cellular state closer to the pluripotent pre-implantation epiblast (embryonic day 3.5)[67,68], and leads to increased and consistent NANOG protein levels in individual mESCs[69].

Immunofluorescence imaging and western blotting confirmed NANOG upregulation in 2i compared with SL and showed that KLF4 is also upregulated, but not OCT4 or SOX2 (Extended Data Figs. 5e,f and 6). Although western blotting showed a less than twofold increase in extracted soluble protein, immunofluorescence results indicated stronger increases in nuclear NANOG and KLF4 protein abundance in 2i compared with SL. Consistent with higher protein concentration in cells, CUT&RUN showed more than threefold increase in KLF4 occupancy, close to twofold increase in NANOG occupancy, less for SOX2, but no notable changes in OCT4 and the ubiquitous TF insulator protein CTCF occupancies (Fig. 5b,c and Extended Data Fig. 5g). BRG1 CUT&Tag showed comparable occupancies in SL and 2i (Fig. 5b,c and Extended Data Fig. 5g), implying that pluripotency TF upregulation upon the switch to 2i does not result in BAF upregulation. Nevertheless, BRG1 CUT&RUN.ChIP in 2i (higher TF expression) compared with SL (lower TF expression) showed a striking loss of BAF-associated partially unwrapped nucleosomes over S5P CUTAC sites and pluripotency TF foci (compare Fig. 5d,e and Extended Data Fig. 5b,c), similar to what we observed upon treating cells with Flavopiridol in SL (Fig. 2c,d). Additionally, S5P CUTAC showed a twofold increase in chromatin accessibility in 2i compared with SL (Fig. 5f), despite comparable RNAPII-S5P occupancy (Fig. 5b,c and Extended Data Fig. 5g). We conclude that NANOG and KLF4 capture transient site exposure due to nucleosome unwrapping by BAF to further destabilize and evict nucleosomes. Increased TF abundance drives this process towards stable NDR formation, consistent with the robustness of the 2i condition for mESC pluripotency maintenance.

## Discussion

Our study shows that RNAPII promoter-proximal pausing promotes BAF chromatin occupancy and ATP-dependent nucleosome eviction in mESCs, suggesting that this mechanism helps maintain nucleosome depletion and chromatin accessibility at transcriptionally active gene promoters and enhancers. We found that BAF partially unwraps nucleosomes at sites of pluripotency TF binding. These dynamic 'fragile' nucleosomes show increased susceptibility to MNase digestion in comparison with MNase-resistant nucleosomes genome-wide[5,70]. Our data suggest that pluripotency TFs trap dynamically exposed DNA sequences within these partially unwrapped nucleosomes to further facilitate their invasion by BAF. We propose that TF–chromatin binding acts as a switch, converting abortive BAF remodeling (discussed below) into productive nucleosome eviction and NDR formation (Fig. 6a). Taken together, our study shows that maintaining steady-state chromatin accessibility patterns involves a functional synergy between RNAPII promoter-proximal pausing, BAF nucleosome remodeling and DNA-sequence-specific chromatin binding by TFs. We envision that this synergy involves a continuous cycle of independent but synchronous RNAPII, BAF and TF action to dynamically evict nucleosomes occupying promoter and enhancer NDR spaces (Fig. 6b). This cycle is absent in PRC2-repressed chromatin, which lacks pluripotency TF binding, so RNAPII and BAF do not show stable occupancy at steady-state. However, this dynamic between BAF and RNAPII does not occur in constitutive heterochromatin repressed by heterochromatin-associated proteins and DNA methylation. Our model is consistent with our previous work in budding yeast[5] and recent studies in mESCs and human cells showing continuous requirement of BAF remodeling and pluripotency TF activity for NDR maintenance[9,40,53]. Our study highlights the dynamic events such as widespread scanning by RNAPII and BAF, which are often hidden under the perceived static appearance of promoter chromatin structures.

These dynamics may also be important at specific times in mitotically active cells. Chromatin undergoes major restructuring during DNA replication as nucleosomes are disassembled ahead of the

replication fork and replaced randomly on nascent DNA[71,72]. We speculate that RNAPII, similar to TFs, broadly scans chromatin for exposed promoter DNA and utilizes the window of opportunity in the wake of DNA replication to bind and initiate transcription (Fig. 6b, steps 1 and 2). RNAPII pausing would then promote BAF occupancy (step 3) to clear away nucleosomes that encroach into the NDR space upstream of the paused RNAPII, ensuring subsequent rounds of RNAPII loading. Consistent with our model, analysis of newly replicated chromatin shows that RNAPII binds to newly synthesized DNA strands and initiates transcription before chromatin maturation, and a delay in the maturation of repressive chromatin facilitates TF binding and chromatin activation[72–74]. In *Drosophila* S2 cells, replication fork passage results in conspicuous changes at promoters that have high levels of RNAPII stalling and show specific enrichment for the *Drosophila* BAF remodeler catalytic subunit BRM, but not other remodeler families[71]. Interestingly, BAF subunits SMARCB1 and SMARCE1 remain bound to promoters during mitosis in mESCs, suggesting that mitotic bookmarking by SMARCB1/E1 and TFs could initiate these dynamics in newly divided cells[75,76].

BAF binding and ATP-dependent nucleosomal DNA translocation activity break histone–DNA contacts to partially unwrap a nucleosome (Fig. 6b, step 3). We speculate that BAF rapidly unbinds even before nucleosome eviction, which may involve multiple rapid cycles of BAF binding, nucleosomal DNA translocation and unbinding events. Live-cell imaging of *Drosophila* BRM and yeast RSC remodelers showing 5-s average residence and turnover times and ATP-dependent dissociation implies that BAF binding is dynamic and suggests that dissociation is part of the remodeling mechanism[18,19]. In vitro single-molecule measurements using physiological ATP concentrations estimate that SWI/SNF remodelers translocate nucleosomal DNA at the average rate of 12 bp s$^{-1}$ (ref. 77). This translocation speed combined with a short chromatin residence time agrees with multi-turnover remodeling for nucleosome eviction, which may require a major part of the 147-bp nucleosomal DNA to be disrupted. In a simplified two-component system (BAF and nucleosomes) where ATP is not limiting, the kinetics of nucleosome eviction would therefore be an outcome of a dynamic competition between nucleosome re-wrapping and BAF re-binding. RNAPII pausing promotes BAF occupancy, but paused RNAPII is also distinctively dynamic and rapidly turns over in a seconds timescale primarily due to premature termination as shown by live-cell imaging as well as by genome-wide mapping and single-molecule footprinting experiments[36,78,79]. Taken together, the short residence times of BAF and paused RNAPII suggest a requirement for additional factors or mechanisms for productive nucleosome eviction and NDR formation, without which BAF and RNAPII functions are abortive (Fig. 6a). Consistent with our model, chemically induced proximity-mediated tethering of BAF to PRC2-repressed promoters was sufficient in evicting H3K27me3 and increased chromatin accessibility in an ATP-dependent manner[80]. Our data for NANOG and KLF4 upregulation resulting in increased nucleosome eviction suggest that some pluripotency TFs might capture DNA motifs transiently exposed by BAF remodeling to further destabilize and evict nucleosomes, forming an NDR (Fig. 6b, step 4)[2,63,81]. Indeed, we had previously shown that yeast Abf1 and Reb1 bind to partially unwrapped nucleosomes that are targets of RSC remodeling[5], as expected if increased TF concentration, or TF DNA-binding affinity, or cooperative binding of multiple TFs, drives the nucleosome-depleted state[63]. TF binding may therefore provide an energetic advantage by reducing the ATP cost associated with abortive BAF remodeling. The lack of pluripotency TF binding in PRC2-repressed chromatin explains why transient RNAPII and BAF activity is insufficient for productive nucleosome eviction.

ATP-dependent nucleosome remodeling by BAF is crucial for tissue-specific transcriptional regulation across various developmental processes[82,83]. Deregulation of BAF remodeling is implicated in >20% of all human cancers and several neurodevelopmental disorders[26,27,84].

How BAF is targeted for precise spatiotemporal gene regulation has remained an open question[76,85]. Although BAF has been shown to interact with a few TFs, it is difficult to envision TF-mediated recruitment given the short chromatin residence times of these factors[14,17–19]. Our dynamic cycle model adequately explains the key roles of BAF and TFs in regulating locus- and cell-type-specific chromatin structure and transcription without the requirement for recruitment per se. Moreover, our study provides mechanistic insights into how changes in BAF dynamics and TF homeostasis in cancers may drive oncogenic gene expression programs.

## Online content

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

## Methods

### Cell culture

AB2.2 (ATCC SCRC-1023; male; strain: 129S5/SvEvBrd) mESCs were used in all experiments. Cells were thawed and initially cultured in 2i medium, consisting of Knockout DMEM (Gibco catalog (cat.) no. 10829018) supplemented with 15% ES-qualified FCS (Gibco cat. no. 16141079), 2 mM L-glutamine (Sigma-Aldrich cat. no. G7513), 0.1 mM MEM nonessential amino acids (Gibco cat. no. 11-140-050), 0.1 mM β-mercaptoethanol (Gibco cat. no. 21985023), 1,000 U ml$^{-1}$ leukemia inhibitory factor (MilliporeSigma cat. no. ESG1107), 3 μM GSK3β inhibitor CHIR99021 (Sigma-Aldrich cat. no. SML1046) and 1 μM MEK inhibitor PD0325901 (Sigma-Aldrich cat. no. PZ0162). SL medium contained all components except the GSK3β and MEK inhibitors. Cells were maintained on six-well plates or cell culture flasks coated with Attachment Factor Protein (Gibco cat. no. S006100) at 37 °C with 5% CO$_2$, and passaged every 48–72 h with daily medium changes. Cells were cultured for at least two passages before transferring to SL medium and cultured for at least two more passages before experiments. Cells were released with Accutase (STEMCELL Technologies cat. no. 07922) for collection, washed with sterile PBS, resuspended in medium supplemented with 10% dimethylsulfoxide (Sigma-Aldrich cat. no. 41640) and slow-frozen at −80 °C in isopropanol freezing chambers. All experiments were performed with cells collected and frozen after six to eight total passages. Cultures were periodically tested for *Mycoplasma* and karyotyped to detect any chromosomal abnormalities. For inhibitor treatments, medium was supplemented with 10 μM Triptolide (Selleckchem cat. no. S3604), 1 μM Flavopiridol hydrochloride hydrate (Sigma-Aldrich cat. no. FL3055), 5 μg ml$^{-1}$ Actinomycin D (Sigma-Aldrich cat. no. A9415) or 1:1,000 v/v dimethylsulfoxide; plates/flasks were transferred to ice at time points, and cells were washed twice with ice-cold PBS and promptly collected.

### CUT&Tag

CUT&Tag involves targeting chromatin-associated proteins in permeabilized cells using antibodies and utilizing a protein A–Tn5 transposo-some fusion to insert DNA adapters into targeted genomic regions for mapping protein binding genome-wide with high specificity and resolution[29]. CUT&Tag was performed using frozen cells as described previously[29], with some modifications. Also see https://www.protocols.io/view/bench-top-cut-amp-tag-kqdg34qdpl25/v3 for a step-by-step protocol. For each CUT&Tag experiment, $0.2 \times 10^6$ cells were bound to 10 μl of Bio-Mag Plus Concanavalin A-coated magnetic beads (Bangs Laboratories cat. no. BP531), equilibrated with binding buffer (20 mM K-HEPES pH 7.9, 10 mM KCl, and 1 mM each CaCl$_2$ and MnCl$_2$). Beads (with bound cells) were magnetized and supernatant removed, then washed once with 400 μl and resuspended in 200 μl of Wash buffer (20 mM Na-HEPES pH 7.5, 150 mM NaCl, 0.5 mM spermidine and EDTA-free protease inhibitor) supplemented with 2 mM EDTA and 0.05% Digitonin (MilliporeSigma cat. no. 3004105GM). Primary antibodies were mixed at optimum dilutions (see below) and incubated overnight at 4 °C on a rotating platform. Beads were washed once with 400 μl of Dig-Wash (Wash buffer supplemented with 0.05% Digitonin), resuspended in 200 μl of Dig-Wash with a secondary antibody (see below) and incubated for 30 min to 1 h at room temperature on a rotating platform. Beads were washed twice with 400 μl of Dig-Wash and resuspended in 200 μl of Dig-Med buffer (Dig-Wash buffer, except containing 300 mM NaCl) with 1:200 dilution (~0.04 μM) of lab-made protein A–Tn5 transposase fusion protein (pA–Tn5) pre-loaded with double-stranded adapters with 19-mer mosaic ends and containing carry-over *Escherichia coli* DNA, useful for spike-in calibration[29]. pA–Tn5 incubations were performed on a rotating platform for 1 h at room temperature. Beads were washed three times with 400 μl of Dig-Med to remove unbound pA–Tn5 and resuspended in 300 μl of Tagmentation buffer (Dig-Med supplemented with 10 mM MgCl$_2$). Tagmentation reactions were performed by incubating samples at 37 °C on a rotating platform for 1 h. Tagmentation reactions were stopped with 10 μl of 0.5 mM EDTA, 3.1 μl of 10% SDS (1% final) and 2 μl of 20 mg ml$^{-1}$ Proteinase K (Invitrogen cat. no. 25530049) and incubated in a 50 °C water bath for 1 h or at 37 °C overnight with rotation. DNA was extracted using the phenol–chloroform extraction method and precipitated using chilled 75% ethanol. DNA pellets were dissolved in 30 μl of 0.1 × TE (1 mM Tris pH 8, 0.1 mM EDTA) supplemented with a 1:400 dilution of 10 mg ml$^{-1}$ RNase A (Thermo Scientific cat. no. EN0531) and incubated in a 37 °C water bath for 15 min. Libraries were amplified by addition of 2 μl each of barcoded 10 mM i5 and i7 primer solutions (Supplementary Data 1) and NEBNext HiFi 2 × PCR Master mix (NEB cat. no. M0541) with 13 rounds of amplification as described previously[29]. Sequencing libraries were purified using a 1.3× ratio of HighPrep PCR Cleanup beads (MagBio genomics cat. no. AC-60500) as per manufacturer's instructions and eluted in 0.1 × TE. Library quality and concentration were evaluated using Agilent TapeStation D1000 capillary gel analysis.

### RNAPII-S5P CUTAC

CUTAC using RNAPII-S5P for accessible site mapping was performed as described in the step-by-step protocol: https://www.protocols.io/view/cut-amp-tag-direct-with-cutac-x54v9mkmzg3e/v3?step=1 (ref. 21). Briefly, nuclei were prepared as previously described[29] and lightly crosslinked (0.1% formaldehyde 2 min), then washed and resuspended in Wash buffer (20 mM HEPES pH 7.5, 150 mM NaCl, 2 mM spermidine and Roche complete EDTA-free protease inhibitor). CUTAC was performed with $0.05 \times 10^6$ nuclei by mixing with 5 μl of Concanavalin A magnetic beads. Primary antibody against RNAPII-S5P (Cell Signaling Technology cat. no. 13523) was added at 1:50 dilution in Wash buffer supplemented with 0.1% BSA and incubated overnight at 4 °C. Beads were magnetized and supernatant was removed, and beads were resuspended in Wash buffer containing 1:100 guinea pig anti-rabbit secondary antibody (Antibodies Online cat. no. ABIN101961) and incubated for 0.5–1 h at room temperature. Beads were magnetized and washed (on the magnet) once with Wash buffer, resuspended in pAG-Tn5 pre-loaded with mosaic-end adapters (EpiCypher cat. no. 15-1117, 1:20 dilution) in 300-Wash buffer (Wash buffer except containing 300 mM NaCl) and incubated for 1 h at room temperature. Beads were washed (on the magnet) three times in 300-Wash, then incubated at 37 °C for 20 min in 50 μl of CUTAC-hex tagmentation solution (5 mM MgCl$_2$, 10 mM TAPS, 10% 1,6-hexanediol). Bead suspensions were chilled on ice and magnetized, supernatant was removed and beads were washed with 10 mM TAPS pH 8.5, 0.2 mM EDTA and resuspended in 5 μl of 0.1% SDS, 10 mM TAPS pH 8.5. Beads were incubated at 58 °C in a thermocycler with heated lid for 1 h, followed by addition of 15 μl of 0.67% Triton X-100 to neutralize the SDS. Libraries were amplified by addition of 2 μl each of barcoded 10 mM i5 and i7 primer solutions (Supplementary Data 1) and NEBNext HiFi 2× PCR Master mix (NEB cat. no. M0541) with gap-filling and 12-cycle PCR: 58 °C 5 min, 72 °C 5 min, 98 °C 30 s, 12 cycles of (98 °C 10-s denaturation and 60 °C 10-s annealing/extension), 72 °C 1 min and 8 °C hold. Sequencing libraries were purified with 1.3× ratio of HighPrep PCR Cleanup beads as per the manufacturer's instructions and eluted in 0.1 × TE. Library quality and concentration were evaluated using Agilent TapeStation D1000 capillary gel analysis.

### CUT&RUN and CUT&RUN.ChIP

CUT&RUN.ChIP used pA–MN-digested native chromatin released by CUT&RUN targeting BRG1 as input for subsequent ChIP of histone epitopes. BRG1 CUT&RUN.ChIP was performed as described previously[5,86] with some modifications. For CUT&RUN, $2.5 \times 10^6$ cells were bound to 50 μl of Concanavalin A-coated magnetic beads. Primary and secondary antibody incubation and washes were performed as described above for CUT&Tag, but in 1 ml of the buffers using 1.5-ml low-binding flip-cap tubes. Incubations were done at 4 °C, and ice-cold buffers were used in every step. After secondary antibody incubation and washes, bead-bound cells were resuspended in ice-cold Dig-Wash

with lab-made pA–MN fusion protein (360 µg ml⁻¹, 1:400 dilution) and incubated for 1 h at 4 °C with rotation. The beads were washed three times in ice-cold Dig-Wash, resuspended in 0.5 ml of ice-cold Dig-Wash and equilibrated to 0 °C. CaCl$_2$ was quickly mixed to a final concentration of 2 mM and the reactions incubated on ice for 5 min for MNase digestion, and reactions were stopped with 0.5 ml of 2 × STOP buffer (150 mM NaCl, 20 mM EDTA, 4 mM EGTA and 50 µg ml⁻¹ RNase A) supplemented with BRG1 peptide (Abcam cat. no. ab241115) to a final concentration of 10 µg ml⁻¹. Samples were incubated at 37 °C for 20 min and centrifuged for 5 min at 16,000$g$ and 4 °C. The supernatant was removed on a magnet stand and divided into five 200-µl aliquots for ChIP. One aliquot was saved (at 4 °C) as the input. To the ChIP samples, respective antibodies (IgG and histone PTMs, see below) were added and incubated at 4 °C overnight. Protein A Dynabeads (Invitrogen cat. no. 10002D) were equilibrated in Wash buffer supplemented with 0.05% Tween-20, and 20 µl of beads were added to each ChIP sample (except the input). Samples were incubated at 4 °C for 30 min and washed once with Wash buffer + Tween-20. The ChIP samples were brought up with DNA-extraction buffer (150 mM NaCl, 10 mM EDTA, 2 mM EGTA, 0.1% SDS and 0.2 mg ml⁻¹ Proteinase K). SDS (0.1%) and Proteinase K (0.2 mg ml⁻¹) were added separately to the input samples. Samples were incubated at 50 °C for 1 h. DNA was extracted using the phenol–chloroform extraction method and 40 µg of glycogen (Roche cat. no. 10901393001) was mixed with the aqueous phase. DNA was precipitated using chilled 75% ethanol and dissolved in 0.1 × TE.

For each TF CUT&RUN, 0.5 × 10⁶ cells were bound to 10 µl of Concanavalin A-coated magnetic beads. Incubations and washes were done as for BRG1 CUT&RUN, except that primary and secondary antibody incubations and pA–MN binding were performed in 200-µl volumes. The MNase digestion reaction was done in 150 µl with incubation on ice for 30 min and the reaction was stopped using 150 µl of 2 × STOP buffer without any peptide, but supplemented with 10 pg µl⁻¹ *S. cerevisiae* MNase-digested nucleosomal-length spike-in DNA. Samples were incubated at 37 °C for 20 min and centrifuged for 5 min at 16,000$g$ and 4 °C. The supernatant containing released chromatin particles was removed on a magnet stand and SDS (0.1%) and Proteinase K (0.2 mg ml⁻¹) were added. Samples were incubated at 50 °C for 1 h and used directly for DNA extraction using the phenol–chloroform extraction method described above.

Libraries were prepared for Illumina sequencing with UDI (unique dual indexes) adapters (Supplementary Data 1), without size-selection, and following the KAPA DNA polymerase library preparation kit protocol (https://www.kapabiosystems.com/product-applications/ products/next-generation-sequencing-2/dna-library-preparation/ kapa-hyper-prep-kits/), optimized to favor exponential amplification of <1,000-bp fragments over linear amplification of large DNA fragments as described previously[5,87]: 98 °C 45 s, 12 cycles of (98 °C 15-s denaturation and 60 °C 10-s annealing/extension), 72 °C 1 min and 8 °C hold. Sequencing libraries were then purified using a 1.3× ratio of HighPrep PCR Cleanup System. Library concentrations were quantified using the D1000 TapeStation system (Agilent).

### Sequencing, data processing, data analysis and data visualization
Libraries were sequenced for 25 cycles in 25-bp paired-end mode on the Illumina HiSeq 2500 or in 50-bp paired-end on the NextSeq 2000 at the Fred Hutchinson Cancer Center Genomics Shared Resource, and data were analyzed as described (https://www.protocols.io/view/ cut-amp-tag-data-processing-and-analysis-tutorial-e6nvw93x7gmk/ v1). Briefly, adapters were clipped and paired-end *Mus musculus* reads were mapped to UCSC mm10 using Bowtie2 (ref. 88) with parameters: --very-sensitive-local --soft-clipped-unmapped-tlen --dovetail --no-mixed --no-discordant -q --phred33 -I10 -X 1000 (for CUT&Tag) or --end-to-end --very-sensitive --no-mixed --no-discordant -q --phred33 -I 10 -X 700 (for CUT&RUN.ChIP). Spike-in *E. coli* reads in CUT&Tag

experiments were mapped to Ensembl masked R64-1-1 with parameters: --end-to-end --very-sensitive --no-overlap --no-dovetail --no-unal --no-mixed --no-discordant -q --phred33 -I10 -X 700. Continuous-valued data tracks (bedGraph and bigWig) were generated using genomecov in bedtools v.2.30.0 (-bg option) and normalized as fraction of total counts (for CUTAC and CUT&RUN.ChIP) or calibrated using total number of spike-in reads (for CUT&Tag)[29]. Genomic tracks were displayed using Integrated Genome Browser. RNAPII-S5P CUTAC H3K9me3 CUT&Tag and TF CUT&RUN peaks were called by SEACR (v.1.3) using the norm and relaxed settings[89]; 20–120-bp fragments were used to call TF peaks. Profile plots, heatmaps and correlation matrices were generated using deepTools v.3.5.1 (ref. 90). Scores were averaged over 50-bp nonoverlapping bins with respect to reference points and plotted as the mean. Violin plots were generated with GraphPad Prism 9. Scores were computed using deepTools v.3.5.1, and extreme outliers were identified using the ROUT method ($Q = 0.2\%$) and removed.

### Immunofluorescence
Cells were cultured as described above and immunofluorescence staining was conducted in-well using 12-well plates at room temperature. After culturing, cells were washed once with 1 ml of PBS with gentle rocking for 5 min, then incubated with 4% paraformaldehyde in 1 ml of PBS for 15 min with gentle rocking. Wells were rinsed once with PBS, then washed twice with 1 ml of PBS supplemented with 0.1% Triton X-100 (PBST) for 5 min each with gentle rocking. Wells were then incubated with 0.5 ml of PBST containing primary antibody in optimum dilution (see below) and 1% BSA overnight at 4 °C with gentle rocking. Wells were rinsed once, then washed twice for 5 min each with 1 ml of PBST. Wells were then incubated with 0.5 ml of PBST containing fluorophore-conjugated secondary antibody (see below) for 1 h at room temperature with gentle rocking. Wells were rinsed once and washed twice for 5 min each with 1 ml of PBST, then incubated with 0.5 ml of PBST with 1:50,000 DAPI for 20 min at room temperature with gentle rocking for nucleic acid staining. Wells were rinsed once and washed three times for 5 min each with 1 ml of PBST, then imaged in 0.5 ml of PBS on an EVOS FL Auto 2 Cell Imaging System (Invitrogen) with ×10 magnification.

### Western blotting
To make whole-cell protein extracts, 5 × 10⁶ mESCs were collected, washed once with PBS and resuspended in 200 µl of standard protein sample buffer. Samples were vortexed, boiled for 5 min, then cooled to room temperature. Benzonase (1 µl) was added, and samples were incubated at room temperature for 5 min before freezing for further use. Samples were spun down and supernatants were used to determine total protein concentration using Pierce BCA Protein Assay Kit (Thermo Scientific cat. no. PI23227). Equal amounts of proteins were run on each well of 4–20% Tris-Glycine polyacrylamide gels then transferred to nitrocellulose membrane, and 1:1,000 dilutions of primary and secondary antibodies were used for blotting. Secondary goat anti-mouse IRDye 680RD and goat anti-rabbit or donkey anti-goat IRDye 800CW (LI-COR Biosystems) were used against the anti-histone H3 primary antibody (mouse) and TF primary antibodies (rabbit/goat), respectively. Images were acquired using Li-Cor Odyssey DLx Imaging System (LI-COR Biosystems). Quantifications were performed using the ImageJ software (v.1.53t 24), accounting for local background.

### Antibodies
RNAPII-S5P: rabbit monoclonal (D9N5I, Cell Signaling Technology cat. no. 13523), 1:50; RNAPII-S2P: rabbit monoclonal (E1Z3G, Cell Signaling Technology cat. no. 13499), 1:100; RPB3: rabbit polyclonal (Bethyl Laboratories cat. no. A303-771A, lot no. A303-771A2), 1:100; BRG1: rabbit monoclonal (EPNCIR111A, Abcam cat. no. ab110641), 1:100, for CUT&Tag and CUT&RUN.ChIP, and rabbit polyclonal (Invitrogen cat. no. 720129, lot. no. 2068859), 1:250, for immunofluorescence; H3K4me1:

rabbit polyclonal (Abcam cat. no. ab8895, lot no. GR3283237), 1:100; H3K4me3: rabbit polyclonal (Active Motif cat. no. 39915, lot. no. 24118008), 1:100, for CUT&Tag, and rabbit monoclonal (EpiCypher cat. no.13-0028), 1:100, for CUT&RUN.ChIP; H3K27me3: rabbit monoclonal (C36B11, Cell Signaling Technology cat. no. 9733), 1:100; H3K9me3: rabbit monoclonal (EPR16601, Abcam cat. no. ab176916), 1:100; guinea pig anti-rabbit secondary: Antibodies Online cat. no. ABIN101961, 1:100; isotype control (IgG) for CUT&RUN.ChIP: rabbit monoclonal (EPR25A, Abcam cat. no. ab172730), 1:100; NANOG: rabbit polyclonal (Bethyl Laboratories cat. no. A300-397A, lot no. 3), 1:100; KLF4: goat polyclonal (R&D Systems cat. no. AF3158, lot. no. WRR0719011), 1:100 for CUT&RUN, 1:50 for immunofluorescence; OCT4: rabbit monoclonal (EPR17929, Abcam cat. no. ab181557), 1:100; SOX2: rabbit monoclonal (EPR3131, Abcam cat. no. ab92494), 1:100; CTCF: rabbit monoclonal (EPR7314(B), Abcam cat. no. ab128873), 1:100; rabbit anti-goat secondary: Abcam cat. no. ab6697, 1:100; goat anti-rabbit-Cy5 secondary: Jackson ImmunoResearch cat. no. 111-175-144, 1:500; donkey anti-goat-rhodamine red secondary: Jackson ImmunoResearch cat. no. 705-295-147, 1:250; mouse anti-H3 for western blot: (mAbcam 24834, Abcam cat. no. ab24834), 1:500; IRDye 800CW goat anti-rabbit: LI-COR cat. no. 926-32211, 1:10,000; IRDye 800CW donkey anti-goat: LI-COR cat. no. 926-32214, 1:10,000; IRDye 680RD goat anti-mouse: LI-COR cat. no. 926-68070, 1:10,000.

## Statistics & reproducibility

Overall data quality was evaluated by peak-calling using SEACR (v.1.3)[89] with default false discovery rate and 'relaxed' parameter and FRiP (Fraction of Reads in Peaks) analysis, which is very sensitive to reproducibility of replicates. For every experiment, at least two biological replicates were performed. No statistical method was used to predetermine sample size nor were data excluded from the analyses. The experiments were not randomized and investigators were not blinded to allocation during experiments and outcome assessment.

## Reporting summary

Further information on research design is available in the Nature Portfolio Reporting Summary linked to this article.

## Data availability

All primary sequencing data have been deposited as paired-end fastq files and all mapped data have been deposited as bigWig files in the Gene Expression Omnibus under the accession number GSE224292. Public datasets used: ATAC-seq: GSM2267967 (ref. 91); START RNA-seq: GSM1551910 (ref. 31); MNase seq: GSE117767 (ref. 49); mESC enhancer annotation: Whyte et al.[37]. Source data are provided with this paper.

## Code availability

No custom codes were used.

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

## Acknowledgements

We thank K. Ahmad and T. Tsukiyama for critical readings of the manuscript, J. Henikoff for help with processing of sequencing data, S. Showman for help with western blotting experiments, and D. Scalzo and X. Wang (Northwestern University) for guidance on mESC culturing. This research was supported by NIH grant no. K99 GM138920 (S.B.), the Howard Hughes Medical Institute (S.H.) and NIH grant no. P30CA015704 (Fred Hutch Shared Resources).

## Author contributions

S.B. conceptualized the project and performed the investigations. S.B. wrote the original draft of the manuscript. S.B. and S.H. reviewed and edited the manuscript. S.H. and S.B. were responsible for funding acquisition. S.H. was responsible for resources. Both authors approved the final manuscript.

## Competing interests

The authors declare no competing interests.

## Additional information

**Extended data** is available for this paper at https://doi.org/10.1038/s41588-023-01603-8.

**Correspondence and requests for materials** should be addressed to Sandipan Brahma or Steven Henikoff.

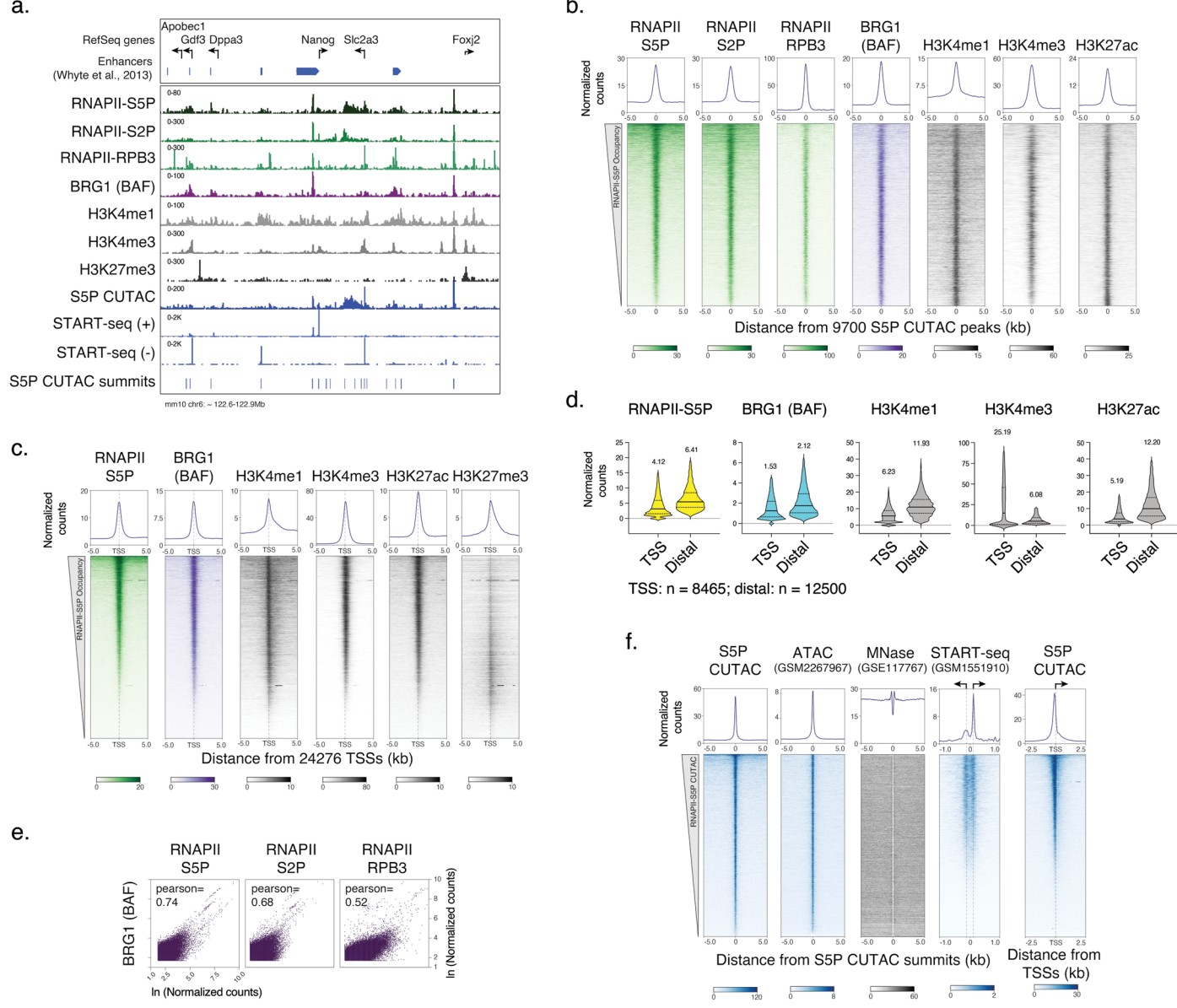

**Extended Data Fig. 1 | CUT&Tag of chromatin epitopes and RNAPII-S5P CUTAC in mESCs. a**, Representative genomic tracks showing RNAPII, BRG1, histone PTM occupancy by CUT&Tag, chromatin accessibility (RNAPII-S5P CUTAC), and transcriptional activity (START-seq) at the *Nanog* promoter and enhancer cluster and flanking genes. Previously annotated enhancer regions[41] are shown on top. **b, c**, Heatmaps (bottom) and average plots (top) comparing RNAPII, BRG1, and histone PTM occupancy by CUT&Tag, relative to the primary peaks (summits) of RNAPII-S5P CUTAC (**b**) and RefSeq annotated gene TSSs (**c**), sorted by decreasing RNAPII-S5P occupancy. **d**, Violin plots of CUT&Tag signal distribution comparing RNAPII-S5P, BRG1, and histone PTM occupancies at specific set of gene promoters (TSS) showing RNAPII-S5P enrichment versus promoter-distal

S5P CUTAC and pluripotency TF-binding peaks (Distal). Median value (solid line), upper and lower quartiles (broken lines) and outliers were calculated using the Tukey method. Numbers on top show mean values. **e**, Scatterplots comparing BRG1 and RNAPII S5P, S2P, and RPB3 CUT&Tag reads in 1,000 bp genome-wide consecutive non-overlapping bins. **f**, Heatmaps (bottom) and average plots (top) comparing chromatin accessibility (RNAPII-S5P CUTAC and ATAC-seq), nucleosome positions (MNase seq), and transcriptional activity (START-seq), relative to the primary peaks (summits) of RNAPII-S5P CUTAC; and RNAPII-S5P CUTAC signal relative to RefSeq annotated gene TSSs (extreme right). All datasets are representative of at least two biological replicates.

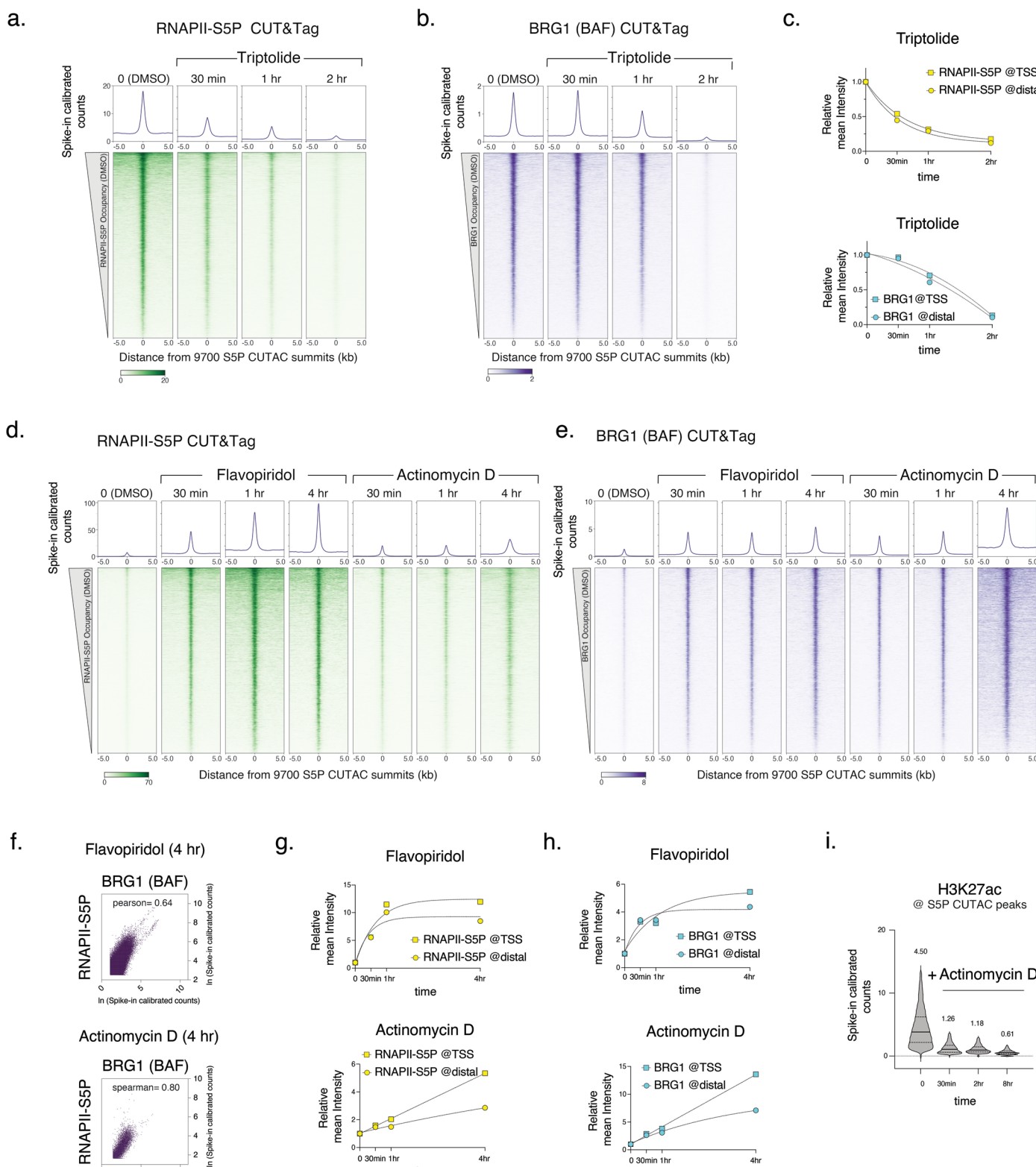

**Extended Data Fig. 2 | CUT&Tag of RNAPII-S5P and BRG1 after inhibitor treatment. a, b, d, e,** Heatmaps (bottom) and average plots (top) comparing RNAPII-S5P (**a, d**) and BRG1 (**b, e**) occupancy by spike-in calibrated CUT&Tag relative to the primary peaks (summits) of RNAPII-S5P CUTAC in untreated cells (DMSO) versus cells treated with Triptolide (**a, b**), Flavopiridol, and Actinomycin D (**d, e**) at indicated time points post drug treatment. **c, g, h,** Comparison of fold changes in mean RNAPII-S5P and BRG1 occupancy (spike-in calibrated CUT&Tag) at gene promoters (TSS, squares) and promoter-distal regulatory regions (Distal, circles) at time points after drug treatments. **f,** Scatterplots comparing BRG1 and RNAPII-S5P CUT&Tag reads in 1000 bp genome-wide consecutive non-overlapping bins in cells treated with Flavopiridol and Actinomycin D. **i,** Violin plots of CUT&Tag signal distribution comparing histone PTM H3K27ac occupancy at S5P CUTAC peaks over time points after Actinomycin D treatment. Median value (solid line), upper and lower quartiles (broken lines) and outliers were calculated using the Tukey method. Numbers on top show mean values. All datasets are representative of at least two biological replicates.

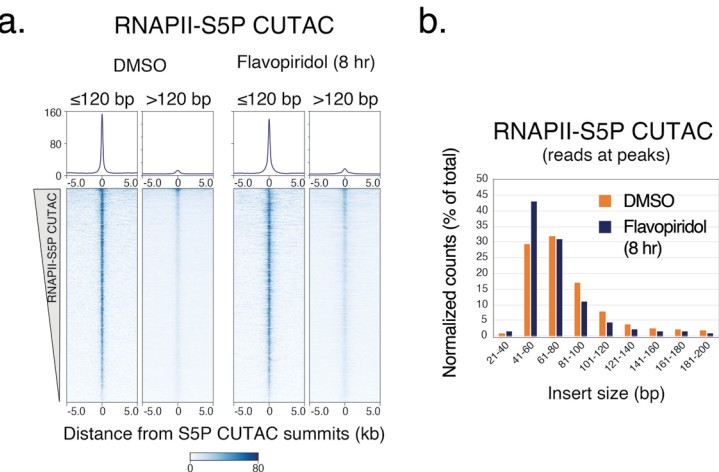

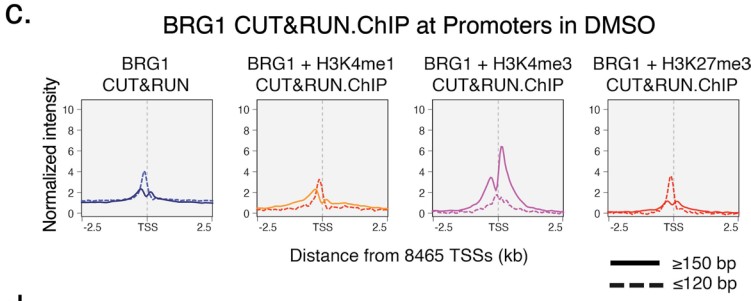

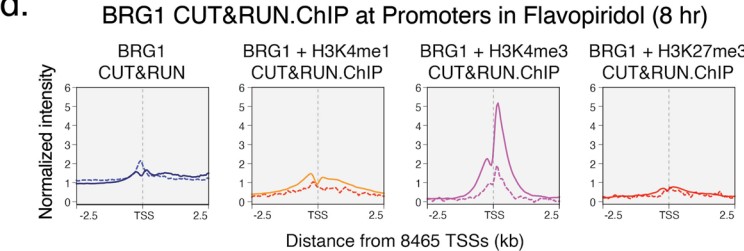

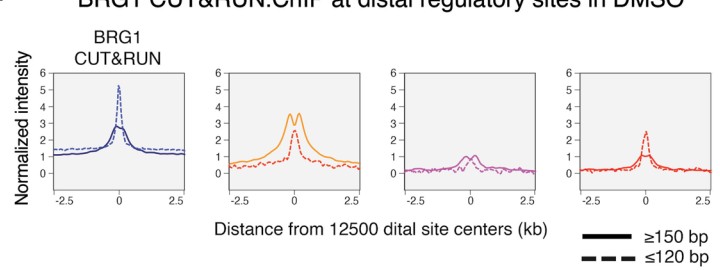

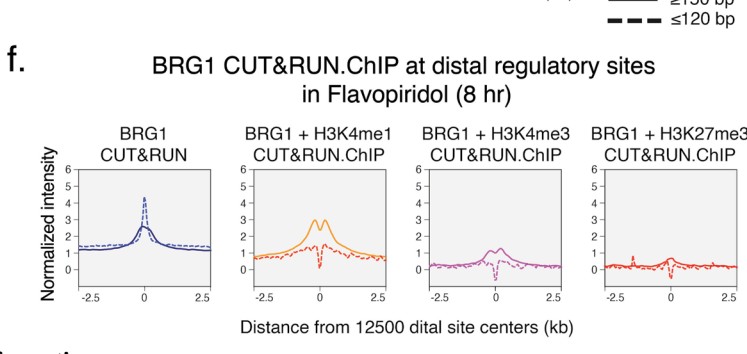

**Extended Data Fig. 3 | See next page for caption.**

**Extended Data Fig. 3 | CUT&RUN.ChIP of BRG1. a**, Heatmaps (bottom) and average plots (top) of RNAPII-S5P CUTAC separated by fragment size, relative to primary peaks (summits) of RNAPII-S5P CUTAC. **b**, Comparison of RNAPII-S5P CUTAC fragment size distribution over peaks (promoter and enhancer NDR spaces) in cells treated with DMSO (control) and Flavopiridol; same data as used for Fig. 2A, plotted differently. **c-f**, Enrichment of nucleosomal (≥150 bp, solid lines) and subnucleosomal (≤120 bp, broken lines) reads from BRG1 CUT&RUN and CUT&RUN.ChIP experiments, relative to gene promoter TSSs showing RNAPII-S5P enrichment (**c, d**) and distal regulatory sites (**e, f**), in DMSO (**c, e**) and Flavopiridol (**d, f**) treated cells. CUT&RUN.ChIP data were plotted as enrichment in histone ChIP over IgG isotype control. All datasets are representative of at least two biological replicates.

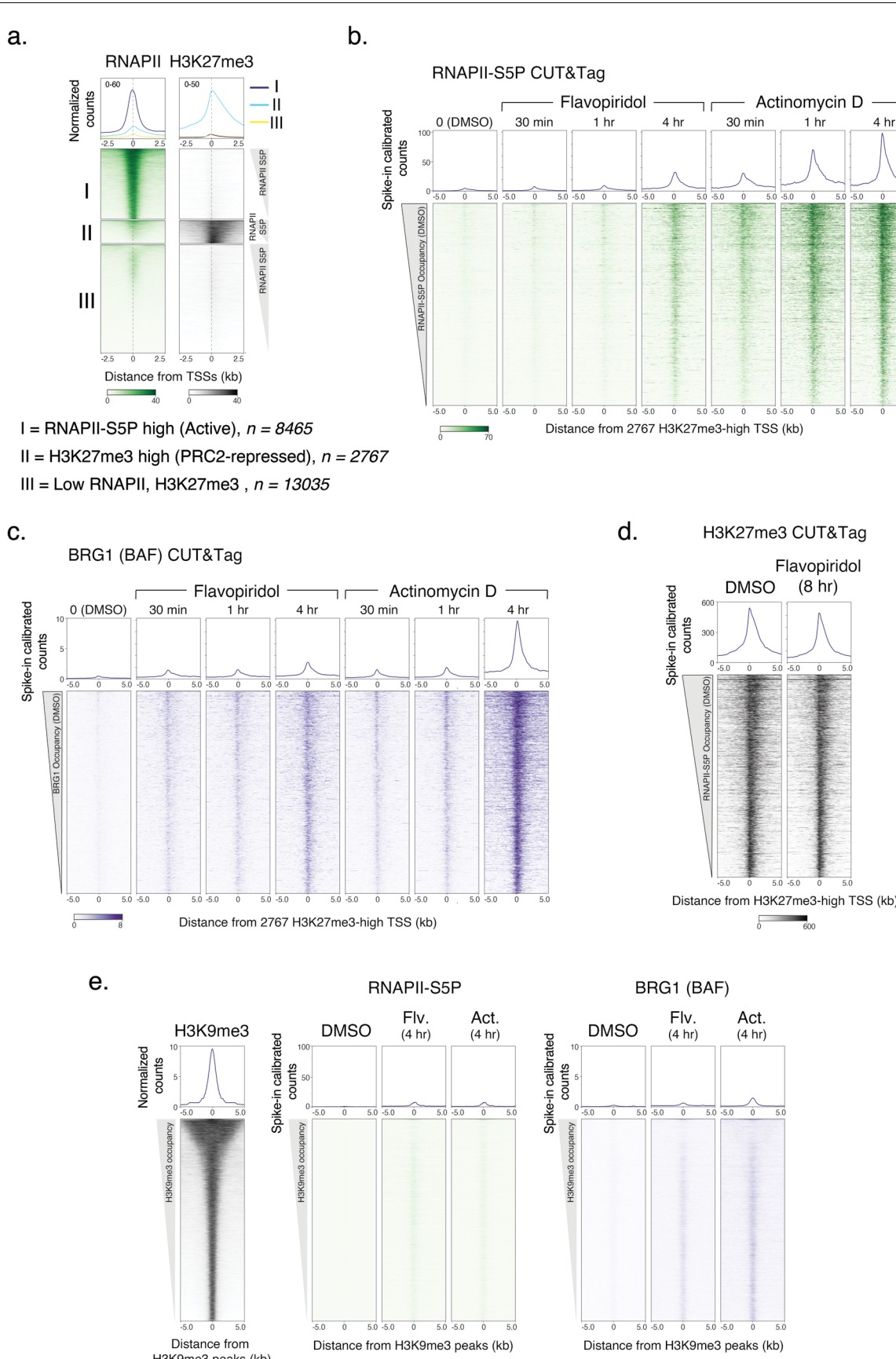

**Extended Data Fig. 4 | See next page for caption.**

**Extended Data Fig. 4 | CUT&Tag of RNAPIIS5P and BRG1 at PcG-repressed promoters. a**, K-means clustering of RNAPII-S5P and H3K27me3 CUT&Tag reads relative to RefSeq annotated gene promoter TSSs to group promoters as active (I, RNAPII-S5P enriched) and PcG-repressed (II, H3K27me3 enriched), and not enriched for either (III). **b, c**, Heatmaps (bottom) and average plots (top) comparing RNAPII-S5P (**b**) and BRG1 (**c**) occupancy by spike-in calibrated CUT&Tag relative to PRC2-repressed promoter TSSs in untreated cells (DMSO) versus cells treated with Flavopiridol or Actinomycin D at indicated time points post drug treatment. **d**, Heatmaps (bottom) and average plots (top) comparing H3K27me3 histone PTM occupancy by spike-in calibrated CUT&Tag relative to PRC2-repressed promoter TSSs in untreated cells (DMSO) and cells treated with Flavopiridol. **e**, Heatmaps (bottom) and average plots (top) comparing H3K9me3 histone PTM occupancy (CUT&Tag) with RNAPII-S5P and BRG1 relative to H3K9me3 peaks in untreated cells (DMSO) versus cells treated with Flavopiridol or Actinomycin D. RNAPII-S5P and BRG1 CUT&Tag reads were spike-in calibrated and plotted to the same scales as in panels b and c, respectively. All datasets are representative of at least two biological replicates.

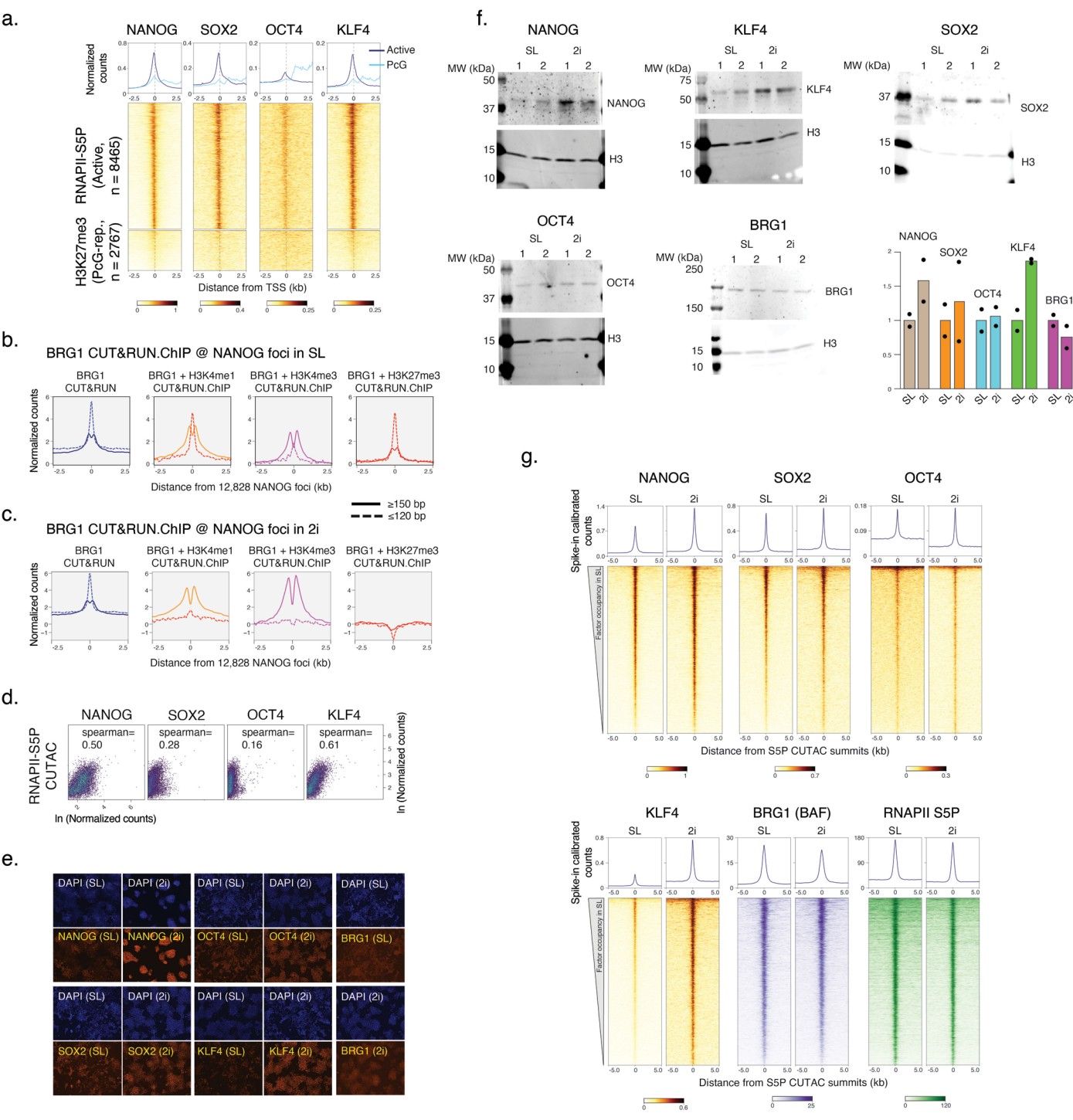

**Extended Data Fig. 5 | CUT&RUN of pluripotency TFs in SL versus 2i culture conditions. a**, Heatmaps (bottom) and average plots (top) comparing pluripotency TF occupancy by CUT&RUN at RNAPII-enriched (active) and H3K27me3-enriched (PRC2-repressed) promoters. Promoters were grouped based on K-means clustering of RNAPII-S5P and H3K27me3 CUT&Tag reads mapping to a 5 kb window around the TSSs of RefSeq-annotated mESC genes, see Extended Data Fig. 4a. **b, c**, Enrichment of nucleosomal (≥150 bp, solid lines) and subnucleosomal (≤120 bp, broken lines) reads from BRG1 CUT&RUN and CUT&RUN.ChIP experiments, relative NANOG foci (smallest fragment within primary peaks called in SL condition), in SL (**b**) and 2i (**c**) mESC culture conditions. CUT&RUN.ChIP data were plotted as enrichment in histone ChIP over IgG isotype control. **d**, Scatterplots comparing pluripotency TF CUT&RUN and RNAPII-S5P CUT&Tag reads over S5P CUTAC peaks in SL mESCs. **e**, Immunofluorescent staining comparing pluripotency TF and BRG1 expression

in SL versus 2i culture conditions. Cy5-conjugated secondary antibodies were used in all experiment except for KLF4, where Rhodamine red-conjugated antibody was used. DAPI (blue) was used to stain the nucleus in cells. **f**, Western blot analysis comparing pluripotency TF and BRG1 expression in SL and 2i culture conditions. Equal amounts of extracted total proteins were loaded in each well of 4–20% gradient polyacrylamide SDS electrophoresis gel, and histone H3 signal is used as control to ensure equivalent protein loading. Bar-graph quantifications represent average of two biological replicates with individual data points shown as black dots. Data were normalized to values in SL. **g**, Heatmaps (bottom) and average plots (top) comparing pluripotency TF occupancy by spike-in calibrated CUT&RUN in SL versus 2i culture conditions. Heatmaps were plotted relative to S5P CUTAC summits showing TF binding at sites of DNA accessibility and sorted by decreasing TF occupancy in SL (CUT&RUN signal). All datasets are representative of at least two biological replicates.

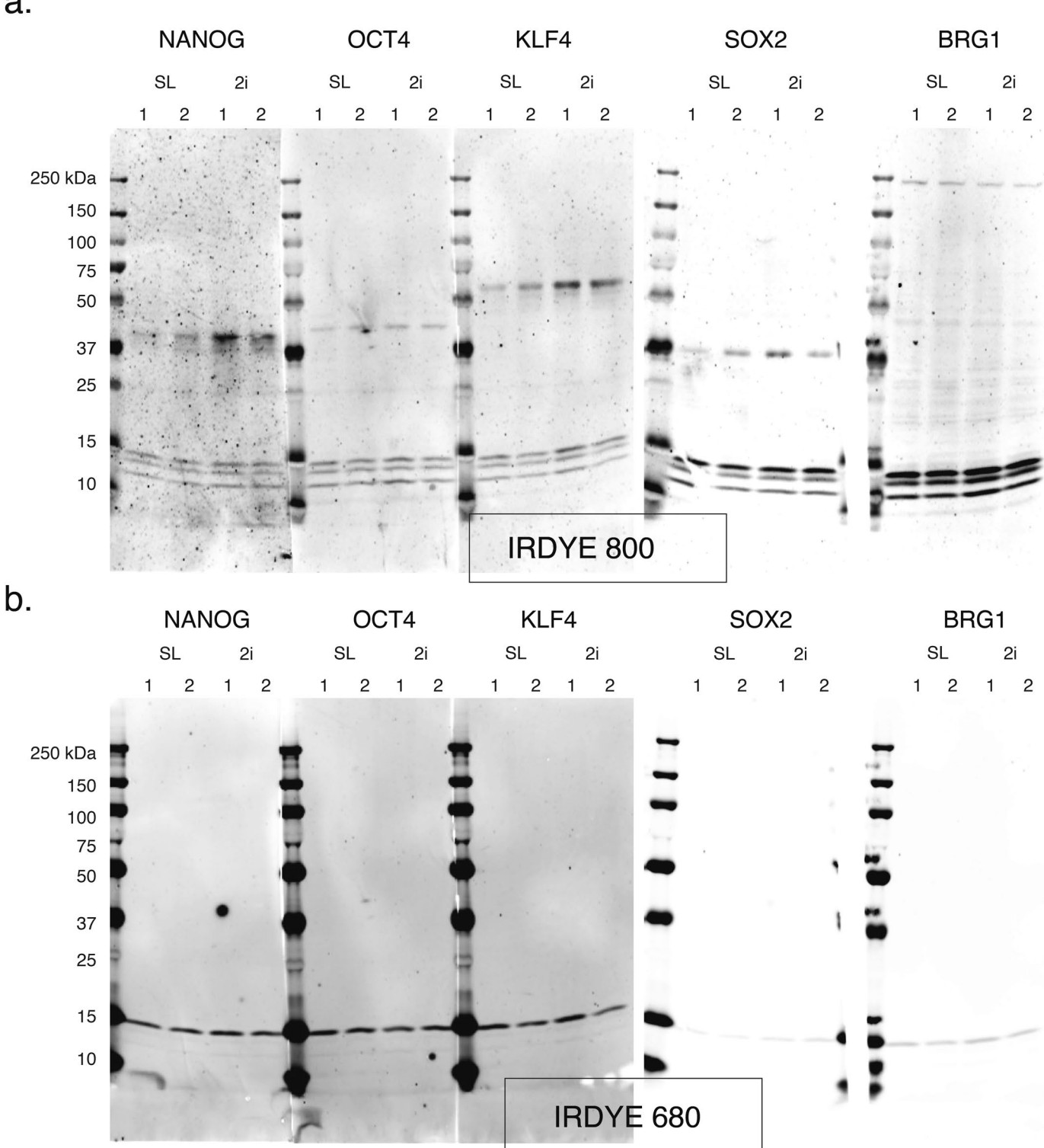

**Extended Data Fig. 6 | Labelled and uncropped Western blots of data shown in Extended Data Fig. 5f.** Western blots were dually stained with IRDye 800CW secondary antibody against primary antibodies targeting the TFs (panel a) and IRDye 680RD secondary antibody against primary antibodies targeting histone H3 (panel b) and imaged using respective filters on the Odyssey DLx Imaging System.

# Reporting Summary

## Statistics

For all statistical analyses, confirm that the following items are present in the figure legend, table legend, main text, or Methods section.

| n/a | Confirmed | |
|---|---|---|
| ☐ | ☒ | The exact sample size (*n*) for each experimental group/condition, given as a discrete number and unit of measurement |
| ☐ | ☒ | A statement on whether measurements were taken from distinct samples or whether the same sample was measured repeatedly |
| ☒ | ☐ | The statistical test(s) used AND whether they are one- or two-sided *Only common tests should be described solely by name; describe more complex techniques in the Methods section.* |
| ☒ | ☐ | A description of all covariates tested |
| ☒ | ☐ | A description of any assumptions or corrections, such as tests of normality and adjustment for multiple comparisons |
| ☐ | ☒ | A full description of the statistical parameters including central tendency (e.g. means) or other basic estimates (e.g. regression coefficient) AND variation (e.g. standard deviation) or associated estimates of uncertainty (e.g. confidence intervals) |
| ☒ | ☐ | For null hypothesis testing, the test statistic (e.g. *F*, *t*, *r*) with confidence intervals, effect sizes, degrees of freedom and *P* value noted *Give P values as exact values whenever suitable.* |
| ☒ | ☐ | For Bayesian analysis, information on the choice of priors and Markov chain Monte Carlo settings |
| ☒ | ☐ | For hierarchical and complex designs, identification of the appropriate level for tests and full reporting of outcomes |
| ☒ | ☐ | Estimates of effect sizes (e.g. Cohen's *d*, Pearson's *r*), indicating how they were calculated |

*Our web collection on statistics for biologists contains articles on many of the points above.*

## Software and code

Policy information about availability of computer code

| | |
|---|---|
| Data collection | The size distributions and molar concentration of libraries were determined using an Agilent 4200 TapeStation. Up to 96 barcoded libraries were pooled at approximately equimolar concentration for sequencing. Paired-end 25×25 bp sequencing on the Illumina HiSeq 2500 platform or PE 50x50 bp sequencing on the Illumina NextSeq 2000 was performed by the Fred Hutchinson Cancer Center Genomics Shared Resources. This yielded 3-6 million reads per antibody/sample. Paired-end reads were aligned using Bowtie2 version 2.3.4.3 to UCSC mm10 with options: --very-sensitive-local --soft-clipped-unmappedtlen --dovetail --no-mixed --no-discordant -q --phred33 -I 10 -X 1000 (for CUT&Tag and CUTAC) or --end-to-end --very-sensitive --no-mixed --no-discordant -q --phred33 -I 10 -X 700 (for CUT&RUN.ChIP). EVOS FL Auto 2 Cell Imaging System (Invitrogen) was used for immunofluroscence imaging. Western blotting images were acquired using Li-Cor Odyssey Dlx Imaging System (LI-COR Biosystems) |
| Data analysis | Bowtie 2; bedtools v2.30.0; Integrated Genome Browser v 8.5.4; SEACR v.1.3; deeptoolsv 3.5.1; GraphPad Prism 9; ImageJ version 1.53t 24; No custom codes were used. |

For manuscripts utilizing custom algorithms or software that are central to the research but not yet described in published literature, software must be made available to editors and reviewers. We strongly encourage code deposition in a community repository (e.g. GitHub). See the Nature Portfolio guidelines for submitting code & software for further information.

## Data

Policy information about availability of data

 All manuscripts must include a data availability statement. This statement should provide the following information, where applicable:

- Accession codes, unique identifiers, or web links for publicly available datasets
- A description of any restrictions on data availability
- For clinical datasets or third party data, please ensure that the statement adheres to our policy

All primary sequencing data have been deposited as paired-end fastq files and all mapped data have been deposited as bigWig files in the Gene Expression Omnibus under the accession number GSE 224292.
Public datasets used: ATAC-seq: GSM2267967; START RNA-seq: GSM1551910; MNase seq: GSE117767; mESC Enhancer annotation: Whyte et al., 2013

## Human research participants

Policy information about studies involving human research participants and Sex and Gender in Research.

| | |
|---|---|
| Reporting on sex and gender | n/a |
| Population characteristics | n/a |
| Recruitment | n/a |
| Ethics oversight | n/a |

Note that full information on the approval of the study protocol must also be provided in the manuscript.

# Field-specific reporting

Please select the one below that is the best fit for your research. If you are not sure, read the appropriate sections before making your selection.

☒ Life sciences ☐ Behavioural & social sciences ☐ Ecological, evolutionary & environmental sciences

For a reference copy of the document with all sections, see nature.com/documents/nr-reporting-summary-flat.pdf

# Life sciences study design

All studies must disclose on these points even when the disclosure is negative.

| | |
|---|---|
| Sample size | In this study we collected chromatin profiling data for comparative analysis from 2 to 3 replicate populations of cells for each condition, with cell numbers from 50,000 - 1,000,000, depending on specific method requirements. Each sample was sequenced to a depth of 3-6 million reads, sufficient for bulk characterization of chromatin in each sample. These sequencing depths and replicate numbers are standard in the field. |
| Data exclusions | Sequencing reads mapping to the mitochondrial genome were removed from all datasets. This was pre-established and is standard practice in the field. The purpose of this study was to perform comparative analysis of chromatin profiles from the nuclear genome and this can be confounded by variable read numbers from the mitochondrial genome. |
| Replication | At least 2 biological replicates were profiled. All attempts at replication were successful. |
| Randomization | n/a. These studies compare the same kinds of cells with and without experimental treatment under laboratory conditions, and data and analysis for this study are objective and not prone to influence by the researchers bias. |
| Blinding | n/a. The data and analysis for this study is objective and not prone to influence by researchers bias. |

# Reporting for specific materials, systems and methods

We require information from authors about some types of materials, experimental systems and methods used in many studies. Here, indicate whether each material, system or method listed is relevant to your study. If you are not sure if a list item applies to your research, read the appropriate section before selecting a response.

## Materials & experimental systems

| n/a | Involved in the study |
|---|---|
| ☐ | ☒ Antibodies |
| ☐ | ☒ Eukaryotic cell lines |
| ☒ | ☐ Palaeontology and archaeology |
| ☒ | ☐ Animals and other organisms |
| ☒ | ☐ Clinical data |
| ☒ | ☐ Dual use research of concern |

## Methods

| n/a | Involved in the study |
|---|---|
| ☐ | ☒ ChIP-seq |
| ☒ | ☐ Flow cytometry |
| ☒ | ☐ MRI-based neuroimaging |

## Antibodies

| | |
|---|---|
| Antibodies used | RNAPII-S5P: rabbit monoclonal (D9N5I, Cell Signaling Technology cat. no. 13523), 1:50; RNAPII-S2P: rabbit monoclonal (E1Z3G, Cell Signaling Technology cat. no. 13499), 1:100; RPB3: rabbit polyclonal (Bethyl Laboratories cat. no. A303-771A, lot no. A303-771A2), 1:100; BRG1: rabbit monoclonal (EPNCIR111A, abcam cat. no. ab110641), 1:100, for CUT&Tag and CUT&RUN.ChIP, and rabbit polyclonal (Invitrogen cat. no. 720129, lot. no. 2068859), 1:250, for immunofluorescence; H3K4me1: rabbit polyclonal (Abcam cat. no. ab8895, lot no. GR3283237), 1:100; H3K4me3: rabbit polyclonal (Active Motif cat. no. 39915, lot. no. 24118008), 1:100, for CUT&Tag, and rabbit monoclonal (EpiCypher cat. no. 13-0028), 1:100, for CUT&RUN.ChIP; H3K27me3: rabbit monoclonal (C36B11, Cell Signaling Technology cat. no. 9733), 1:100; H3K9me3: rabbit monoclonal (EPR16601, Abcam cat. no. ab176916), 1:100; guinea pig anti-rabbit secondary: Antibodies Online cat. no. ABIN101961, 1:100; isotype control (IgG) for CUT&RUN.ChIP: rabbit monoclonal (EPR25A, Abcam cat. no. ab172730), 1:100; NANOG:¬¬ rabbit polyclonal (Bethyl Laboratories cat. no. A300-397A, lot no. 3), 1:100; KLF4: goat polyclonal (R&D Systems cat. no. AF3158, lot. no. WRR0719011), 1:100 for CUT&RUN, 1:50 for immunofluorescence; OCT4: rabbit monoclonal (EPR17929, Abcam cat. no. ab181557), 1:100; SOX2: rabbit monoclonal (EPR3131, abcam cat. no. ab92494), 1:100; CTCF: rabbit monoclonal (EPR7314(B), Abcam cat. no. ab128873), 1:100; rabbit anti-goat secondary: Abcam cat. no. ab6697, 1:100; goat anti-rabbit-Cy5 secondary: Jackson ImmunoResearch Cat. no. 111-175-144, 1:500; donkey anti-goat-rhodamine red secondary: Jackson ImmunoResearch Cat. no. 705-295-147, 1:250; mouse anti-H3 for Western blot: (mAbcam 24834, Abcam cat. no. ab24834), 1:500; IRDye 800CW goat anti-rabbit: LI-COR INC. Cat no. 926-32211, 1:10,000; IRDye 800CW donkey anti-goat: LI-COR INC. Cat no. 926-32214, 1:10,000; IRDye 680RD goat anti-mouse: LI-COR INC. Cat no. 926-68070, 1:10,000. |
| Validation | All antibodies were sourced commercially. RNAPII-S5P (CST 13523) and RNAPII-S2P (CST 13499) antibodies were validated by Cell Signaling Technologies using SimpleChIP® Enzymatic Chromatin IP Kits, and do not cross react.<br>BRG1 antibody (ab110641) was knock-out validated by abcam.<br>CTCF antibody (ab128873) was ChIP-Seq validated with ChIP-Kit Transcription Factors ChIP-Seq (ab270813) by abcam. |

## Eukaryotic cell lines

Policy information about cell lines and Sex and Gender in Research

| | |
|---|---|
| Cell line source(s) | AB2.2 (Male; strain: 129S5/SvEvBrd) mouse embryonic stem cells were used in all experiments. Frozen stock of cells were obtained from ATCC (ATCC SCRC-1023). |
| Authentication | Cell cultures were tested for Mycoplasma and karyotyped to detect any chromosomal abnormalities after thawing and at periodic intervals. Cells were not cultured for more than seven passages for any experiment. |
| Mycoplasma contamination | All cell lines were confirmed as Mycoplasma negative on a tri-monthly basis. |
| Commonly misidentified lines (See ICLAC register) | No commonly misidentified lines were used in this study. |

## ChIP-seq

### Data deposition

☒ Confirm that both raw and final processed data have been deposited in a public database such as GEO.

☒ Confirm that you have deposited or provided access to graph files (e.g. BED files) for the called peaks.

| | |
|---|---|
| Data access links<br>*May remain private before publication.* | https://urldefense.com/v3/__https://www.ncbi.nlm.nih.gov/geo/query/acc.cgi?acc=GSE224292__;!!GuAItXPztq0!mkMYN8s5c8cDPzIRwHc7Ok6WhX3or4EV-xQ1CZoZO73zAker5r1qbMQbfa7gjsZlHviPdoygyAUjG1E5$<br>(reviewer token: qvkzowmodnithgf) |
| Files in database submission | RNAPII-S5P_CUTAC_SL_0(SB_Mm_050222_SL_0_CUTAC)<br>RNAPII-S5P_CUTAC_SL_FLV_8h(SB_Mm_050222_SL_FL8_CUTAC)<br>RNAPII-S5P_CUTAC_SL(SB_Mm_SL_0506_0_S5_CUTAC)<br>RNAPII-S5P_CUTAC_2i(SB_Mm_2i_0506_0_S5_CUTAC)<br>RNAPII-S5P_CUTnTag_SL_DMSO(SB_Mm_ABSL_032421_0_S5P)<br>RNAPII-S5P_CUTnTag_SL_TRP_30m(SB_Mm_ABSL_032421_TR30_S5P)<br>RNAPII-S5P_CUTnTag_SL_TRP_1h(SB_Mm_ABSL_032421_TR1_S5P)<br>RNAPII-S5P_CUTnTag_SL_TRP_2h(SB_Mm_ABSL_032421_TR2_S5P) |

RNAPII-S5P_CUTnTag_SL_DMSO_2(SB_MmDmEc_0715_0_S5P)
RNAPII-S5P_CUTnTag_SL_FLV_30m(SB_MmDmEc_0715_F30_S5P)
RNAPII-S5P_CUTnTag_SL_FLV_1h(SB_MmDmEc_0715_F1_S5P)
RNAPII-S5P_CUTnTag_SL_FLV_4h(SB_MmDmEc_0715_F2_S5P)
RNAPII-S5P_CUTnTag_SL_ACT_30m(SB_MmDmEc_0715_A30_S5P)
RNAPII-S5P_CUTnTag_SL_ACT_1h(SB_MmDmEc_0715_A1_S5P)
RNAPII-S5P_CUTnTag_SL_ACT_4h(SB_MmDmEc_0715_A4_S5P)
BRG1_CUTnTag_SL_DMSO(SB_Mm_ABSL_032421_0_BRG)
BRG1_CUTnTag_SL_TRP_30m(SB_Mm_ABSL_032421_TR30_BRG)
BRG1_CUTnTag_SL_TRP_1h(SB_Mm_ABSL_032421_TR1_BRG)
BRG1_CUTnTag_SL_TRP_2h(SB_Mm_ABSL_032421_TR2_BRG)
BRG1_CUTnTag_SL_DMSO_2(SB_MmDmEc_0715_0_BRG1)
BRG1_CUTnTag_SL_FLV_30m(SB_MmDmEc_0715_F30_BRG1)
BRG1_CUTnTag_SL_FLV_1h(SB_MmDmEc_0715_F1_BRG1)
BRG1_CUTnTag_SL_FLV_4h(SB_MmDmEc_0715_F4_BRG1)
BRG1_CUTnTag_SL_ACT_30m(SB_MmDmEc_0715_A30_BRG1)
BRG1_CUTnTag_SL_ACT_1h(SB_MmDmEc_0715_A1_BRG1)
BRG1_CUTnTag_SL_ACT_4h(SB_MmDmEc_0715_A4_BRG1)
RNAPII-S5P_CUTnTag_SL(SB_Mm_AB2SL_0319_S5P)
RNAPII-S5P_CUTnTag_2i(SB_Mm_AB2S2i_0319_S5P)
BRG1_CUTnTag_SL(SB_Mm_AB2SL_0319_BRG1)
BRG1_CUTnTag_2i(SB_Mm_AB2S2i_0319_BRG1)
RNAPII-S2P_CUTnTag_SL(SB_MmDmEc_0715_0_S2P)
RNAPII-RPB3_CUTnTag_SL(TL_Mm_AB22_DMSO_Rpb3_TL2_122021)
H3K4me1_CUTnTag_SL(SB_Mm_AB2SL_0319_K4m1)
H3K4me3_CUTnTag_SL(SB_Mm_AB2SL_0319_K4m3)
H3K27me3_CUTnTag_SL(SB_Mm_AB2SL_0319_K27m3)
H3K27me3_CUTnTag_SL_DMSO(SB_Mm_0625_0_K27m)
H3K27me3_CUTnTag_SL_FLV_8h(SB_Mm_0625_FL8_K27m)
H3K9me3_CUTnTag_SL(SB_Mm_G4X40_0514_K9m3)
IgG_CUTnTag_SL(SB_Mm_AB2SL_0319_IgG)
BRG1_CUTnRUN_SL_DMSO(SB_MmSc_0620_0_Brg_in)
BRG1_H3K4me1_CUTnRUN.ChIP_SL_DMSO(SB_MmSc_0620_0_Brg_K4m1)
BRG1_H3K4me3_CUTnRUN.ChIP_SL_DMSO(SB_MmSc_0620_0_Brg_K4m3)
BRG1_H3K27me3_CUTnRUN.ChIP_SL_DMSO(SB_MmSc_0620_0_Brg_K27m3)
BRG1_IgG_CUTnRUN.ChIP_SL_DMSO(SB_MmSc_0620_0_Brg_IgG)
BRG1_CUTnRUN_SL_FLV(SB_MmSc_0620_FL8_Brg_in)
BRG1_H3K4me1_CUTnRUN.ChIP_SL_FLV(SB_MmSc_0620_FL8_Brg_K4m1)
BRG1_H3K4me3_CUTnRUN.ChIP_SL_FLV(SB_MmSc_0620_FL8_Brg_K4m3)
BRG1_H3K27me3_CUTnRUN.ChIP_SL_FLV(SB_MmSc_0620_FL8_Brg_K27m3)
BRG1_IgG_CUTnRUN.ChIP_SL_FLV(SB_MmSc_0620_FL8_Brg_IgG)
BRG1_CUTnRUN_SL(SB_MmScEc_SL_1217_BRG1_in)
BRG1_H3K4me1_CUTnRUN.ChIP_SL(SB_MmScEc_SL_1217_BRG1_K4m1)
BRG1_H3K4me3_CUTnRUN.ChIP_SL(SB_MmScEc_SL_1217_BRG1_K4m3)
BRG1_H3K27me3_CUTnRUN.ChIP_SL(SB_MmScEc_SL_1217_BRG1_K27m3)
BRG1_IgG_CUTnRUN.ChIP_SL(SB_MmScEc_SL_1217_BRG1_IgG)
BRG1_CUTnRUN_2i(SB_MmScEc_2i_1217_BRG1_in)
BRG1_H3K4me1_CUTnRUN.ChIP_2i(SB_MmScEc_2i_1217_BRG1_K4m1)
BRG1_H3K4me3_CUTnRUN.ChIP_2i(SB_MmScEc_2i_1217_BRG1_K4m3)
BRG1_H3K27me3_CUTnRUN.ChIP_2i(SB_MmScEc_2i_1217_BRG1_K27m3)
BRG1_IgG_CUTnRUN.ChIP_2i(SB_MmScEc_2i_1217_BRG1_IgG)
NANOG_CUTnRUN_SL(SB_MmScEc_SL_1217_NAN2)
SOX2_CUTnRUN_SL(SB_MmScEc_SL_1217_Sox2)
OCT4_CUTnRUN_SL(SB_MmScEc_SL_1217_Oct4)
KLF4_CUTnRUN_SL(SB_MmScEc_SL_1217_Klf4)
IgG_CUTnRUN_SL(SB_MmScEc_SL_1217_IgG)
NANOG_CUTnRUN_2i(SB_MmScEc_2i_1217_NAN2)
SOX2_CUTnRUN_2i(SB_MmScEc_2i_1217_Sox2)
OCT4_CUTnRUN_2i(SB_MmScEc_2i_1217_Oct4)
KLF4_CUTnRUN_2i(SB_MmScEc_2i_1217_Klf4)
IgG_CUTnRUN_2i(SB_MmScEc_2i_1217_IgG)

Genome browser session
(e.g. UCSC)

Integrated Genome Browser (IGB) sessions:
Fig 3: https://drive.google.com/file/d/1uu7zFtg4d2JqvuKoCBNtkaZA1-9dEyWG/view?usp=sharing
Fig 4: https://drive.google.com/file/d/1hj-vvZl7TjUfLijV71gBl-gvGUcVXvTl/view?usp=sharing
Extended Data Fig 1: https://drive.google.com/file/d/1cVAHxUt5iPaRDkBi1P273aWUHcCXorna/view?usp=sharing

## Methodology

Replicates

At least 2 biological replicates for each dataset were performed. Timepoints during drug treatment provide additional validation. All experiments were reproducible and replicates were consistent.

Sequencing depth

All Experiments were paired-end sequenced for 3-6 million reads

| Antibodies | RNAPII-S5P: rabbit monoclonal (D9N5I, Cell Signaling Technology cat. no. 13523), 1:50; RNAPII-S2P: rabbit monoclonal (E1Z3G, Cell Signaling Technology cat. no. 13499), 1:100; RPB3: rabbit polyclonal (Bethyl Laboratories cat. no. A303-771A, lot no. A303-771A2), 1:100; BRG1: rabbit monoclonal (EPNCIR111A, abcam cat. no. ab110641), 1:100, for CUT&Tag and CUT&RUN.ChIP, and rabbit polyclonal (Invitrogen cat. no. 720129, lot. no. 2068859), 1:250, for immunofluorescence; H3K4me1: rabbit polyclonal (Abcam cat. no. ab8895, lot no. GR3283237), 1:100; H3K4me3: rabbit polyclonal (Active Motif cat. no. 39915, lot. no. 24118008), 1:100, for CUT&Tag, and rabbit monoclonal (EpiCypher cat. no. 13-0028), 1:100, for CUT&RUN.ChIP; H3K27me3: rabbit monoclonal (C36B11, Cell Signaling Technology cat. no. 9733), 1:100; H3K9me3: rabbit monoclonal (EPR16601, Abcam cat. no. ab176916), 1:100; guinea pig anti-rabbit secondary: Antibodies Online cat. no. ABIN101961, 1:100; isotype control (IgG) for CUT&RUN.ChIP: rabbit monoclonal (EPR25A, Abcam cat. no. ab172730), 1:100; NANOG:¬¬¬ rabbit polyclonal (Bethyl Laboratories cat. no. A300-397A, lot no. 3), 1:100; KLF4: goat polyclonal (R&D Systems cat. no. AF3158, lot. no. WRR0719011), 1:100 for CUT&RUN, 1:50 for immunofluorescence; OCT4: rabbit monoclonal (EPR17929, Abcam cat. no. ab181557), 1:100; SOX2: rabbit monoclonal (EPR3131, abcam cat. no. ab92494), 1:100; CTCF: rabbit monoclonal (EPR7314(B), Abcam cat. no. ab128873), 1:100; rabbit anti-goat secondary: Abcam cat. no. ab6697, 1:100; goat anti-rabbit-Cy5 secondary: Jackson ImmunoResearch Cat. no. 111-175-144, 1:500; donkey anti-goat-rhodamine red secondary: Jackson ImmunoResearch Cat. no. 705-295-147, 1:250; mouse anti-H3 for Western blot: (mAbcam 24834, Abcam cat. no. ab24834), 1:500; IRDye 800CW goat anti-rabbit: LI-COR INC. Cat no. 926-32211, 1:10,000; IRDye 800CW donkey anti-goat: LI-COR INC. Cat no. 926-32214, 1:10,000; IRDye 680RD goat anti-mouse: LI-COR INC. Cat no. 926-68070, 1:10,000. |
|---|---|
| Peak calling parameters | SEACR v 1.3 was used for peak calling. Parameters are described in the Methods section. |
| Data quality | SEACR peaks were called using default FDR and "relaxed" parameter |
| Software | Bowtie 2 was used to clip adapters post sequencing and map paired-end Mus musculus reads to UCSC mm10, and spike-in E. coli reads to Ensembl masked R64-1-1. Continuous-valued data tracks (bedGraph and bigWig) were generated using genomecov in bedtools v2.30.0 Genomic tracks were displayed using Integrated Genome Browser v 8.5.4. Peaks were called by SEACR (v.1.3). Profile plots, heatmaps, and correlation matrices were generated using deepTools v3.5.1. Violin plots were generated with GraphPad Prism 9, scores were computed using deepTools v3.5.1. Western blot images were developed and bands quantified using ImageJ version 1.53t 24. |

