## [Peer Review File · Nature Genetics]

Peer Review Information

Manuscript Title: The BAF chromatin remodeler synergizes with RNA polymerase II and transcription factors to evict nucleosomes

Corresponding author name(s): Dr Steven Henikoff, Dr Sandipan Brahma

Reviewer Comments & Decisions:

Decision Letter, initial version:

13th Mar 2023

Dear Steve,

Your Article, entitled "RNA Polymerase II, the BAF remodeler and transcription factors synergize to evict nucleosomes", has now been seen by 4 referees. You will see from their comments copied below that while they find your work of potential interest, they find the conclusions preliminary and they have raised quite substantial concerns that must be thoroughly addressed. In light of these comments, we cannot accept the manuscript for publication, but would be interested in considering a revised version that addresses these serious concerns.

We hope you will find the referees' comments useful as you decide how to proceed. If you wish to submit a substantially revised manuscript, please bear in mind that we will be reluctant to approach the referees again in the absence of major revisions.

If you choose to revise your manuscript taking into account all reviewer comments, please highlight all changes in the manuscript text file. At this stage we will need you to upload a copy of the manuscript in MS Word .docx or similar editable format.

We are committed to providing a fair and constructive peer-review process. Do not hesitate to contact me if there are specific requests from the reviewers that you believe are technically impossible or unlikely to yield a meaningful outcome. I would be happy to discuss the reviewers' comments in detail to help guide the revision process.

*1) Include a "Response to referees" document detailing, point-by-point, how you addressed each referee comment. If no action was taken to address a point, you must provide a compelling argument.

This response will be sent back to the referees along with the revised manuscript.

*2) If you have not done so already please begin to revise your manuscript so that it conforms to our Article format instructions, available [here](http://www.nature.com/ng/authors/article_types/index.html). Refer also to any guidelines provided in this letter.

[redacted]

If you wish to submit a suitably revised manuscript we would hope to receive it within 6 months. If you cannot send it within this time, please let us know. We will be happy to consider your revision so long as nothing similar has been accepted for publication at Nature Genetics or published elsewhere. Should your manuscript be substantially delayed without notifying us in advance and your article is eventually published, the received date would be that of the revised, not the original, version.

Thank you for the opportunity to review your work.

Sincerely,

Tiago

Tiago Faial, PhD
Chief Editor
Nature Genetics
<https://orcid.org/0000-0003-0864-1200>

Reviewers' Comments:

Reviewer #1:

Remarks to the Author:

This paper claims to demonstrate that RNAPII promoter-proximal-pausing stabilizes the chromatin occupancy of the SWI/SNF type remodeler BAF, and this paused RNAPII enhances nucleosome eviction by BAF. The mechanistic relationship of chromatin remodelers, sequence-specific transcription factors, and paused RNAPII at promoters and enhancers is an important component of gene regulation and warrants investigation. The Henikoff lab has an impressive record of technology development and associated discoveries; unfortunately, the current manuscript overstates conclusions and has several experimental shortcomings. As it stands, the conclusions that can be drawn from these data do not in my opinion move the field sufficiently forward to warrant publication in Nature Genetics.

Major issues:

1. The authors overinterpret their data in the text throughout this manuscript. One early example of this is the claimed dependence of BRG1 on paused polymerase for recruitment, i.e. the section subheading "BAF chromatin occupancy is strongly dependent on RNAPII". The authors show that polymerase and BRG1 share similar occupancy patterns and those patterns change in similar, though not identical, ways after perturbation of transcription; however, this does not conclusively demonstrate dependence of one factor on the other for recruitment. The assertion that RNAPII occupancy is required for BAF recruitment is particularly strange and does not make sense logically. How is RNAPII initiating, if nucleosomes have not yet been evicted from the promoter region? To me, a more likely explanation is that BAF first binds and clears nucleosomes so RNAPII can initiate, then pause RNAPII stabilizes the BAF binding by holding promoter chromatin in an open state. In any case, the authors do not have data to conclusively support one model over other possibilities.
2. The novelty and significance of their findings are overstated by the authors. This relates to point 1, as the authors claim to have data supporting a surprising finding (that paused Pol II is required for BAF to bind), but their data does not actually support this conclusion and in fact, the data that they report is very similar to what has been reported previously (PMID: 27768892, PMID: 33303640, PMID: 30955888), and these paper drew different conclusions.
3. Proper controls for RNAPII-S5P and BRG1 CUT&Tag are lacking. It is well known that Tn5 transposition patterns closely mimic genome-wide binding profiles of RNAPII (ATAC-seq). In this paper the authors use CUT&Tag, which uses antibody-directed Tn5, to profile genome wide binding patterns of RNAPII-S5P and BRG1. They then correlate changes in this signal upon treatment with various inhibitors of transcription and claim that this demonstrates co-dependence of RNAPII-S5P and BRG1 binding. However, the null expectation should be that perturbing transcription will cause changes in accessibility at polymerase-bound sites, therefore, without a measurement of non-specific transposition under the same conditions to compare to, we cannot be sure whether the change in signal observed is specific (antibody-directed) or nonspecific background. The authors must complete experiments to demonstrate that the reported changes in BRG1 occupancy are not simply due to accessibility changes after loss of transcription.
4. The statement that "BRG1 CUT&Tag showed the same patterns as RNAPIIS5P, which further demonstrates the strong dependence of BAF on paused RNAPII for chromatin binding (Fig. 1f-h)" is not supported quantitatively by the actual plots in f and h, where the curves of S5P and BRG for the two inhibitors behave differently.
5. On page 5, the statement "Flavopiridol resulted in a dramatic depletion of the partially unwrapped

nucleosomal intermediates over S5P CUTAC peaks and promoter NDRs" is supported by Fig.2d to 2c comparison but not by the comparable Extended data Fig. 3 d to c comparison for the H3K4me3 mapping."

6. Why is the long 8 hr Flavopiridol time chosen for experiments reported in Fig 2 and Extended data Fig 3? Changes in Pol II following treatment happen at much shorter times. Are the cells still healthy at 8 hrs?

7. Fig. 4. In testing the role of sequence-specific, pluripotency TFs in the generation of nucleosome depleted regions (NDRs), this paper describes the use of different culture conditions (LIF vs. 2i) to change levels of these TFs. This strategy and the analysis of TF changes has its shortcomings.

a) The effects of media change can be complex on the cells and resulting effects hard to assign to a specific TF or set of TFs. Also, if SL cell handling is optimal, the differences from 2i conditions can be extremely modest. It strikes me that a more directed approach would be to use short time frame degen experiments to deplete a specific TF or sets of TFs.

b) The quantification of the TF depletion by Western blot would be helpful in assessing these experiments. The immunofluorescent signals shown in Extended Data Fig 5 are hard to interpret quantitatively.

c) The fact that only some of these TFs shows a major difference in occupancy as seen in CUT&RUN ChIP could mean the other pluripotent TFs are sufficient to complement. A controlled depletion of combinations of this set of TFs could make a more compelling conclusion. As it stands, the conclusion "that pluripotency TFs do not recruit BAF to chromatin" is unwarranted.

8. The conclusion that seems somewhat solid is that "RNAPII promoter-proximal pausing (appears to) stabilize BAF chromatin occupancy and enhances nucleosome eviction by BAF." However, the result that RNAPII promoter-proximal pausing is associated with nucleosome eviction was first shown over a dozen years ago (PMID: 21074046)

9. Finally, the elaborate speculative Discussion with selective citation of the literature goes far beyond the observations made in this paper.

Reviewer #2:

Remarks to the Author:

What recruits BAF?

Why only dependence at enhancers?

Is it dependent on BAF?

S5/BAF disconnect at bivalent promoters

Why is h3k27me3 more efficiently evicted with fla

Brahma and Henikoff explore the relationship between RNA Pol II pausing/elongation and chromatin remodeling in mouse ESCs using small molecule inhibitors of transcription. Using a low salt CUT&TAG approach to identify RNA Pol II S5-phosphorylated associated accessible regions (S5 CUTAC) that they previously published, the authors show that BRG1 associates with S5 CUTAC regions. While triptolide decreases both Pol II-S5 and BRG1, flavopiridol and actinomycin D increase Pol II-S5 and BRG1 binding, suggesting that Pol II pausing increases BRG1 occupancy. Further, looking at BRG1-associated nucleosomes, they find that flavopiridol increases eviction of partially remodeled nucleosomes. Flavopiridol and actinomycin D also increase Pol II-S5 and BRG1 occupancy at bivalent promoters, resulting in minor decreases in H3K27me3 and accessible fragment length. Finally, the authors show that mouse ESCs cultured in 2i have increased binding of OCT4, SOX2, NANOG, KLF4 (but not BRG1) than those cultured in LIF+serum. 2i conditions also result in more eviction of BRG1-

associated partially remodeled nucleosomes.

This is an interesting study and offers a lot to consider. The data are well-executed and overall propose a new model for the generation of nucleosome depleted regions at transcriptional start sites. However, the model lacks genetic proof and the data presented are largely correlative. Additional genetic/chemical genetic evidence for BAF and TF requirements for nucleosome eviction are required.

Major Points:

- 1) The BRG1 CUT&RUN.ChIP is a nice assay to show that BRG1-associated fragments smaller than 120bp are nevertheless enriched in histone modifications, showing that these fragments arise from partially remodeled nucleosomes. It is this population that the authors see are preferentially evicted with flavopiridol or 2i treatment. However, there is no formal evidence that BAF is evicting these nucleosomes, versus perhaps another co-associated or co-bound remodeler given that other remodelers are also bound at active regulatory regions (Morrison NSMB 2014, Ram Cell 2011). Inhibition studies with a number of different chemical BAF inhibitors (BRM014) or degraders (ACBI1, AU-15330) would formally prove this.
- 2) Related, prior studies have not shown a strong BAF dependence for promoter/TSS accessibility by ATAC studies, but rather, a preferential dependence at enhancers/distal regulatory elements (Iurlaro Nat Genetics 2021, Schick Nat Genetics 2021, Kelso eLife 2017, Nakayama Nat Genetics 2017). Does eviction of BRG1-associated nucleosomes differ at promoters versus enhancers in flavopiridol treatment? From data in Extended Data Figure 3 c,d, it looks frankly like there is very little change in the broken line between DMSO and flavopiridol. Similarly, following point 1, would the authors see different dependence on BAF at promoters versus enhancers for flavopiridol-induced eviction?
- 3) Data from Tolstorukov et al PNAS 2013 demonstrate that genetic inactivation of Brg1 or Snf5 reduces nucleosome occupancy and phasing at the +1 and -1 nucleosomes flanking the NDR. How, if at all, do these results contribute to the authors' model?
- 4) The authors determine that BRG1-associated eviction is stronger in 2i conditions and correlate this with increased TF binding genome wide in 2i conditions. I think it's fair to say that this is highly correlative as a number of known and unknown things are different between LIF+serum and 2i conditions. Again, it would be imperative to show the dependence on pluripotency factors for this productive remodeling using genetic experiments. This could be accomplished for example using addback of TF mutants in a dox-inducible deletion line or by introducing a mutation within a TF binding site using CRISPR editing at a one/several sites in the genome.
- 5) Likewise, the authors make bold claims about the lack of global BAF increase in 2i conditions with increased TF binding being conclusive evidence that TFs do not recruit BAF to chromatin. It's not obvious to me that TFs and BAF would scale 1:1 globally in this way as the authors predict. Further, this is in direct contrast to papers showing a role for TFs in BAF recruitment (Minderjahn Nat Comm 2020, Priam Nat Genetics 2017, Sandoval Mol Cell 2018) and dependence on TFs for BAF binding (King and Klose eLife 2017 in mESCs, others in other cell systems). OCT4 in particular was shown to be required to maintain BAF binding in mESCs using dox-inducible Oct4 deletion (King and Klose eLife 2017). In Figure 4c, among the TFs shown, OCT4 is the one TF that is minimally increased in 2i (actually, the OCT4 data quality looks quite poor – the authors will need to revisit this). The authors could analyze clusters of sites based on gained/maintained/lost TF binding between 2i and serum+LIF conditions and inspect BAF gains at these regions, possibly also separated into promoters and enhancers. However, independent of what the authors find with this clustering, without genetic evidence, this is for me an unsubstantiated argument.
- 6) Related to point 4...what does recruit BAF if not TFs? Based on the authors' logic for TFs now

applied to RNA Pol II, more BAF in conditions that promote more Pol II binding mean that Pol II recruits BAF. The authors will need to add their thoughts to the discussion.

7) The authors make the observation that Pol II-S5 and BAF can be increased with flavopiridol at bivalent promoters. BRG1-associated H3K27me3 modified nucleosomes are more efficiently evicted under these conditions (and 2i), very similar to H3K4me1 modified histones. Curiously, there seems to be no preference for the different modification states, which is in contrast to work showing that the binding and activity of BAF is regulated by histone PTMs (Mashtalir Science 2021). Despite productive eviction of H3K27me3 modified histones following flavopiridol treatment as measured by the BRG1 CUT&RUN.ChIP assay, this is not sufficient for NDR establishment. This doesn't agree with the authors' model—they link productive eviction following TF binding with NDR establishment (step 3 in Figure 5). This indicates that there is either a flaw in the interpretation of the BRG1 CUT&RUN.ChIP assay or a flaw in the model.

Minor Points: Extended Figure 4b is cut off.

Reviewer #3:

Remarks to the Author:

BAF chromatin remodeling complexes are key for nearly all developmental changes in gene expression, and they have a dominant influence on the accessibility landscape of active genes promoters and enhancers. Previous work has shown that subunits of the ESC-specific BAF complex (esBAF), Brg1, is essential for chromatin accessibility at a large number of active genes in ESCs, and that BAF is essential for ESC differentiation. Furthermore, studies have found that esBAF is dependent on pluripotent transcription factors, such as OCT4, for promoter occupancy. Here, Brahma and Henikoff investigate the role of promoter-proximal paused RNAPII in BAF occupancy and nucleosome depletion at promoters (NDR formation). Using high resolution Cut&Tag methodologies and chemical treatments that block steps in the RNAPII cycle, the authors present compelling evidence that paused RNAPII facilitates recruitment/retention of Brg1 at active genes. Furthermore, enhancing the level of Brg1/BAF at genes, by increasing the levels of paused RNAPII, leads to more efficient NDR formation.

In general, the data are of high quality and there were no technical concerns. My only issues surround the interpretation of the role of TFs in BAF recruitment, for which the authors have not provided much, if any, compelling data. Their conclusions surrounding the role of TFs need to be tempered considerably in their discussion and the model presented in Figure 7, taking into account other data published previously.

Specific points:

The authors have over-simplified their description of mammalian BAF complexes. They should note that BAF complexes have several different forms – cBAF, PBAF, ncBAF, and esBAF. The esBAF complex is specific to ESCs and contains the BAF250a subunit, and not BAF200. Note that this complex also lacks the polybromo subunit, BAF180. Both the cBAF and esBAF complexes are most highly related to yeast SWI/SNF, not RSC complex. The RSC complex is most related to the PBAF complex. These distinctions should be taken into account when the authors compare their work in ESCs to their previous work investigating the role of yeast RSC in NDR formation. Perhaps just stating that esBAF is related to yeast RSC, rather than “homologous”.

The authors cite key work that has demonstrated essential roles for BAF complexes in promoting the binding of pluripotency TFs in ESCs, however, they fail to mention that these TFs (such as OCT4 and SOX2) are also essential for Brg1/BAF occupancy (ref. 45 and 46). Notably, these previous data used rapid removal of OCT4 or SOX2, leading to loss of BAF.

The authors use different media conditions to vary the expression levels of several pluripotency TFs, and then the monitor impact on Brg1 occupancy and NDR formation. By changing from SL to 2i, they see a 3x increase in KLF4 occupancy, small increases for SOX2 and NANOG, and no changes in OCT4. Based on these data, the authors made the broad conclusion that pluripotency TFs do not play a role in BAF recruitment (Discussion and Figure 7). This seems to be an over-interpretation of the data. There are several reports that OCT4 is key for Brg1 recruitment to targets in ESCs (see above), and the authors' data do not address this. This reviewer is not aware of data suggesting that KLF4 would be the key, limiting TF for BAF recruitment. Furthermore, it would seem that the authors should mention the large amount of data showing that BAF complexes physical interact with many TFs, and that these TFs can recruit BAF to target loci both in vitro and in vivo (e.g. recent work from the Zaret group in NSMB, and older work on the role of non-DNA binding domains of PU.1).

In the authors' model (Figure 7), but it is not clear to this reviewer how specificity is provided if gene-specific TFs play no role in RNAPII occupancy. If RNAPII occupies all exposed promoters following replication fork passage, then all genes and cryptic promoters would be occupied. Their model may explain maintenance of an established transcription program, but it does not explain how cell-type specificity is established. This model would predict that loss of TFs, like OCT4, would have little impact on the levels of paused RNAPII. If this has not been shown previously, then perhaps this could be tested.

Reviewer #4:

Remarks to the Author:

In the presented study, Brahma and Henikoff use a wide array of assays (CUT&Tag, S5P CUTAC, CUT&RUN.ChIP) in combination with RNAPII inhibition to test the role of RNAPII in targeting BRG1 (~SWI/SNF) to promoter NDRs in mESCs. In addition, they assessed the role of the pluripotency transcription factors (Nanog, Sox2, Oct4 and Klf4) in NDR formation. The authors used three different inhibitors of RNAPII transcription to either block initiation (Triptolite) or block elongation (Flavopiridol and Actinomycin D). Blocking elongation results in the accumulation of paused RNAPII. Interestingly, increased RNAPII pausing is accompanied by increased BRG1 occupancy at promoter NDRs, and across the genome (Fig. 1 & extended data Fig. 1 & 2). Flavopiridol also promotes increased nucleosome eviction at NDRs (Fig. 2). Transcription elongation inhibition also leads to increased RNAPII and BRG1 occupancy at Polycomb-repressed promoters (Fig. 3), suggesting these factors transiently probe these regions. Finally, the authors show that the pluripotency transcription factors promote NDR formation (Fig. 4). These findings are summarized in a model showing that transcription factors and RNAPII cooperate in the recruitment of the SWI/SNF remodeler to establish/maintain NDRs at promoters.

The work described in this manuscript is a technical tour-de-force and the data is convincing. Although the interplay between sequence-specific transcription factors, RNAPII and remodelers have been described extensively, the results presented here provide new insights into the recruitment of the SWI/SNF remodeler to promoters. However, in my opinion this study is in need of additional functional

experiments.

Major concerns:

- 1) It is important to show that the changes in chromatin structure are actually dependent on BRG1. This could e.g. be tested by using the inhibitor BRM014. Additionally, the effect of BRG1 inhibition (or loss) on gene transcription needs to be included in the analysis.
- 2) The authors emphasize BRG1 recruitment by paused RNAPII. However, the effects of transcriptional inhibition are apparently not restricted to promoter regions. It is imperative that a clear overview is presented of all BRG1 peaks and all RNAPII peaks and to what extent they overlap. The ext. data provided in Fig. 1d and Fig. 2c-j does not suffice.
- 3) Related to point 2), S5P-CUTAC selects for paused RNAPII loci. What is the effect on SWI/SNF remodeling at intergenic loci? This needs to be addressed by including ATAC-seq.
- 4) My understanding from the manuscript is that transcription inhibition with Flavopiridol also promotes BRG1 recruitment to intergenic loci. Assuming these sites lack RNAPII, doesn't this contradict with the model presented?

Minor comments:

- 5) Could drug-induced RNAPII pausing simply create a physical barrier to nucleosome sliding, resulting in increased eviction?
- 6) I don't want to start a discussion on bivalency, but the heatmaps in Fig. 3b suggest to me that the sites labelled " bivalent" are simply Polycomb-repressed. H3K27me3 is much more prevalent than H3K4me3.
- 7) There is a graded scale between unwrapping and eviction. The effects are presented a bit too much as all or nothing.

Author Rebuttal to Initial comments

Your Article, entitled "RNA Polymerase II, the BAF remodeler and transcription factors synergize to evict nucleosomes", has now been seen by 4 referees. You will see from their comments copied below that while they find your work of potential interest, they find the conclusions preliminary and they have raised quite substantial concerns that must be thoroughly addressed. In light of these comments, we cannot accept the manuscript for publication, but would be interested in considering a revised version that addresses these serious concerns.

We thank the Reviewers for thorough and thoughtful review of our manuscript, critical comments, and overall positive feedback. We have addressed each of the Reviewer's concerns point-by-point.

Reviewer #1:

Remarks to the Author:

This paper claims to demonstrate that RNAPII promoter-proximal-pausing stabilizes the chromatin occupancy of the SWI/SNF type remodeler BAF, and this paused RNAPII enhances nucleosome eviction by BAF. The mechanistic relationship of chromatin remodelers, sequence-specific transcription factors, and paused RNAPII at promoters and enhancers is an important component of gene regulation and warrants investigation.

We thank Reviewer 1 for highlighting the significance of our study.

The Henikoff lab has an impressive record of technology development and associated discoveries; unfortunately, the current manuscript overstates conclusions and has several experimental shortcomings. As it stands, the conclusions that can be drawn from these data do not in my opinion move the field sufficiently forward to warrant publication in Nature Genetics.

We have added experiments and clarified the text to address these concerns, described below.

Major issues:

1. The authors overinterpret their data in the text throughout this manuscript. One early example of this is the claimed dependence of BRG1 on paused polymerase for recruitment, i.e. the section subheading "BAF chromatin occupancy is strongly dependent on RNAPII". The authors show that polymerase and BRG1 share similar occupancy patterns and those patterns change in similar, though not identical, ways after perturbation of transcription; however, this does not conclusively demonstrate dependence of one factor on the other for

recruitment. The assertion that RNAPII occupancy is required for BAF recruitment is particularly strange and does not make sense logically. How is RNAPII initiating, if nucleosomes have not yet been evicted from the promoter region? To me, a more likely explanation is that BAF first binds and clears nucleosomes so RNAPII can initiate, then pause RNAPII stabilizes the BAF binding by holding promoter chromatin in an open state. In any case, the authors do not have data to conclusively support one model over other possibilities.

We have revised the mentioned section subheading to "RNAPII promoter-proximal pausing strongly promotes BAF chromatin occupancy". We have also revised the text thoroughly to make our interpretations clearer. Our interpretation is not that RNAPII is required for BAF recruitment. Rather, we say that paused RNAPII facilitates or stabilizes BAF occupancy, or the likelihood of a particular loci to be bound by BAF in a population of cells, which Reviewer 1 agrees with. Our model posits that BAF and RNAPII can bind independently, and RNAPII takes advantage of a cycle of nucleosome deposition and eviction at promoters and enhancers, which we proposed earlier in yeast (PMID: 30554944), and was later demonstrated in mouse ESCs (PMID: 33558757). Therefore, the question of which comes first (RNAPII or BAF) is moot. We have also thoroughly revised relevant text to clarify our logic and avoid misinterpretation about BAF recruitment, for example, entirely removing the metaphors "recruit" and "recruitment", which can have multiple physical interpretations. Rather, based on our data, we suggest that BAF need not be "recruited" but binds genomic regions broadly and dynamically while paused RNAPII stabilizes its occupancy, and both paused RNAPII and pluripotency TF chromatin binding in mESCs drive nucleosome eviction by BAF, implying a functional synergy between these factors for nucleosome eviction and maintaining locus-specific chromatin accessibility patterns. This mechanistic relationship between RNAPII transcription and BAF chromatin remodeling was previously unknown and broadly provides new insights into how chromatin dynamics is regulated in development and can be deregulated in cancers with recurrent mutations in the BAF complex.

2. The novelty and significance of their findings are overstated by the authors. This relates to point 1, as the authors claim to have data supporting a surprising finding (that paused Pol II is required for BAF to bind), but their data does not actually support this conclusion and in

fact, the data that they report is very similar to what has been reported previously (PMID: 27768892, PMID: 33303640, PMID: 30955888), and these paper drew different conclusions.

Our Triptolide experiments (Fig. 1 and Extended Data Fig. 2) clearly show that without paused RNAPII, BAF chromatin occupancy is rapidly lost, showing that paused RNAPII is indeed required for stable or persistent BAF chromatin occupancy. However, we do not claim that paused RNAPII is required for BAF to bind. BAF may bind independently of RNAPII, as we have explained above in response to point 1. The three publications mentioned by Reviewer 1 do not show “very similar” data but “different conclusions”, None of these studies used drugs to inhibit transcriptional steps, which is the crux of our study. PMID: 27768892 is an in vitro reconstitution study of how yeast remodelers regulate nucleosome positioning, with no data on RNAPII. Nevertheless, this study demonstrates that RSC (a SWI/SNF family remodeler in yeast) generates NDRs, which is consistent with our previous work on yeast RSC (PMID: 30554944) and our current manuscript. PMID: 33303640 shows that Drosophila BAF (SWI/SNF) along with the TF GAF facilitates RNAPII binding and promoter-proximal pausing but did not examine the effect of RNAPII on BAF, which is central and unique to our study. However the idea of TF-SWI/SNF cooperativity in this paper is consistent with our model, and we have appropriately cited it in the Discussion section (lines 360-362). PMID: 30955888 showed that chromatin binding of the TF NANOG is dependent on BAF in blastocysts, consistent with our model and other published literature that BAF remodeling makes way for pluripotency TF binding, and we have cited this paper (line 305). Interestingly, the difference that the authors observed between blastocysts and embryonic stem cells in culture might be partly explained by differential NANOG expression and its effect on BAF remodeling, similar to our observation of differences between mESC SL vs 2i culture conditions as shown in Fig. 5 of our revised manuscript.

3. Proper controls for RNAPII-S5P and BRG1 CUT&Tag are lacking. It is well known that Tn5 transposition patterns closely mimic genome-wide binding profiles of RNAPII (ATAC-seq). In this paper the authors use CUT&Tag, which uses antibody-directed Tn5, to profile genome wide binding patterns of RNAPII-S5P and BRG1. They then correlate changes in this signal upon treatment with various inhibitors of transcription and claim that this demonstrates co-dependence of RNAPII-S5P and BRG1 binding. However, the null expectation should be that

perturbing transcription will cause changes in accessibility at polymerase-bound sites, therefore, without a measurement of non-specific transposition under the same conditions to compare to, we cannot be sure whether the change in signal observed is specific (antibody-directed) or nonspecific background. The authors must complete experiments to demonstrate that the reported changes in BRG1 occupancy are not simply due to accessibility changes after loss of transcription.

This null expectation is not correct. It is correct that the Tn5 moiety used in ATAC-seq binds exposed DNA and so tags accessible sites, but this non-specific binding is abolished in CUT&Tag. As demonstrated by us (PMID: 32913232, Box 2 and Figure 6) and others (bioRxiv: <https://doi.org/10.1101/2021.08.14.456176>), some antibody-directed Tn5 methods (ACT-Seq PMID: 31431618 and CoBATCH PMID: 31471188) indeed show strong accessible chromatin patterns. To avoid this ATAC-seq artifact, our original CUT&Tag method involves extensive 300 mM NaCl washes to remove unbound pA-Tn5 and the tagmentation step is also carried out in 300 mM NaCl, which effectively suppresses tagmentation of accessible DNA to ensure high-fidelity mapping of the antibody-targeted proteins (PMID: 31036827). Indeed, the current manuscript includes experiments that demonstrate CUT&Tag does not reproduce genome-wide accessibility or non-specific background: we find that there are no changes in BRG1 or RNAPII-S5P occupancy despite increased accessibility and TF binding in 2i media (Fig. 5c, Extended Data Fig. 6g). To further demonstrate that the antibodies are specific and CUT&Tag is not reporting on DNA accessibility, we performed CUT&Tag targeting the histone post-transcriptional modification H3K27ac, which marks both active promoters and enhancers, which are also occupied by RNAPII-S5P. Interestingly, CUT&Tag for H3K27ac showed a gradual decrease in signal over a time course of transcription inhibition in contrast to the increase in BRG1 and RNAPII-S5P at the same time points (Extended Data Fig. 2i). These results further confirm that our CUT&Tag protocol does not simply capture increased accessibility at RNAPII bound sites upon transcription inhibition and support our observation of increased nucleosome eviction upon transcription inhibition. We have included the H3K27ac CUT&Tag results in Extended data Fig 2i and discussed in lines 184-188

4. The statement that "BRG1 CUT&Tag showed the same patterns as RNAPII-S5P, which further demonstrates the strong dependence of BAF on paused RNAPII for chromatin

binding (Fig. 1f-h)" is not supported quantitatively by the actual plots in f and h, where the curves of S5P and BRG for the two inhibitors behave differently.

The reviewer is correct, and we have modified this statement in the text to clarify our point and avoid misunderstanding. We suggest that the quantitative differences in how BRG1 and RNAPIIS5p respond to inhibitors are due to the distinct modes of action of Actinomycin D and Flavopiridol. Actinomycin D traps paused RNAPII on chromatin, while RNAPII that is inhibited from going on to productive elongation by Flavopiridol rapidly turns over. Correspondingly, trapped RNAPII that does not turn over results in a stronger buildup of BRG1 (BAF) over time. These differences are discussed on lines 163-170 of the text.

5. On page 5, the statement "Flavopiridol resulted in a dramatic depletion of the partially unwrapped nucleosomal intermediates over S5P CUTAC peaks and promoter NDRs" is supported by Fig.2d to 2c comparison but not by the comparable Extended data Fig. 3 d to c comparison for the H3K4me3 mapping."

We thank Reviewer 1 for pointing us to this difference, which Reviewer 2 has also mentioned. We have explained this difference in the revised manuscript as stronger dependence of active enhancer regions on BAF remodeling as compared to active promoters. This explanation is supported by previous publication from Dirk Schübeler's lab (PMID: 33558757) and a recent preprint from Karen Adelman's lab (doi.org/10.1101/2023.03.07.531379) testing the effects of BRG1 (BAF) ATPase inhibition on chromatin accessibility and transcription. We now refer to these results in clarifying this point in the text (lines 228-232). In revised Extended Data Fig. 3, we have now more carefully compared promoters with distal regulatory regions (enhancers) called by RNAPII-S5P CUTAC accessibility and TF occupancy, which shows the distinction even better.

6. Why is the long 8 hr Flavopiridol time chosen for experiments reported in Fig 2 and Extended data Fig 3? Changes in Pol II following treatment happen at much shorter times. Are the cells still healthy at 8 hrs?

We chose 8 hr time points to demonstrate more robust effects. Colony morphology looked normal under the microscope and there was no reduction in cell number. We now make this point in the text (lines 225-228).

7. Fig. 4. In testing the role of sequence-specific, pluripotency TFs in the generation of nucleosome depleted regions (NDRs), this paper describes the use of different culture conditions (LIF vs. 2i) to change levels of these TFs. This strategy and the analysis of TF changes has its shortcomings.

a) The effects of media change can be complex on the cells and resulting effects hard to assign to a specific TF or set of TFs.

To minimize potential indirect effects, we analyzed only the specific sets of loci that were bound by the TFs in both serum-LIF (SL) and 2i conditions. Furthermore, this strategy overcomes the drawbacks of ectopically overexpressing TFs which has somewhat biased interpretations for TF pioneering activity (see PMIDs: 34984978 and 36253868). Additionally, this strategy mimics stages in embryonic development, and is therefore physiologically relevant, as we have noted in the manuscript (lines 322-329).

Also, if SL cell handling is optimal, the differences from 2i conditions can be extremely modest. It strikes me that a more directed approach would be to use short time frame degron experiments to deplete a specific TF or sets of TFs.

Our goal was not to assign specific TFs or sets of TFs to effects at particular sites, and TF redundancy means that the effects of depletion cannot be easily predicted. Also, depletion of essential TFs is drastic and non-physiological, and we wanted to test our model using a non-toxic manipulation (i.e., using the 2i inhibitors) that is routinely used in many mouse ES cell studies (e.g., PMIDs: 30914406, 25720369, 29636490, 31142785). Specifically, our goal in performing this experiment was to determine whether modest genome-wide upregulation of

TF occupancies that result from a widely used culture medium change result in enhanced nucleosome eviction, and this is what we observed. We have modified the text to better explain our rationale (lines 312-323).

b) The quantification of the TF depletion by Western blot would be helpful in assessing these experiments. The immunofluorescent signals shown in Extended Data Fig 5 are hard to interpret quantitatively.

We now quantitatively compared expression of the TFs NANOG, SOX2, OCT4, and KLF4, as well as BRG1 in 2i vs SL by Western blotting, and included the data in revised Extended Data Fig. 5f. The results are consistent with the immunofluorescence assays and CUT&RUN mapping.

c) The fact that only some of these TFs shows a major difference in occupancy as seen in CUT&RUN ChIP could mean the other pluripotent TFs are sufficient to complement. A controlled depletion of combinations of this set of TFs could make a more compelling conclusion. As it stands, the conclusion "that pluripotency TFs do not recruit BAF to chromatin" is unwarranted.

The experiments proposed by Reviewer 1 to deplete TFs or sets of TFs might show a loss of BAF occupancy, but that has already been established by others (e.g., ref. 45-46, as pointed out below by Reviewer 3, are now ref. 58-59 in the revised manuscript), and is not the subject of our manuscript. The reason for performing the SL-2i experiments was to test whether modest upregulation of pluripotency TFs has the same effect in promoting nucleosome eviction at these sites as treatments that upregulate paused RNAPII and stabilized BAF, which is the subject of our manuscript. However, we found that BAF occupancy does not change despite enhanced eviction of nucleosomes upon culture medium change. Since the culture medium change resulted in NANOG and KLF4 upregulation, our expectation is that this change should have increased BAF occupancy at the TF-binding sites if BAF had indeed been recruited by these TFs, which was not observed. Nevertheless, since BAF recruitment is not the main focus of our manuscript, we have reworded the statement to avoid confusion and removed the point

about recruitment: "BRG1 CUT&Tag showed that BAF occupancy is comparable in SL and 2i (Fig. 5b,c, Extended Data Fig. 5g), implying that global upregulation of pluripotency TFs upon the switch to 2i does not result in BAF upregulation" (lines 338-340).

8. The conclusion that seems somewhat solid is that "RNAPII promoter-proximal pausing (appears to) stabilize BAF chromatin occupancy and enhances nucleosome eviction by BAF." However, the result that RNAPII promoter-proximal pausing is associated with nucleosome eviction was first shown over a dozen years ago (PMID: 21074046)

We appreciate that Reviewer 1 for agreeing with our interpretation that RNAPII promoter-proximal pausing stabilizes BAF chromatin occupancy and enhances nucleosome eviction by BAF. However, we point out that this agreement contradicts Reviewer 1's earlier Points 1 and 2, such as "The assertion that RNAPII occupancy is required for BAF recruitment is particularly strange and does not make sense logically". This is in part due to our use of the metaphor "recruit", which seems to have led to multiple physical interpretations.

It is incorrect that the paper cited by Reviewer 1 (PMID: 21074046) showed this phenomenon already. That study used NELF knockdowns in Drosophila S2 cells to manipulate RNAPII pausing and determined effects on promoter nucleosome occupancy, with no examination of remodeler activity, unlike our study. That paper compared two sets of promoters: in one, nucleosome depletion is mostly intrinsic to DNA sequence, while in the other, DNA sequence favors nucleosome occupancy which is countered by RNAPII pausing. While that paper did not provide any further mechanistic insights, we show that the mechanism involves paused RNAPII stabilizing BAF occupancy and enhancing nucleosome eviction by BAF. We have cited this paper in the Introduction and discussed this point (lines 50-59)

9. Finally, the elaborate speculative Discussion with selective citation of the literature goes far beyond the observations made in this paper.

We have simplified our model and rewritten the Discussion section to better explain the model which we have supported with diverse evidence from the literature.

Reviewer #2:

Remarks to the Author:

What recruits BAF?

Why only dependence at enhancers?

Is it dependent on BAF?

S5/BAF disconnect at bivalent promoters

Why is h3k27me3 more efficiently evicted with fla

See below for our response to these points.

Brahma and Henikoff explore the relationship between RNA Pol II pausing/elongation and chromatin remodeling in mouse ESCs using small molecule inhibitors of transcription. Using a low salt CUT&TAG approach to identify RNA Pol II S5-phosphorylated associated accessible regions (S5 CUTAC) that they previously published, the authors show that BRG1 associates with S5 CUTAC regions. While triptolide decreases both Pol II-S5 and BRG1, flavopiridol and actinomycin D increase Pol II-S5 and BRG1 binding, suggesting that Pol II pausing increases BRG1 occupancy. Further, looking at BRG1-associated nucleosomes, they find that flavopiridol increases eviction of partially remodeled nucleosomes. Flavopiridol and actinomycin D also increase Pol II-S5 and BRG1 occupancy at bivalent promoters, resulting in minor decreases in H3K27me3 and accessible fragment length. Finally, the authors show that mouse ESCs cultured in 2i have increased binding of OCT4, SOX2, NANOG, KLF4 (but not BRG1) than those cultured in LIF+serum. 2i conditions also result in more eviction of BRG1-associated partially remodeled nucleosomes.

This is an interesting study and offers a lot to consider. The data are well-executed and overall propose a new model for the generation of nucleosome depleted regions at transcriptional start sites.

We thank Reviewer 2 for their enthusiasm.

However, the model lacks genetic proof and the data presented are largely correlative. Additional genetic/chemical genetic evidence for BAF and TF requirements for nucleosome eviction are required.

We disagree that the data presented is largely correlative. The power of drug inhibition of transcriptional steps is that we precisely implicate certain steps in gene promoter initiation in chromatin remodeling by acute effects.

Major Points:

1) The BRG1 CUT&RUN.ChIP is a nice assay to show that BRG1-associated fragments smaller than 120bp are nevertheless enriched in histone modifications, showing that these fragments arise from partially remodeled nucleosomes. It is this population that the authors see are preferentially evicted with flavopiridol treatment. However, there is no formal evidence that BAF is evicting these nucleosomes, versus perhaps another co-associated or co-bound remodeler given that other remodelers are also bound at active regulatory regions (Morrison NSMB 2014, Ram Cell 2011). Inhibition studies with a number of different chemical BAF inhibitors (BRM014) or degraders (ACBI1, AU-15330) would formally prove this.

We thank Reviewer 2 (and Reviewer 4) for suggesting BAF inhibition studies to provide direct evidence for the role of BAF in nucleosome eviction that we observed with CUT&RUN.ChIP. We now include these experiments. We performed BRG1 CUT&RUN.ChIP experiments with cells treated with the BRG1 ATPase inhibitor BRM014 or BRM014 + Flavopiridol and include the results in the new Fig. 3. Briefly, the results show increased subnucleosomal fragments enriched in histones over S5P CUTAC sites, confirming that the preferential loss of these

subnucleosomes upon Flavopiridol treatment is dependent on BRG1 ATPase and remodeling activity.

2) Related, prior studies have not shown a strong BAF dependence for promoter/TSS accessibility by ATAC studies, but rather, a preferential dependence at enhancers/distal regulatory elements (Iurlaro Nat Genetics 2021, Schick Nat Genetics 2021, Kelso eLife 2017, Nakayama Nat Genetics 2017). Does eviction of BRG1-associated nucleosomes differ at promoters versus enhancers in flavopiridol treatment? From data in Extended Data Figure 3 c,d, it looks frankly like there is very little change in the broken line between DMSO and flavopiridol. Similarly, following point 1, would the authors see different dependence on BAF at promoters versus enhancers for flavopiridol-induced eviction?

We agree with Reviewer 2 that our results are consistent with previous publications in that enhancer regions show stronger dependence on BAF for accessibility/nucleosome eviction compared to promoter TSSs. We have performed additional data analysis to better demonstrate this difference (revised Extended data Fig. 3) and revised the text in accordance with this observation (lines 228-232). The new BRM014 inhibitor experiments show the same trend, as we observed a weaker retention of subnucleosomal intermediates at promoter regions upon inhibition of BRG1 ATPase.

3) Data from Tolstorukov et al PNAS 2013 demonstrate that genetic inactivation of Brg1 or Snf5 reduces nucleosome occupancy and phasing at the +1 and -1 nucleosomes flanking the NDR. How, if at all, do these results contribute to the authors' model?

Our data primarily explains nucleosome dynamics at promoter and enhancer NDR spaces, so we have restricted our discussion and model to NDRs. While BAF could be acting directly upon +1 and -1 nucleosomes, their reduced occupancy upon SNF5 or BRG1 depletion observed by Tolstorukov et al. is also attributable to these nucleosomes encroaching into the NDR space, which would be consistent with our model for BAF evicting nucleosomes to maintain NDRs. In the absence of BAF function, the -1 and +1 nucleosomes might be acted upon by other remodelers whose function is to oppose BAF or SWI/SNF. This is best demonstrated in yeast (e.g., PMID 31384063). We have now cited relevant literature to discuss potential functions of other remodelers in the absence of BAF, and our new BRG1 inhibition experiments provide additional insights (see Fig. 3 and lines 231-232, and 258-261.).

4) The authors determine that BRG1-associated eviction is stronger in 2i conditions and correlate this with increased TF binding genome wide in 2i conditions. I think it's fair to say that this is highly correlative as a number of known and unknown things are different between LIF+serum and 2i conditions. Again, it would be imperative to show the dependence on pluripotency factors for this productive remodeling using genetic experiments. This could be accomplished for example using addback of TF mutants in a dox-inducible deletion line or by introducing a mutation within a TF binding site using CRISPR editing at a one/several sites in the genome.

The experiments proposed by Reviewer 2 to mutate TFs or their binding sites might show that BAF occupancy and remodeling activity depends on these TFs at these sites, but that has already been established by others (e.g. ref. 45-46, as pointed out below by Reviewer 3, which are now ref. 58-59 in the revised manuscript) The low-throughput one-at-a-time genetic experiments suggested suffer from many knowns and unknowns, such as effects of changing the balance of TFs that have evolved to work in concert with one another, and the redundancy

in TF action that makes results difficult to predict. As for CRISPR editing if we see a drop in BAF occupancy at the edited site, we can only conclude that we've broken something there, without testing our model. To do these non-physiological experiments on a sufficient number and combinations of TFs and sites would take several months or years and are beyond the scope of our study, without testing any critical aspect of our model. Rather, we were testing whether a treatment that modestly upregulates pluripotency TFs has the same effect in promoting nucleosome eviction at these sites as treatments that upregulate paused RNAPII and stabilize BAF, which is the subject of our manuscript. Using a culture medium change to upregulate TFs is not merely correlative, as we document the increases in TF abundances and observe changes specifically at loci that were bound by the TFs in both SL and 2i conditions. This strategy overcomes the drawbacks of ectopically overexpressing TFs which has biased interpretations for TF pioneering activity (see PMIDs: 34984978 and 36253868). Additionally, the culture medium strategy we use mimics stages in embryonic development and is therefore physiologically relevant. Nevertheless, these suggestions have led us to reanalyze our data to determine whether the effects we see with culture change are consistent with TF binding affinities, which differ from site to site based on motif base composition among other factors (revised Extended Data Fig. 5d). Scatterplot analysis of NANOG and KLF4 chromatin binding affinity (as CUT&RUN signal intensity) and S5P CUTAC chromatin accessibility showed positive association with moderately high correlation coefficients. This suggest that increased TF-binding affinity is directly associated with increased chromatin accessibility, as expected if higher TF expression and chromatin binding results in enhanced nucleosome depletion in 2i (revised Fig. 6).

5) Likewise, the authors make bold claims about the lack of global BAF increase in 2i conditions with increased TF binding being conclusive evidence that TFs do not recruit BAF to chromatin. It's not obvious to me that TFs and BAF would scale 1:1 globally in this way as the authors predict.

Reviewers 1 and 3 had similar criticisms of this statement in the Results section, which we have addressed by removing the reference to recruitment and just describing the observation: "BRG1 CUT&Tag showed that BAF occupancy is comparable in SL and 2i (Fig. 5b,c, Extended Data Fig. 5g), implying that global upregulation of pluripotency TFs upon the switch to 2i does not result in BAF upregulation." (lines 338-340).

Further, this is in direct contrast to papers showing a role for TFs in BAF recruitment (Minderjahn Nat Comm 2020, Priam Nat Genetics 2017, Sandoval Mol Cell 2018) and dependence on TFs for BAF binding (King and Klose eLife 2017 in mESCs, others in other cell systems). OCT4 in particular was shown to be required to maintain BAF binding in mESCs using dox-inducible Oct4 deletion (King and Klose eLife 2017). In Figure 4c, among the TFs shown, OCT4 is the one TF that is minimally increased in 2i (actually, the OCT4 data quality looks quite poor – the authors will need to revisit this). The authors could analyze clusters of sites based on gained/maintained/lost TF binding between 2i and serum+LIF conditions and inspect BAF gains at these regions, possibly also separated into promoters and enhancers. However, independent of what the authors find with this clustering, without genetic evidence, this is for me an unsubstantiated argument.

We agree that the above-mentioned papers show dependence on TFs for BAF binding, and we have appropriately discussed and cited several of these and additional relevant literature (lines 299-302). Several of these studies used TF depletion to show that BAF requires TFs for binding, so as noted above in response to point 4, additional genetic experiments to perturb TF-chromatin binding is unlikely to show anything new other than loss of BAF binding and chromatin accessibility. Furthermore, as in the case of OCT4, the same TFs that “recruit” BAF are also dependent on BAF for chromatin binding (King and Klose eLife 2017), showing that genetic experiments cannot resolve this chicken-and-egg problem. Instead, our dynamic cycle model for BAF facilitating TF binding by dynamically unwrapping nucleosomes and TFs driving nucleosome eviction as a function of DNA binding affinity and TF concentration for the cycle to continue, reconciles these observations and is sufficient in explaining the widely observed cooperativity between TFs and BAF for nucleosome eviction. The absence or loss of DNA-binding TFs breaks the cycle, for e.g., at Polycomb domains, where us and others do not observe stable or persistent BAF occupancy at steady state.

We agree that the OCT4 mapping data is not as robust as the other TFs. We repeated OCT4 CUT&RUN with other antibodies which did not produce higher quality data. Nevertheless, these other antibodies also showed that OCT4 binding did not change in 2i.

6) Related to point 4...what does recruit BAF if not TFs? Based on the authors' logic for TFs now applied to RNA Pol II, more BAF in conditions that promote more Pol II binding mean that Pol II recruits BAF. The authors will need to add their thoughts to the discussion.

We have expanded our discussion to clarify our model. We envision a cycle, in which RNAPII, BAF and TFs act independently but successively. Successive independent action in a cycle is different from recruitment as is generally understood, because once a cycle gets started at a regulatory site, such as behind the replication fork, it continues as long as components are in high enough concentration. That is, they do not need to interact with one another such as is implied by a recruitment model to keep the cycle going. We had previously proposed such a cycle to explain the interaction between RSC and general regulatory factors Abf1 and Reb1 that dynamically maintain yeast promoters free of nucleosomes (Brahma & Henikoff, Mol Cell 2019; TiBS2020). A dynamic cycle can explain live-cell imaging results showing dwell times of seconds for most TFs and chromatin remodelers (Lu & Lionnet, 2021; Kim et al., 2021; Tilli et al., 2021. We have also appropriately cited these papers). The cartoon in Fig. 5 was intended to illustrate this concept, but it was complicated in part because we were also trying to also illustrate abortive remodeling that our results suggest occurs over Polycomb domains. We have now simplified the illustration (now Fig. 6) and have rewritten the relevant discussion so that the distinction between our dynamic model and recruitment models are clear.

7) The authors make the observation that Pol II-S5 and BAF can be increased with flavopiridol at bivalent promoters. BRG1-associated H3K27me3 modified nucleosomes are more efficiently evicted under these conditions (and 2i), very similar to H3K4me1 modified histones. Curiously, there seems to be no preference for the different modification states, which is in contrast to work showing that the binding and activity of BAF is regulated by histone PTMs (Mashtalir Science 2021). Despite productive eviction of H3K27me3 modified histones following flavopiridol treatment as measured by the BRG1 CUT&RUN.ChIP assay, this is not sufficient for NDR establishment. This doesn't agree with the authors' model—they link productive eviction following TF binding with NDR establishment (step 3 in Figure 5). This indicates that there is either a flaw in the interpretation of the BRG1 CUT&RUN.ChIP assay or a flaw in the model.

Mashtalir et al. observed only a modest repression of BAF activity with H3K27me3 modified nucleosomes in their in vitro assay. Nevertheless, there are possible explanations for this apparent difference in their in vitro result and our in vivo study that are consistent with both our interpretation of our CUT&RUN.ChIP assay and our model. The in vitro assay using recombinant and synthetically modified histones have both copies of H3 in the nucleosomes modified. One possibility is that BAF activity is not repressed by hemi-methylated H3K27me3 nucleosomes. This would be consistent with our observation of nucleosome eviction at active enhancers and promoters, possibly containing hemi-methylated H3K27me3 due to nucleosome turnover, versus the lack of productive remodeling over Polycomb domains under the same conditions of Flavopiridol inhibition of paused RNAPII release. We have now reconciled our results with that of Mashtalir et al. using this argument in the revision (lines 232-237). Additional difficulties in comparing our in vivo study with theirs is that the effects they observed differed depending on the particular complexes (BAF, PBAF and ncBAF), whereas we do not attempt to differentiate between BAF isoforms in our current study, which is the subject of future studies to follow.

Minor Points: Extended Figure 4b is cut off.

Fixed.

Reviewer #3:

Remarks to the Author:

BAF chromatin remodeling complexes are key for nearly all developmental changes in gene expression, and they have a dominant influence on the accessibility landscape of active genes promoters and enhancers. Previous work has shown that subunits of the ESC-specific BAF complex (esBAF), Brg1, is essential for chromatin accessibility at a large number of active genes in ESCs, and that BAF is essential for ESC differentiation. Furthermore, studies have found that esBAF is dependent on pluripotent transcription factors, such as OCT4, for promoter occupancy. Here, Brahma and Henikoff investigate the role of promoter-proximal paused RNAPII in BAF occupancy and nucleosome depletion at promoters (NDR formation). Using high resolution Cut&Tag methodologies and chemical treatments that block steps in

the RNAPII cycle, the authors present compelling evidence that paused RNAPII facilitates recruitment/retention of Brg1 at active genes. Furthermore, enhancing the level of Brg1/BAF at genes, by increasing the levels of paused RNAPII, leads to more efficient NDR formation.

In general, the data are of high quality and there were no technical concerns. My only issues surround the interpretation of the role of TFs in BAF recruitment, for which the authors have not provided much, if any, compelling data. Their conclusions surrounding the role of TFs need to be tempered considerably in their discussion and the model presented in Figure 7, taking into account other data published previously.

We thank Reviewer 3 for the positive comments.

Specific points:

The authors have over-simplified their description of mammalian BAF complexes. They should note that BAF complexes have several different forms – cBAF, PBAF, ncBAF, and esBAF. The esBAF complex is specific to ESCs and contains the BAF250a subunit, and not BAF200. Note that this complex also lacks the polybromo subunit, BAF180. Both the cBAF and esBAF complexes are most highly related to yeast SWI/SNF, not RSC complex. The RSC complex is most related to the PBAF complex. These distinctions should be taken into account when the authors compare their work in ESCs to their previous work investigating the role of yeast RSC in NDR formation. Perhaps just stating that esBAF is related to yeast RSC, rather than “homologous”.

We had not intended to simplify the description of mammalian BAF complexes, but rather we used the term BAF to apply to all of them. That is because we follow BAF occupancy on chromatin using an antibody to Brg1, which is the catalytic subunit in all BAF complexes in mouse ES cells. Following Reviewer 3’s suggestion we have included a description of BAF complex isoforms (lines 66-70) and have made it clear that we use BAF to collectively refer to all three known BAF complex isoforms in mouse ESCs, all of which contain the BRG1 catalytic subunit (lines 114-117). Also, we now refer to the mammalian PBAF as related to rather than homologous to RSC.

The authors cite key work that has demonstrated essential roles for BAF complexes in promoting the binding of pluripotency TFs in ESCs, however, they fail to mention that these TFs (such as OCT4 and SOX2) are also essential for Brg1/BAF occupancy (ref. 45 and 46). Notably, these previous data used rapid removal of OCT4 or SOX2, leading to loss of BAF.

This is an excellent point, as co-dependence of pluripotency TFs and BAF is expected from our dynamic cycle model, and we now make this point in the revision (lines 299-302).

The authors use different media conditions to vary the expression levels of several pluripotency TFs, and then they monitor the impact on Brg1 occupancy and NDR formation. By changing from SL to 2i, they see a 3x increase in KLF4 occupancy, small increases for SOX2 and NANOG, and no changes in OCT4. Based on these data, the authors made the broad conclusion that pluripotency TFs do not play a role in BAF recruitment (Discussion and Figure 7). This seems to be an over-interpretation of the data. There are several reports that OCT4 is key for Brg1 recruitment to targets in ESCs (see above), and the authors' data do not address this.

Reviewers 1 and 2 had similar criticisms about our interpretation of the observation that there was no change in BAF occupancy over sites where pluripotency TF occupancy increased. In our rewritten Discussion section we clarify the distinction between recruitment models such as that assumed by Reviewer 3 and one in which successive action of RNAPII, BAF and TFs results in a dynamic cycle.

This reviewer is not aware of data suggesting that KLF4 would be the key, limiting TF for BAF recruitment.

We agree with Reviewer 3 that although the association of BAF with NANOG and OCT4 has been studied before, its association with KLF4 was previously unknown. However, given the well-studied role of KLF4 in pluripotency gene regulation (e.g., PMIDs: 23159369, 31548608,

31722212), and that it is one of the Yamanaka factors, its association with BAF and RNAPII for nucleosome eviction in ESCs is not surprising.

Furthermore, it would seem that the authors should mention the large amount of data showing that BAF complexes physical interact with many TFs, and that these TFs can recruit BAF to target loci both in vitro and in vivo (e.g. recent work from the Zaret group in NSMB, and older work on the role of non-DNA binding domains of PU.1).

We are not disputing that physical interactions between BAF and TFs occurs, but rather are providing a dynamic interpretation based on our CUT&RUN.ChIP results. Enhancer and promoters are complex with many potential ways of promoting BAF and RNAPII binding/occupancy, and our data shows an alternative way distinct from TF-mediated recruitment. We think that our rewritten text associated with revised Fig. 5 and Extended Data Fig. 5, revised Discussion section, and simplified model in Fig. 6 will help clarify the distinction between recruitment models, whether or not via physical interactions (with PU.1, OCT4, Drosophila GAF as some examples where interaction has been shown), and our model based on RNAPII, BAF and TFs acting independently but successively in a dynamic cycle. In doing so, we have now cited and discussed previous reports of TFs facilitating BAF binding to target loci via direct protein-protein interaction or other mechanisms.

In the authors' model (Figure 7), but it is not clear to this reviewer how specificity is provided if gene-specific TFs play no role in RNAPII occupancy. If RNAPII occupies all exposed promoters following replication fork passage, then all genes and cryptic promoters would be occupied. Their model may explain maintenance of an established transcription program, but it does not explain how cell-type specificity is established. This model would predict that loss of TFs, like OCT4, would have little impact on the levels of paused RNAPII. If this has not been shown previously, then perhaps this could be tested.

Our experiments do not address the question of establishment, but rather maintenance of accessibility once established, and we have replaced the term "hypothesize" with "speculate" to make clear that this is only one plausible scenario, and now mention others. For example, we

had previously shown that in Drosophila cells, replication fork passage results in conspicuous changes at promoters that have high levels of RNAPII stalling and DNA accessibility and show specific enrichment for Brahma, the catalytic subunit of the BAF complex, but not other remodeler families (INO80, CHD1 and ISWI) (Ramachandran and Henikoff Cell 2016). This suggests that our dynamic cycle model for events occurring throughout the cell cycle in mESCs reflects what we showed at replication forks in S2 cells. Furthermore, in a very recent preprint from Constance Alabert's lab that became available after the initial review of our manuscript, metabolic labeling of nascent chromatin followed by quantitative Mass spectrometry (iPOND technique) in mouse ESCs showed that both paused RNAPII and subunits of the BAF complex including BRG1 (SMARCA4) are enriched on nascent chromatin and paused RNAPII stabilizes BRG1 on nascent chromatin; further supporting our model (<https://doi.org/10.1101/2023.04.19.537523>).

Reviewer #4:

Remarks to the Author:

In the presented study, Brahma and Henikoff use a wide array of assays (CUT&Tag, S5P CUTAC, CUT&RUN.ChIP) in combination with RNAPII inhibition to test the role of RNAPII in targeting BRG1 (~SWI/SNF) to promoter NDRs in mESCs. In addition, they assessed the role of the pluripotency transcription factors (Nanog, Sox2, Oct4 and Klf4) in NDR formation. The authors used three different inhibitors of RNAPII transcription to either block initiation (Triptolite) or block elongation (Flavopiridol and Actinomycin D). Blocking elongation results in the accumulation of paused RNAPII. Interestingly, increased RNAPII pausing is accompanied by increased BRG1 occupancy at promoter NDRs, and across the genome (Fig. 1 & extended data Fig. 1 & 2). Flavopiridol also promotes increased nucleosome eviction at NDRs (Fig. 2). Transcription elongation inhibition also leads to increased RNAPII and BRG1 occupancy at Polycomb-repressed promoters (Fig. 3), suggesting these factors transiently probe these regions. Finally, the authors show that the pluripotency transcription factors promote NDR formation (Fig. 4). These findings are summarized in a model showing that transcription factors and RNAPII cooperate in the recruitment of the SWI/SNF remodeler to establish/maintain NDRs at promoters.

The work described in this manuscript is a technical tour-de-force and the data is

convincing. Although the interplay between sequence-specific transcription factors, RNAPII and remodelers have been described extensively, the results presented here provide new insights into the recruitment of the SWI/SNF remodeler to promoters. However, in my opinion this study is in need of additional functional experiments.

We thank Reviewer 4 for the enthusiastic comments.

Major concerns:

1) It is important to show that the changes in chromatin structure are actually dependent on BRG1. This could e.g. be tested by using the inhibitor BRM014. Additionally, the effect of BRG1 inhibition (or loss) on gene transcription needs to be included in the analysis.

We thank Reviewer 4 (and Reviewer 2) for suggesting BAF inhibition studies to provide direct evidence for the role of BAF in nucleosome eviction that we observed with CUT&RUN.ChIP. We now include these experiments. We performed BRG1 CUT&RUN.ChIP experiments with cells treated with BRM014 or BRM014 + Flavopiridol and include the results in new Fig. 3. Briefly, the results show increased subnucleosomal fragments enriched in histones over S5P CUTAC sites, confirming that the preferential loss of these subnucleosomes upon Flavopiridol treatment is dependent on BRG1 ATPase and remodeling activity.

The effects of BRG1/BAF loss on gene transcription has been studied by several groups (e.g., PMIDs: 28287392, 34117481, 34731603) which we have cited in the revised manuscript, and Karen Adelman's lab has a very recent preprint on the effects of BRG1 inhibition using BRM014 on RNAPII transcription (doi.org/10.1101/2023.03.07.531379). This preprint appeared while our manuscript was in review, and nicely complements our work.

2) The authors emphasize BRG1 recruitment by paused RNAPII. However, the effects of transcriptional inhibition are apparently not restricted to promoter regions. It is imperative that a clear overview is presented of all BRG1 peaks and all RNAPII peaks and to what

extend they overlap. The ext. data provided in Fig. 1d and Fig. 2c-j does not suffice.

We have shown the extent of overlap between paused RNAPII (RNAPII-S5P) and BAF (BRG1) as a scatterplot in Extended Data Fig 1e, and the Pearson's correlation coefficient of 0.74 shows that they strongly correlate. This scatterplot represents CUT&Tag signal in genome-wide 1kb consecutive non-overlapping bins and is not restricted to select sets of genomic regions such as promoters. This is a more unbiased approach than comparing peaks. In addition, we performed similar analysis with RNAPII-S5P and BRG1 CUT&Tag after transcription inhibition, which confirmed their strong genome-wide correspondence (Extended Data Fig. 2f).

3) Related to point 2), S5P-CUTAC selects for paused RNAPII loci. What is the effect on SWI/SNF remodeling at intergenic loci? This needs to be addressed by including ATAC-seq.

We have previously shown that S5P-CUTAC detects the same sites of accessibility as ATAC-seq, but more sensitively with higher signal-to-noise based on peak-calling using MACS2 (Janssens et al. Genome Biology (2022), Figure 1C shown below). Our analysis would therefore include transposase-accessible sites within intergenic loci. We make this point in our current manuscript as well and show that RNAPII-S5P CUTAC in mESCs recapitulates ATAC-seq and maps immediately upstream of TSSs within genic promoters and well as START-seq mapped nascent RNA transcription start sites which include intergenic enhancer regions (Extended Data Fig. 1f). Our analysis of RNAPII-S5P and BRG1 occupancy as well as BAF remodeling (CUT&RUN.ChIP) includes promoter-distal regulatory regions which are intergenic (Extended Data Figs. 1d, 2g-h, 3e-f)

Janssens et al. *Genome Biology* (2022), Figure 1C:

To evaluate the data quality of RNAPII-S5P (Pol2S5p) CUTAC, random samples of mapped fragments were drawn, mitochondrial reads were removed and MACS2 was used to call (narrow) peaks. The number of peaks called for each sample (left) is a measure of sensitivity, and the fraction of reads in peaks (FRiP, right) is a measure of specificity calculated for each sampling in a doubling series from 50,000 to 6.4 million fragments. For comparison, an ENCODE ATAC-seq sample was used for K562 cells and a published ATAC-seq sample from our lab (GSE128499) was used for H1 cells. Hex samples were tagged in the presence of 10% 1,6-hexanediol.

4) My understanding from the manuscript is that transcription inhibition with Flavopiridol also promotes BRG1 recruitment to intergenic loci. Assuming these sites lack RNAPII, doesn't this contradict with the model presented?

This assumption is incorrect. As explained in response to point 3 above, intergenic loci such as enhancer regions are enriched for both RNAPII and BAF, and our analysis include these regions. Furthermore, the scatterplot presented in Extended Data Figs. 1e and 2f show that paused RNAPII and BRG1 occupancy strongly correlate genome wide, including intergenic regions.

Minor comments:

5) Could drug-induced RNAPII pausing simply create a physical barrier to nucleosome sliding, resulting in increased eviction?

Reduced nucleosome sliding due to a physical barrier will cause nucleosomes to build up upstream of the barrier, or the promoter and enhancer NDR spaces, which is opposite of what we and others have observed (PMIDs 21074046, 31384063).

6) I don't want to start a discussion on bivalency, but the heatmaps in Fig. 3b suggest to me that the sites labelled " bivalent" are simply Polycomb-repressed. H3K27me3 is much more prevalent than H3K4me3.

Yes, we agree, and in the revised manuscript we now only refer to Polycomb-group protein repressed (PcG-repressed) domains without using the term bivalent.

7) There is a graded scale between unwrapping and eviction. The effects are presented a bit too much as all or nothing.

Previous biochemical studies have shown that SWI/SNF family remodeling complexes partially unwrap or unwind DNA from nucleosomes (e.g., PMID: 12620227), which appear to be stable intermediates that facilitate TF binding (e.g., PMID: 8702824), eventually leading to eviction (e.g., PMID: 32496195). Our in situ CUT&RUN.ChIP assay shows the population average of these biochemical activities at gene promoters and enhancers under steady state, while transcription inhibition or upregulated TF activity drives it to stable or near-complete nucleosome depletion. In our experiment using the BRM014 catalytic inhibitor of Brg1, which Reviewers 2 and 4 suggested, we observed unwrapping without eviction, which confirms that partial unwrapping and complete eviction are distinct steps in the process of nucleosome core removal (new Figure 3).

Decision Letter, first revision:

10th Aug 2023

Dear Steve,

Your Article, entitled "RNA Polymerase II, the BAF remodeler and transcription factors synergize to evict nucleosomes", has now been seen by the 4 original referees. You will see from their comments below that while they find your work improved and reviewers #2-#4 are satisfied overall, some important points are raised reviewer #1. Reviewer #1 acknowledges that the paper has improved. However, they continue to think that the conclusions are not fully supported by the data and that, even if they were, at this stage they do not warrant publication in Nature Genetics.

We are interested in the possibility of publishing your study in Nature Genetics, but would like to consider your response to these concerns in the form of a revised manuscript before we make a final decision on publication.

We therefore invite you to revise your manuscript taking into account all reviewer comments. Please highlight all changes in the manuscript text file. At this stage we will need you to upload a copy of the manuscript in MS Word .docx or similar editable format.

We are committed to providing a fair and constructive peer-review process. Do not hesitate to contact me if there are specific requests from the reviewers that you believe are technically impossible or unlikely to yield a meaningful outcome.

*1) Include a "Response to referees" document detailing, point-by-point, how you addressed each referee comment. If no action was taken to address a point, you must provide a compelling argument. This response may be sent back to the referees along with the revised manuscript.

*2) If you have not done so already please begin to revise your manuscript so that it conforms to our Article format instructions, available [here](http://www.nature.com/ng/authors/article_types/index.html). Refer also to any guidelines provided in this letter.

Please be aware of our <https://www.nature.com/nature-research/editorial-policies/image->

integrity">guidelines on digital image standards.

[redacted]

We hope to receive your revised manuscript within eight weeks. If you cannot send it within this time, please let us know.

Sincerely,

Tiago

Tiago Faial, PhD
Chief Editor
Nature Genetics
<https://orcid.org/0000-0003-0864-1200>

Reviewers' Comments:

Reviewer #1:

Remarks to the Author:

The authors have addressed nicely some concerns of reviewers, and in many cases, the rewriting of the text and inclusion of relevant information and additional references have made this much more

readable, and the paper integrates better the strong existing literature on TF recruitment of SWI/SNF remodelers. Some serious concerns remain and some of these are mentioned below. However, the major limitation of this work, in my opinion, is that the main new conclusion, that pause Pol II at promoters may enhance BRG binding (described also as “facilitate” or “facilitate its retention” or “promotes BAF chromatin occupancy”), is not supported with sufficiently strong data nor is this conclusion on its own important enough to warrant publication in a high-profile journal like Nature Genetics. This work seems more appropriately targeted for another journal, like Genetics or PLoS Genetics, once the final issues are resolved.

Specific concerns:

1. The kinetics of loss of RNAPII-S5P peak signals in response to the initiation inhibitor triptolide Fig. 1cd is considerably faster than the loss of BRG1. While the gain in RNAPII-S5P in response to the pause release inhibitor flavopiridol does not quantitatively correlate with BRG1 peak signals (Fig. 1ef). While one can argue around this, this does leave the reader with a sense that the role of paused RNAPII on BRG1 occupancy may not be direct or easily interpreted. The actinomycin results are particularly hard to interpret, as this drug affects DNA structure broadly and would influence the progress of both paused and elongating RNAPII as well as many other factor and nucleosome interactions with DNA. Also, in Fig 2, the 8hr treatment with flavopiridol is much longer than the time needed to affect RNAPII occupancy at promoters. The argument that the “Colony morphology looked normal under the microscope and there was no reduction in cell number” is mildly reassuring, but secondary effects that change promoter architecture may take a long time to show up as a change in cell appearance.
2. The 2i treatment of mESCs can have many effects on cells. To ascribe the changes in BAF-associated nucleosomes to modest increases in NANOG and KLF4 seems risky to me without additional experiments that more specifically target these factors.
3. The speculation in the discussion that “RNAPII broadly scans chromatin for exposed promoter DNA and utilizes the window of opportunity in the wake of DNA replication to bind and initiate transcription RNAPII pausing would then promote BAF occupancy (step 3) to clear away nucleosomes that encroach into the NDR space upstream” is strongly at odds with published, compelling findings that promoter-proximal pausing depends on TFs, especially TFs that have known association with remodelers.

Reviewer #2:

Remarks to the Author:

The authors have largely addressed my concerns. I appreciated revisions pertaining to:

- 1) Use of BRM014 to establish dependence on BAF ATPase activity (new Figure 3)
- 2) Clarification of BAF activity at promoters versus enhancers (new Ext Fig 3e,f)
- 3) Clarification of the model and specifically the role of TF in BAF recruitment (line 339-340)

Points to consider in revising the text:

- 1) Concerning the role of other remodelers at promoters and the authors’ new findings that nucleosome sized fragments are depleted upon dual inhibition by Fla and BRM014, the authors may consider discussing new findings on the role of EP400 at promoters from the Adelman lab (<https://www.biorxiv.org/content/10.1101/2023.03.07.531379v1>).

2) Concerning the discussion positing that RNA Pol II scans the DNA following DNA replication and promotes BAF occupancy (382-386), the authors should incorporate new findings from the Roberts lab (<https://www.nature.com/articles/s41586-023-06085-6>) indicating that BAF complex subunits SMARCB1/E1 remain bound to promoters during mitosis in mouse ESCs. This would potentially place BAF and 'bookmarked' TFs upstream of Pol II.

Reviewer #3:

Remarks to the Author:

The authors have done an outstanding job at revising their text, tempering their conclusions, and more clearly describing their hypothesis. Addition of the BAF inhibitor studies is a strong new addition. This is a very strong manuscript that makes an excellent contribution.

Reviewer #4:

Remarks to the Author:

This manuscript provides an interesting contribution to the increasingly more detailed literature on remodelers. Ties in nicely with recent work from the Adelman lab.

I think it is unfair to expect the authors to deal with the already enormous literature on remodeler recruitment (esp. since the mESC literature tends to ignore earlier work in yeast and flies that uncovered the main principles of remodeler function).

Author Rebuttal, first revision:

Point-by-point response to Reviewers' comments:

Reviewer #1:

Remarks to the Author:

The authors have addressed nicely some concerns of reviewers, and in many cases, the rewriting of the text and inclusion of relevant information and additional references have made this much more readable, and the paper integrates better the strong existing literature on TF recruitment of SWI/SNF remodelers. Some serious concerns remain and some of these are mentioned below. However, the major limitation of this work, in my opinion, is that the main new conclusion, that pause Pol II at promoters may enhance BRG binding (described also as "facilitate" or "facilitate its retention" or "promotes BAF chromatin occupancy"), is not supported with sufficiently strong data nor is this conclusion on its own important enough to warrant publication in a high-profile journal like Nature Genetics. This

work seems more appropriately targeted for another journal, like Genetics or PLoS Genetics, once the final issues are resolved.

We thank Reviewer #1 for finding the revised manuscript to be considerably improved.

In the first round of review, Reviewer #1 had agreed with our conclusion (point 8 of the previous review report: *"The conclusion that seems somewhat solid is that RNAPII promoter-proximal pausing (appears to) stabilize BAF chromatin occupancy and enhances nucleosome eviction by BAF."* However, now Reviewer #1 appears to contradict that more positive earlier opinion despite finding the paper improved.

We wish to emphasize that the BRG1 ATPase inhibition experiments (new Figure 3) suggested by both Reviewers #2 and #4 further solidifies that RNAPII promotes BAF occupancy leading to enhanced nucleosome eviction and provides additional new insights. However, Reviewer #1 seems to have overlooked these new functional experiments and the strong additional evidence they provided.

In contrast to the opinion of Reviewer #1, Reviewers #2 (in the first round of review) and #3 and #4 (in the first and second rounds) expressed the opinion that our work provides a compelling new model and important conceptual advance for how factor dynamics modulate nucleosome organization and chromatin accessibility.

Specific concerns:

1. The kinetics of loss of RNAPII-S5P peak signals in response to the initiation inhibitor triptolide Fig. 1cd is considerably faster than the loss of BRG1. While the gain in RNAPII-S5P in response to the pause release inhibitor flavopiridol does not quantitatively correlate with BRG1 peak signals (Fig. 1ef). While one can argue around this, this does leave the reader with a sense that the role of paused RNAPII on BRG1 occupancy may not be direct or easily interpreted. The actinomycin results are particularly hard to interpret, as this drug affects

DNA structure broadly and would influence the progress of both paused and elongating RNAPII as well as many other factor and nucleosome interactions with DNA.

Reviewer #1 had raised this point earlier and we provided mechanistic explanations for the quantitative differences and cited relevant literature, which the other reviewers found satisfactory. Reviewer #1 appears to have missed the logic of these experiments: Triptolide and flavopiridol directly inhibit RNAPII itself, and this is why we chose these drugs. Thus, the effects of these drugs on BAF must be due to disruption of RNAPII. The faster response of RNAPII to triptolide is exactly as expected, and the delay in BAF loss indicates the persistence of previously loaded BAF on active promoters. We have now discussed these points more elaborately in the revision (lines 150-173).

Also, in Fig 2, the 8hr treatment with flavopiridol is much longer than the time needed to affect RNAPII occupancy at promoters. The argument that the "Colony morphology looked normal under the microscope and there was no reduction in cell number" is mildly reassuring, but secondary effects that change promoter architecture may take a long time to show up as a change in cell appearance.

Again, we had already responded to this comment in the first round of reviews and explained our controls. Reviewer #1 seems to have overlooked the fact that we did the 8hr treatment only for the CUT&RUN.ChIP experiment to show more robust effects. All other experiments including those in the later figures were up to 4 hours, and it was the 4 hr experiments that led to most of our major conclusions.

It should be noted that a change in promoter structure is not a "secondary effect", but rather a direct consequence of increased paused RNAPII and BAF occupancy causing enhanced nucleosome eviction, which is a critical observation we make in the manuscript.

2. The 2i treatment of mESCs can have many effects on cells. To ascribe the changes in BAF-

associated nucleosomes to modest increases in NANOG and KLF4 seems risky to me without additional experiments that more specifically target these factors.

Multiple reviewers raised this issue in the first review round, and our revisions satisfied the other reviewers. Reviewer #1 seems to have overlooked the fact that we detected these changes specifically at NANOG- and KLF4-bound promoters, which are increased by the treatment. Thus, these changes do indeed demonstrate a specific effect. We also showed that the effects we see with media change are consistent with NANOG and KLF4 binding affinities, which differ from site to site based on motif base composition among other factors (Extended Data Fig. 5d), which was an analysis Reviewer #2 had suggested.

3. The speculation in the discussion that “RNAPII broadly scans chromatin for exposed promoter DNA and utilizes the window of opportunity in the wake of DNA replication to bind and initiate transcription RNAPII pausing would then promote BAF occupancy (step 3) to clear away nucleosomes that encroach into the NDR space upstream” is strongly at odds with published, compelling findings that promoter-proximal pausing depends on TFs, especially TFs that have known association with remodelers.

We agree that this statement is strongly at odds with the *perception* of published findings. Our point is that the static appearance of promoter chromatin structures hides the underlying dynamics, such as widespread transient scanning by RNAPII. These are not incompatible ideas.

As we had already pointed out in the first round of review, our model that RNAPII rapidly associates with newly replicated chromatin is strongly supported by orthogonal studies, which we have cited in the manuscript. For example:

- Transcription Restart Establishes Chromatin Accessibility after DNA Replication - Anja Groth's lab (PMID 31126739)
- Transcriptional Regulators Compete with Nucleosomes Post-replication – Henikoff lab (PMID 27062929)
- Transcription promotes the restoration of chromatin following DNA replication - Constance Alabert's lab (biorXiv, 2023)

Additionally, a just-published Nature article from Alex Mazo's lab shows direct evidence of RNAPII binding to both newly-synthesized DNA strands and RNAPII likely transiting with PCNA, which we now cite in the revision (lines 392-394).

- RNA polymerase II associates with active genes during DNA replication (PMID 37468626)

Reviewer #2:

Remarks to the Author:

The authors have largely addressed my concerns. I appreciated revisions pertaining to:

- 1) Use of BRM014 to establish dependence on BAF ATPase activity (new Figure 3)
- 2) Clarification of BAF activity at promoters versus enhancers (new Ext Fig 3e,f)
- 3) Clarification of the model and specifically the role of TF in BAF recruitment (line 339-340)

We thank Reviewer #2 for highlighting the major improvements in the revised manuscript, including the BRG1 inhibition experiment which in addition to proving BAF ATPase dependence for nucleosome eviction, provides critical new insights into chromatin dynamics at gene regulatory elements.

Points to consider in revising the text:

- 1) Concerning the role of other remodelers at promoters and the authors' new findings that nucleosome sized fragments are depleted upon dual inhibition by Fla and BRM014, the authors may consider discussing new findings on the role of EP400 at promoters from the Adelman lab (<https://www.biorxiv.org/content/10.1101/2023.03.07.531379v1>).

We have cited this new preprint article from the Adelman lab (line 264).

- 2) Concerning the discussion positing that RNA Pol II scans the DNA following DNA replication and promotes BAF occupancy (382-386), the authors should incorporate new findings from the Roberts lab (<https://www.nature.com/articles/s41586-023-06085-6>)

indicating that BAF complex subunits SMARCB1/E1 remain bound to promoters during mitosis in mouse ESCs. This would potentially place BAF and 'bookmarked' TFs upstream of Pol II.

We agree and have now cited the Roberts lab paper (which was published after we had revised our manuscript) and another paper to discuss mitotic bookmarking (lines 402-404).

Reviewer #3:

Remarks to the Author:

The authors have done an outstanding job at revising their text, tempering their conclusions, and more clearly describing their hypothesis. Addition of the BAF inhibitor studies is a strong new addition. This is a very strong manuscript that makes an excellent contribution.

We thank Reviewer #3 for enthusiastically supporting our manuscript for publication.

Reviewer #4:

Remarks to the Author:

This manuscript provides an interesting contribution to the increasingly more detailed literature on remodelers. Ties in nicely with recent work from the Adelman lab.

I think it is unfair to expect the authors to deal with the already enormous literature on remodeler recruitment (esp. since the mESC literature tends to ignore earlier work in yeast and flies that uncovered the main principles of remodeler function).

We thank Reviewer #4 for enthusiastically supporting our manuscript for publication.

Decision Letter, second revision:

18th Sep 2023

Dear Steve,

Thank you for submitting your revised manuscript entitled "RNA Polymerase II, the BAF remodeler and transcription factors synergize to evict nucleosomes" (NG-A61775R1). My colleagues and I find that the paper has improved in revision, and therefore we'll be happy in principle to publish it in Nature Genetics, pending minor revisions to comply with our editorial and formatting guidelines.

We will be now performing detailed checks on your paper and will send you a checklist detailing our editorial and formatting requirements soon. Please do not upload the final materials and make any revisions until you receive this additional information from us.

Thank you again for your interest in Nature Genetics. Please do not hesitate to contact me if you have any questions.

Congratulations!

Sincerely,

Tiago

Tiago Faial, PhD
Chief Editor
Nature Genetics
<https://orcid.org/0000-0003-0864-1200>

Final Decision Letter:

30th Oct 2023

Dear Steve,

I am delighted to say that your manuscript "The BAF chromatin remodeler synergizes with RNA polymerase II and transcription factors to evict nucleosomes" has been accepted for publication in an upcoming issue of Nature Genetics.

Your paper will be published online after we receive your corrections and will appear in print in the next available issue. You can find out your date of online publication by contacting the Nature Press Office (press@nature.com) after sending your e-proof corrections. Now is the time to inform your Public Relations or Press Office about your paper, as they might be interested in promoting its publication. This will allow them time to prepare an accurate and satisfactory press release. Include your manuscript tracking number (NG-A61775R2) and the name of the journal, which they will need when they contact our Press Office.

Please note that *Nature Genetics* is a Transformative Journal (TJ). Authors may publish their research with us through the traditional subscription access route or make their paper immediately open access through payment of an article-processing charge (APC). Authors will not be required to make a final decision about access to their article until it has been accepted. [Find out more about Transformative Journals](https://www.springernature.com/gp/open-research/transformative-journals)

Authors may need to take specific actions to achieve [compliance with funder and institutional open access mandates](https://www.springernature.com/gp/open-research/funding/policy-compliance-faqs). If your research is supported by a funder that requires immediate open access (e.g. according to [Plan S principles](https://www.springernature.com/gp/open-research/plan-s-compliance)) then you should select the gold OA route, and we will direct you to the compliant route where possible. For authors selecting the subscription publication route, the journal's standard licensing terms will need to be accepted, including [self-archiving-and-license-to-publish](https://www.nature.com/nature-portfolio/editorial-policies/self-archiving-and-license-to-publish). Those licensing terms will supersede any other terms that the author or any third party may assert apply to any version of the manuscript.

If you have posted a preprint on any preprint server, please ensure that the preprint details are updated with a publication reference, including the DOI and a URL to the published version of the

article on the journal website.

Sincerely,

Tiago

Tiago Faial, PhD
Chief Editor
Nature Genetics
<https://orcid.org/0000-0003-0864-1200>